# A Unified View on Learning Unnormalized Distributions via Noise-Contrastive Estimation

J. Jon Ryu [1]   Abhin Shah [1]   Gregory W. Wornell [1]

## Abstract

This paper studies a family of estimators based on noise-contrastive estimation (NCE) for learning unnormalized distributions. The main contribution of this work is to provide a unified perspective on various methods for learning unnormalized distributions, which have been independently proposed and studied in separate research communities, through the lens of NCE. This unified view offers new insights into existing estimators. Specifically, for exponential families, we establish the finite-sample convergence rates of the proposed estimators under a set of regularity assumptions, most of which are new.

## 1. Introduction

Unnormalized distributions, also known as energy-based models, arise in various applications, such as generative modeling, density estimation, and reinforcement learning; we refer an interested reader to a comprehensive overview paper (Song & Kingma, 2021) and references therein. Such distributions capture complex dependencies and provide representational flexibility, making them attractive in fields ranging from statistical physics to machine learning. Despite their widespread use, estimating parameters within these models poses significant challenges due to the intractability of their normalization constants.

In this paper, we consider the problem of parameter estimation for unnormalized distributions, through the lens of the noise-contrastive estimation (NCE) framework (Gutmann & Hyvärinen, 2012). Our contributions are as follows:

1. As variants of the $f$-NCE (Pihlaja et al., 2010) (Sec. 1.2), we study a family of NCE-based estimators, the $\alpha$-centered NCE ($\alpha$-CentNCE; Sec. 2.1) and $f$-conditional NCE ($f$-CondNCE; Sec. 2.2). With this unifying view on different estimators, we clarify previously unrecognized and/or potentially misleading connections among existing estimators proposed for learning unnormalized distributions, as well as provide unified analysis.

2. Specifically, via the lens of $\alpha$-CentNCE, we reveal that several different estimators for learning unnormalized distributions can be connected and unified, including MLE (Fisher, 1922), MC-MLE (Geyer, 1994), and GlobalGISO (Shah et al., 2023) as special instances. A *local* version of centered NCE estimators subsumes pseudo likelihood (Besag, 1975) and interaction screening objectives (ISO) (Vuffray et al., 2016; 2021; Ren et al., 2021; Shah et al., 2021a), which were proposed for learning exponential families corresponding to Markov random fields (MRFs).

3. For $f$-CondNCE, we show that, in contrast to the original claim in (Ceylan & Gutmann, 2018), the behavior of the $f$-CondNCE estimator does *not* converge to the score matching (SM) estimator (Hyvärinen, 2005) in a small noise regime. In fact, we show that the variance of $f$-CondNCE diverges in the vanishing noise regime, if the number of conditional samples is not sufficiently large.

4. As a concrete consequence of such connections, we establish the finite-sample convergence guarantees of the proposed estimators for learning bounded exponential family distributions, by building upon the analysis of GlobalGISO by (Shah et al., 2023). To the best of our knowledge, such guarantees are the first of the type for almost all the NCE estimators considered in this paper.

### 1.1. Related Work

While the celebrated maximum likelihood estimator (MLE), advocated by Fisher (1922), is arguably the de facto standard for parameter estimation problems, it is not directly applicable for high-dimensional unnormalized distributions due to the computational intractability of calculating the normalization constant. Several methods have been proposed as alternatives, including MLE with Monte-Carlo approximation of partition function (MC-MLE) (Geyer, 1994; Riou-Durand & Chopin, 2018; Jiang et al., 2023), score match-

[1] Department of EECS, MIT, Cambridge, Massachusetts, USA. Correspondence to: J. Jon Ryu <jongha.ryu@gmail.com>.

*Proceedings of the 42nd International Conference on Machine Learning*, Vancouver, Canada. PMLR 267, 2025. Copyright 2025 by the author(s).

$$\mathcal{L}_f^{\mathsf{nce}}(\phi_\theta; q_\mathsf{d}, q_\mathsf{n}) \triangleq \mathbb{E}_{q_\mathsf{n}(x)}\left[\Delta_f\left(\frac{q_\mathsf{d}(x)}{\nu q_\mathsf{n}(x)}, \frac{\phi_\theta(x)}{\nu q_\mathsf{n}(x)}\right)\right] - \mathbb{E}_{q_\mathsf{n}(x)}\left[f\left(\frac{q_\mathsf{d}(x)}{\nu q_\mathsf{n}(x)}\right)\right] \tag{1}$$

$$= -\frac{1}{\nu}\mathbb{E}_{q_\mathsf{d}(x)}[f'(\rho_\theta(x))] + \mathbb{E}_{q_\mathsf{n}(x)}[\rho_\theta(x)f'(\rho_\theta(x)) - f(\rho_\theta(x))]. \tag{2}$$

ing (Hyvärinen, 2005; 2007; Song et al., 2020; Liu et al., 2022; Pabbaraju et al., 2023), NCE (Gutmann & Hyvärinen, 2012; Pihlaja et al., 2010; Gutmann & Hirayama, 2011; Ceylan & Gutmann, 2018; Uehara et al., 2018; Chehab et al., 2022; 2023), contrastive divergence (Hinton, 2002), among many other techniques. A comprehensive overview of these methods can be found in (Song & Kingma, 2021).

For exponential families, there is a specialized literature, with a focus on learning undirected graphical models such as MRFs. In a pioneering work (Besag, 1975), Besag proposed the so-called *pseudo likelihood* estimator, which can be understood as a *local* counterpart of MLE. A recent and representative line of recent work includes ISO, GISO, and ISODUS, based on an estimation principle called *interaction screening* (Vuffray et al., 2016; 2021; Ren et al., 2021; Shah et al., 2021a). More broadly, for exponential family in general, Shah et al. (2021b), and in a follow-up work with refinement in (Shah et al., 2023), studied a variant of the interaction screening objective for training a general exponential family without a local structure, which we refer to as *GlobalGISO* in this paper. We emphasize that these estimators have been proposed and analyzed in several different communities, and the literature lacks on a comprehensive understanding how different estimators can be compared. In this paper, our primary goal is to provide a unifying view on these different principles for learning unnormalized distributions in a unified way via the NCE principle (Gutmann & Hyvärinen, 2012; Pihlaja et al., 2010).

### 1.2. Preliminaries: $f$-Noise-Contrastive Estimation

We consider an unnormalized density model $\{\phi_\theta(x) \colon \theta \in \Theta\}$ for a $d$-dimensional random vector $x$ with support $\mathcal{X} \subset \mathbb{R}^d$, where $\theta \in \mathbb{R}^p$ is a parameter and $\Theta \subset \mathbb{R}^p$ is the set of feasible parameters. Our goal is to find the best $\theta \in \Theta$ so that $\phi_\theta(x)$ is closest possible to the data generating distribution $q_\mathsf{d}(x)$. We consider the well-specified case, where there exists $\theta^\star \in \Theta$ such that $\phi_{\theta^\star}(x) \propto q_\mathsf{d}(x)$.

We start the investigation with an extension of the original NCE (Gutmann & Hyvärinen, 2012), which we call $f$-NCE. This family of estimators was first derived in (Pihlaja et al., 2010) in a rather convoluted way. Here, we introduce them as an instance of Bregman divergence minimization for density ratio estimation (DRE) (Sugiyama et al., 2012), in which way the consistency of the resulting estimator is

straightforward.

The idea of NCE is to train the model $\phi_\theta(x)$, so that it can be used to discriminate samples of the data distribution $q_\mathsf{d}(x)$ from samples of a *noise* (or reference) distribution $q_\mathsf{n}(x)$. A necessary condition for discrimination is that the support of $q_\mathsf{n}$, i.e., $\mathsf{supp}(q_\mathsf{n})$, subsumes the support of $q_\mathsf{d}(x)$, i.e., $\mathsf{supp}(q_\mathsf{d})$. Hence, we define the (scaled) model density ratio $\rho_\theta(x) \triangleq \frac{\phi_\theta(x)}{\nu q_\mathsf{n}(x)}$ for a hyperparameter $\nu > 0$, and we wish to fit this to the underlying density ratio $\frac{q_\mathsf{d}(x)}{\nu q_\mathsf{n}(x)}$. For a differentiable function $h \colon \mathcal{Z} \to \mathbb{R}$ with $\mathcal{Z} \subset \mathbb{R}^k$, we define and denote the *Bregman divergence* as

$$\Delta_h(\mathbf{z}, \mathbf{z}') \triangleq h(\mathbf{z}) - h(\mathbf{z}') - \langle \nabla h(\mathbf{z}'), \mathbf{z} - \mathbf{z}' \rangle$$

for $\mathbf{z}, \mathbf{z}' \in \mathcal{Z}$, which is the approximation error of the first-order Taylor approximation of $h(\mathbf{z})$ at $\mathbf{z}'$. For a given strictly convex function $f \colon \mathbb{R}_{\geq 0} \to \mathbb{R}$ and a reference distribution $q_\mathsf{n}(x)$, we propose the $f$-NCE objective as in Eq. (1). The intermediate expression in Eq. (1) is used as a conceptual device to derive the final objective in Eq. (2). We define the $f$-NCE estimator as a minimizer of the objective function:

$$\theta_f^{\mathsf{nce}}(q_\mathsf{d}, q_\mathsf{n}) \in \arg\min_{\theta \in \Theta} \mathcal{L}_f^{\mathsf{nce}}(\phi_\theta; q_\mathsf{d}, q_\mathsf{n}).$$

Given data samples $x_1, \ldots, x_{n_\mathsf{d}}$ drawn from $q_\mathsf{d}(x)$ and noise samples $x_1', \ldots, x_{n_\mathsf{n}}'$ from $q_\mathsf{n}(x)$, the empirical estimator is $\theta_f^{\mathsf{nce}}(\hat{q}_\mathsf{d}, \hat{q}_\mathsf{n})$, where $\hat{q}_\mathsf{d}$ and $\hat{q}_\mathsf{n}$ denote the corresponding empirical distributions. We remark that directly inheriting the property of the Bregman divergence, the $f$-NCE objective is invariant to adding or subtracting a linear function and translation by constants; see Appendix B.1.1 for a formal statement.

By constructing the $f$-NCE objective in terms of a Bregman divergence, we can easily prove that the objective is *consistent* in the population limit, which we call *Fisher consistency*, provided that the generating function $f$ is strictly convex and the model is well-specified.

**Proposition 1.1** ($f$-NCE: Fisher consistency). *Let $f \colon \mathbb{R}_{\geq 0} \to \mathbb{R}$ be a strictly convex function and assume $\mathsf{supp}(q_\mathsf{d}) \subset \mathsf{supp}(q_\mathsf{n})$. If there exists $\theta^\star$ such that $\phi_{\theta^\star}(\cdot) = q_\mathsf{d}(\cdot)$, then $\phi_{\theta_f^{\mathsf{nce}}(q_\mathsf{d}, q_\mathsf{n})}(\cdot) = q_\mathsf{d}(\cdot)$.*

**Remark 1.1.** *Since the original family of unnormalized distributions $\{\phi_\theta(x) \colon \theta \in \Theta\}$ may not contain normalized distributions, we consider an augmented family $\phi_{\underline{\theta}}(x) \triangleq e^c \phi_\theta(x)$ for $\underline{\theta} \triangleq (\theta, c)$ for $c > 0$ for $f$-NCE. Then, we*

Table 1. Examples of the NCE objective. Recall that $\underline{\theta} \triangleq (\theta, \nu) \in \Theta \times \mathbb{R}$.

| Name | Generator function $f(\rho)$ | NCE objective $\mathcal{L}_f^{\mathrm{nce}}(\underline{\theta})$ |
|---|---|---|
| Log (Gutmann & Hyvärinen, 2012) | $f_{\log}(\rho) \triangleq \rho \log \rho - (\rho+1)\log(\rho+1)$ | $-\frac{1}{\nu}\mathbb{E}_{q_{\mathsf{d}}}[\log \frac{\rho_{\underline{\theta}}}{\rho_{\underline{\theta}}+1}] - \mathbb{E}_{q_{\mathsf{n}}}[\log \frac{1}{\rho_{\underline{\theta}}+1}]$ |
| Asymmetric power ($\alpha$) | $f_\alpha(\rho) \triangleq \frac{\rho^\alpha - 1}{\alpha(\alpha-1)}$ for $\alpha \notin \{0,1\}$ | $\frac{1}{1-\alpha}\mathbb{E}_{q_{\mathsf{d}}}[(\frac{q_{\mathsf{n}}}{\phi_\theta})^{1-\alpha}] + \frac{1}{\alpha}\mathbb{E}_{q_{\mathsf{n}}}[(\frac{\phi_\theta}{q_{\mathsf{n}}})^\alpha]$ |
| Asymmetric inverse log | $f_0(\rho) \triangleq \lim_{\alpha\downarrow0} f_\alpha(\rho) = -\log \rho$ | $\mathbb{E}_{q_{\mathsf{d}}}[\frac{q_{\mathsf{n}}}{\phi_\theta}] + \mathbb{E}_{q_{\mathsf{n}}}[\log \frac{\phi_\theta}{q_{\mathsf{n}}}]$ |
| Asymmetric log | $f_1(\rho) \triangleq \lim_{\alpha\uparrow1}(f_\alpha(\rho) + \frac{\rho-1}{\alpha-1}) = \rho \log \rho$ | $\mathbb{E}_{q_{\mathsf{d}}}[\log \frac{q_{\mathsf{n}}}{\phi_\theta}] + \mathbb{E}_{q_{\mathsf{n}}}[\frac{\phi_\theta}{q_{\mathsf{n}}}]$ |

*assume that $\{\phi_{\underline{\theta}}(x) : \underline{\theta} \in \Theta \times \mathbb{R}\}$ is well-specified, i.e., there exists $c^\star \in \mathbb{R}$ and $\theta^\star$ such that $q_{\mathsf{d}}(\cdot) = e^{c^\star}\phi_{\theta^\star}(\cdot)$. Hereafter, $\underline{\theta}$ denotes the augmented parameter, where $\theta$ without an underline denotes the original parameter.*

We consider the examples of $f$ in Table 1 as the canonical examples; each $f$ (or the corresponding $f$-NCE objective) is named based on its correspondence to a proper scoring rule (Gneiting & Raftery, 2007). It is easy to check that $\nu$ does not affect the objective function for the case of power scores $f_\alpha(\rho)$, and we thus set $\nu = 1$ in this case. We note that in the DRE literature, a similar objective based on the generator function $f_1(\rho)$ is known as Kullback–Leibler Importance Estimation Procedure (Sugiyama et al., 2008).

## 2. Two Variants of NCE

In this section, we introduce two variants of the $f$-NCE framework: $\alpha$-centered NCE and $f$-conditional NCE.

### 2.1. $\alpha$-Centered NCE

Consider the *asymmetric power* generator function $f_\alpha(\rho)$ for $\alpha \in \mathbb{R}$ (with $\nu = 1$); see the second row of Table 1. We will introduce a transformation called *$\alpha$-centering* in Eq. (3), which normalizes a given parametric model $\phi_\theta(x)$ in an $\alpha$- and $q_{\mathsf{n}}$-dependent manner. Applying the normalized model to $f_\alpha$-NCE (i.e., NCE induced by the asymmetric power score) results in a new variant of NCE. In Sec. 3.1, we show that this variant provides a unified view on several existing estimators, seemingly different at a first glance.

We define a *normalized* model of $\phi_\theta(x)$ called the *$\alpha$-centered model* as

$$\tilde{\phi}_{\theta;\alpha}(x) \triangleq \frac{\phi_\theta(x)}{Z_\alpha(\theta)}, \quad \text{where} \qquad (3)$$

$$Z_\alpha(\theta) \triangleq \begin{cases} \mathbb{E}_{q_{\mathsf{n}}(x)}[(\frac{\rho_\theta(x)}{q_{\mathsf{n}}(x)})^\alpha]^{1/\alpha} & \text{if } \alpha \neq 0, \\ \exp(\mathbb{E}_{q_{\mathsf{n}}(x)}[\log \frac{\rho_\theta(x)}{q_{\mathsf{n}}(x)}]) & \text{if } \alpha = 0. \end{cases}$$

Note that $Z_0(\theta) = \lim_{\alpha\downarrow0} Z_\alpha(\theta)$. Applying the $f_\alpha$-NCE objective to the $\alpha$-centered model, we define

$$\mathcal{L}_\alpha^{\mathrm{cent}}(\theta; q_{\mathsf{d}}, q_{\mathsf{n}}) \triangleq \mathcal{L}_{f_\alpha}^{\mathrm{nce}}(\tilde{\phi}_{\theta;\alpha}; q_{\mathsf{d}}, q_{\mathsf{n}})$$

$$\overset{(2)}{=} \frac{\mathbb{E}_{q_{\mathsf{d}}}[\tilde{\rho}_{\theta;\alpha}^{\alpha-1}(x)]}{1 - \alpha}$$

$$\overset{(3)}{=} \frac{\mathbb{E}_{q_{\mathsf{d}}}[\rho_\theta^{\alpha-1}(x)](\mathbb{E}_{q_{\mathsf{n}}}[\rho_\theta^\alpha(x)])^{\frac{1-\alpha}{\alpha}}}{1 - \alpha},$$

which we call the *$\alpha$-CentNCE* objective. Here, note that the second term in the $f_\alpha$-NCE objective becomes constant, since we design the $\alpha$-centered model such that $\mathbb{E}_{q_{\mathsf{n}}}[\tilde{\rho}_{\theta;\alpha}^\alpha(x)] = 1$. Note that the expectation with respect to the reference distribution $q_{\mathsf{n}}$ is embedded in the normalization term of the new model. In Table 2, we provide a side-by-side comparison between $f_\alpha$-NCE and $\alpha$-CentNCE objectives for $\alpha \in \{0, \frac{1}{2}, 1\}$.

We define the $\alpha$-CentNCE estimator as a minimizer of the objective function:

$$\theta_\alpha^{\mathrm{cent}}(q_{\mathsf{d}}, q_{\mathsf{n}}) \in \arg\min_{\theta \in \Theta} \mathcal{L}_\alpha^{\mathrm{cent}}(\phi_\theta; q_{\mathsf{d}}, q_{\mathsf{n}}).$$

In this case, since any multiplicative scaling to $\phi_\theta(x)$ is canceled out in the centered model in Eq. (3), the Fisher consistency follows even when the model is well-specified up to a constant, unlike the strict well-specifiedness required in Proposition 1.1.

**Proposition 2.1** ($\alpha$-CentNCE: Fisher consistency). *Let $\alpha \in \mathbb{R}$. Assume $\mathrm{supp}(q_{\mathsf{d}}) \subset \mathrm{supp}(q_{\mathsf{n}})$. If there exists $\theta^\star$ and $c > 0$ such that $c\phi_{\theta^\star}(\cdot) = q_{\mathsf{d}}(\cdot)$, then $\phi_{\theta_\alpha^{\mathrm{cent}}(q_{\mathsf{d}}, q_{\mathsf{n}})}(\cdot) \propto q_{\mathsf{d}}(\cdot)$.*

### 2.2. $f$-Conditional NCE

In the NCE literature, it is known that the noise distribution $q_{\mathsf{n}}$ must be carefully chosen to guarantee good convergence of the resulting estimator, generally considered hard in practice (Chehab et al., 2022). Alternatively, Ceylan & Gutmann (2018) proposed a new framework called the conditional NCE (CondNCE), where the idea is to draw noisy samples *conditioned* on the data samples. CondNCE was further justified via a connection to the *score matching* framework of Hyvärinen (2005). In this paper, we clarify the connection to score matching (in Sec. 3.2), and establish the first finite-sample convergence rate of this estimator (in Sec. 4).

Here, we introduce *$f$-CondNCE*, a general CondNCE framework for a convex function $f$. The idea is same as $f$-NCE:

*Table 2.* Special cases of the $f_\alpha$-NCE and $\alpha$-CentNCE objectives. The view on the estimators highlighted in blue and boldface via $\alpha$-CentNCE are new; see Theorem 3.2.

| Objectives | $\alpha = 0$ | $\alpha = \frac{1}{2}$ | $\alpha = 1$ |
|---|---|---|---|
| $f_\alpha$-NCE | $\mathbb{E}_{q_d}[\frac{q_n}{\phi_\theta}] + \mathbb{E}_{q_n}[\log \frac{\phi_\theta}{q_n}]$ (InvIS (Pihlaja et al., 2010)) | $2(\mathbb{E}_{q_d}[\sqrt{\frac{q_n}{\phi_\theta}}] + \mathbb{E}_{q_n}[\sqrt{\frac{\phi_\theta}{q_n}}])$ (eNCE (Liu et al., 2021)) | $\mathbb{E}_{q_d}[\log \frac{q_n}{\phi_\theta}] + \mathbb{E}_{q_n}[\frac{\phi_\theta}{q_n}]$ (Importance Sampling (IS) (Pihlaja et al., 2010; Riou-Durand & Chopin, 2018)) |
| $\alpha$-CentNCE | $\mathbb{E}_{q_d}[\frac{q_n}{\phi_\theta}]e^{\mathbb{E}_{q_n}[\log \frac{\phi_\theta}{q_n}]}$ **(GlobalGISO** (Shah et al., 2023)) | $2\mathbb{E}_{q_d}[\sqrt{\frac{q_n}{\phi_\theta}}]\mathbb{E}_{q_n}[\sqrt{\frac{\phi_\theta}{q_n}}]$ | $\mathbb{E}_{q_d}[\log \frac{q_n}{\phi_\theta}] + \log \mathbb{E}_{q_n}[\frac{\phi_\theta}{q_n}]$ **(MLE** (Fisher, 1922), **MC-MLE** (Geyer, 1994; Jiang et al., 2023)) |

we aim to minimize the Bregman divergence between two density ratios with respect to $f$. In this case, instead of the noise distribution $q_n$, we consider a channel (conditional distribution) $\pi(y|x)$, and aim to contrast the joint distributions $q_d(x)\pi(y|x)$ vs. $q_d(y)\pi(x|y)$. Comparing to $q_d(x)$ vs. $q_n(x)$ in the standard NCE, the contrast is *self-referential* in the sense that the data distribution $q_d$ appears on the both sides. Let $\rho_\theta(x,y) \triangleq \frac{\phi_\theta(x)\pi(y|x)}{\phi_\theta(y)\pi(x|y)}$ be the model density ratio in this case, implicitly assuming $\nu = 1$. We define the generalized conditional NCE objective $\mathcal{L}_f^{\mathsf{cond}}(\phi_\theta; q_d, \pi)$ as in Eq. (4), where the last equality follows from $\rho_\theta(y,x) = \rho_\theta(x,y)^{-1}$. For further simplicity, we focus on *symmetric* channels, i.e., $\pi(y|x) = \pi(x|y)$, in which case the ratio simplifies to $\rho_\theta(x,y) = \frac{\phi_\theta(x)}{\phi_\theta(y)}$. For $\mathrm{supp}(q_d) = \mathcal{X} = \mathbb{R}^d$, canonical examples are $(i)$ a Gaussian noise $\pi(y|x) = \mathcal{N}(y; x, \sigma^2 I)$ and $(ii)$ a uniform noise over a $\ell_s$-norm ball or sphere for some $s \geq 1$. We define the $f$-CondNCE estimator as a minimizer of the objective:

$$\theta_f^{\mathsf{cond}}(q_d, \pi) \in \arg\min_{\theta \in \Theta} \mathcal{L}_f^{\mathsf{cond}}(\phi_\theta; q_d, \pi).$$

Similar to $\alpha$-CentNCE, the Fisher consistency follows even when the model is well-specified up to a constant as any multiplicative scaling to $\phi_\theta(x)$ is cancelled out.

**Proposition 2.2** ($f$-CondNCE: Fisher consistency). *Let $f$ be a strictly convex function. Let $\pi(y|x)$ be a conditional distribution such that $\mathrm{supp}(q_d(x)\pi(y|x)) = \mathrm{supp}(q_d(y)\pi(x|y))$. If there exists a unique $\theta^\star$ and $c > 0$ such that $c\phi_{\theta^\star}(\cdot) = q_d(\cdot)$, then $\phi_{\theta_f^{\mathsf{cond}}(q_d,\pi)}(\cdot) \propto q_d(\cdot)$.*

In practice, given $n_d$ samples $\{(x_i)\}_{i=1}^{n_d}$ drawn i.i.d. from $q_d(x)$ and conditional samples $\{y_{ij}\}_{j=1}^{K}$ conditionally independent from $\pi(y|x_i)$ for each $i$, we let $\mathcal{L}_f^{\mathsf{cond}}(\phi_\theta; \hat{q}_d, \hat{\pi})$ denote the corresponding empirical objective with a slight abuse of notation.

## 3. Connecting the Dots

In this section, we explain how the estimators introduced in the previous section unify and generalize the existing estimators and provide new theoretical insights.

### 3.1. MLE, MC-MLE, and GlobalGISO as Limiting Instances of Centered NCE

As alluded to above, $\alpha$-CentNCE estimators *interpolate* between MLE (Fisher, 1922) ($\alpha = 1$) and GlobalGISO (Shah et al., 2023) ($\alpha = 0$, specifically for exponential family), provided that $Z_\alpha(\theta)$ can be computed analytically, i.e., without estimation. In the case of estimating $Z_\alpha(\theta)$ with samples, $\alpha$-CentNCE objective recovers MC-MLE (Geyer, 1994) when $\alpha = 1$. We formally summarize the connections in the next statement and Table 2.

**Theorem 3.1** ($\alpha$-CentNCE subsumes MLE and Global-GISO). *The following holds:*

1. *($\alpha = 0$: GlobalGISO) For an exponential family $\phi_\theta(x)$, if $\mathcal{X}$ is bounded and $q_n(x)$ is a uniform distribution over $\mathcal{X}$, the 0-CentNCE objective $\tilde{\mathcal{L}}_0(\theta; q_d, q_n)$ is equivalent to GlobalGISO (Shah et al., 2021b).*

2. *($\alpha = 1$: MLE) If $Z_1(\theta)$ is assumed to be computable for each $\theta$, the 1-CentNCE objective $\tilde{\mathcal{L}}_1(\theta; \hat{q}_d, q_n)$ is equivalent to MLE (Fisher, 1922).*

3. *($\alpha = 1$: MC-MLE) If $Z_1(\theta) = \mathbb{E}_{q_n}[\frac{\phi_\theta(x)}{q_n(x)}]$ is estimated with empirical noise distribution $\hat{q}_n(x)$, the 1-CentNCE objective $\tilde{\mathcal{L}}_1(\theta; \hat{q}_d, \hat{q}_n)$ is equivalent to MC-MLE (Geyer, 1994).*

**Remark 3.1.** *Note that the connection between GlobalGISO and MLE can be made for the case when $Z_\alpha(\theta)$ is assumed to be computable for any $\theta$. At one extreme when $\alpha = 1$, in which case the objective boils down to that of MLE, it is clear that $Z_1(\theta) = \mathbb{E}_{q_n}[\frac{\phi_\theta(x)}{q_n(x)}] = \int \phi_\theta(x)dx$ becomes the standard partition function. In the other extreme case where $\alpha \to 0$, if $\phi_\theta(x) = \exp(\langle \theta, \psi(x) \rangle)$ is an exponential family, computing $Z_0(\theta)$ boils down to computing $\mathbb{E}_{q_n(x)}[\psi(x)]$ since $Z_0(\theta) \propto \exp(\langle \theta, \mathbb{E}_{q_n}[\psi] \rangle)$. For a special choice of $q_n$ (e.g., uniform distribution) and $\psi$ (e.g., polynomial and sinusoidal functions), this term can be computed analytically, as concretely illustrated by (Shah et al., 2023). We also provide an alternative theoretical view of the 0-CentNCE objective as a certain KL divergence minimization problem,*

$$\mathcal{L}_f^{\mathsf{cond}}(\phi_\theta; q_{\mathsf{d}}, \pi) \triangleq \mathbb{E}_{q_{\mathsf{d}}(y)\pi(x|y)}\left[\Delta_f\left(\frac{q_{\mathsf{d}}(x)\pi(y|x)}{q_{\mathsf{d}}(y)\pi(x|y)}, \frac{\phi_\theta(x)\pi(y|x)}{\phi_\theta(y)\pi(x|y)}\right)\right] - \mathbb{E}_{q_{\mathsf{d}}(x)\pi(y|x)}\left[f\left(\frac{q_{\mathsf{d}}(x)\pi(y|x)}{q_{\mathsf{d}}(y)\pi(x|y)}\right)\right]$$

$$= \mathbb{E}_{q_{\mathsf{d}}(x)\pi(y|x)}\left[-f'(\rho_\theta(x,y)) + \rho_\theta(y,x)f'(\rho_\theta(y,x)) - f(\rho_\theta(y,x))\right]. \tag{4}$$

*generalizing the justification for GlobalGISO given in (Shah et al., 2023); see Theorem B.1.*

Next, we provide a result connecting $f_\alpha$-NCE and $\alpha$-CentNCE estimators, under the assumption that we have an optimization oracle that finds the global minima of a given objective.

**Theorem 3.2** ($f_\alpha$-NCE and $\alpha$-CentNCE estimators are equivalent). *For a set $A \subset \Theta \times \mathbb{R}$ in the augmented parameter space, let $A|_\Theta \triangleq \{\theta \colon (\theta,\nu) \in A \text{ for some } \nu \in \mathbb{R}\}$ denote the subset corresponding to $\Theta$. Then,*

$$\left.\underset{\underline{\theta}=(\theta,\nu)\in\Theta\times\mathbb{R}}{\arg\min}\ \mathcal{L}_{f_\alpha}^{\mathsf{nce}}(\underline{\theta}; \hat{q}_{\mathsf{d}}, \hat{q}_{\mathsf{n}})\right|_\Theta = \underset{\theta\in\Theta}{\arg\min}\ \mathcal{L}_\alpha^{\mathsf{cent}}(\theta; \hat{q}_{\mathsf{d}}, \hat{q}_{\mathsf{n}}).$$

**Remark 3.2.** *We remark that, for $\alpha = 1$, Riou-Durand & Chopin (2018) proposed to convert the MC-MLE objective by the inverse of the 1-centering operation, which they call the* Poisson transform *(Barthelmé & Chopin, 2015), into the $f_1$-NCE objective, which they call the* importance sampling (IS) objective. *In this view, our $\alpha$-centering can be understood as the inverse of the* generalized Poisson transform. *Via the equivalence, Riou-Durand & Chopin (2018) analyzed the asymptotic property of MC-MLE by studying the $f_1$-NCE. Similarly, one can analyze the statistical property of GlobalGISO (with any valid choice of $q_{\mathsf{n}}$ beyond the uniform distribution) when $Z_0(\theta)$ is estimated with samples from $q_{\mathsf{n}}(x)$ via analyzing the $f_0$-NCE objective.*

### 3.2. Revisiting the Connection Between CondNCE and Score Matching

Ceylan & Gutmann (2018) argued that for a continuous domain $\mathcal{X}$, the original CondNCE objective is related to the score matching objective of Hyvärinen (2005), justifying the consistency of CondNCE. Here, we demonstrate that this interpretation can be misleading in a realistic setting with finite samples. To revisit this connection, we further restrict the type of channels to $\pi_\epsilon(y|x)$ parameterized by a parameter $\epsilon > 0$, such that $y \sim \pi_\epsilon(y|x)$ is equivalent to $y = x + \epsilon v$ for some $v \sim q_{\mathsf{s}}(\cdot)$ with zero mean and identity covariance, i.e., $\mathbb{E}_{q_{\mathsf{s}}}[v] = 0$ and $\mathbb{E}_{q_{\mathsf{s}}}[vv^\intercal] = I_d$. With this simplification, we denote the objective function as

$$\mathcal{L}_f^{\mathsf{cond}}(\phi_\theta; q_{\mathsf{d}}, q_{\mathsf{s}}; \epsilon)$$
$$\triangleq \mathbb{E}_{q_{\mathsf{d}}(x)q_{\mathsf{s}}(v)}\big[-f'(\rho_\theta(x,y))$$
$$+ \rho_\theta(y,x)f'(\rho_\theta(y,x)) - f(\rho_\theta(y,x))\big],$$

where $y \triangleq x + \epsilon v$. Then, we show that the $f$-CondNCE objective behaves as the score matching objective (Hyvärinen, 2005) in the limit of $\epsilon \to 0$. Formally:

**Theorem 3.3** (Asymptotic behavior of population $f$-CondNCE for small $\epsilon$). *The population $f$-CondNCE objective can be written as*

$$\mathcal{L}_f^{\mathsf{cond}}(\phi_\theta; q_{\mathsf{d}}, q_{\mathsf{s}}; \epsilon) = -f(1) + f''(1)\mathcal{L}^{\mathsf{sm}}(\phi_\theta; q_{\mathsf{d}})\epsilon^2 + o(\epsilon^2),$$

*where*

$$\mathcal{L}^{\mathsf{sm}}(\phi_\theta; q_{\mathsf{d}}) \triangleq \mathbb{E}_{q_{\mathsf{d}}(x)}\left[\mathrm{tr}(\nabla_x^2 \log \phi_\theta(x)) + \frac{1}{2}\|\nabla_x \log \phi_\theta(x)\|^2\right]$$

*denotes the (population) score matching (SM) objective (Hyvärinen, 2005).*

This statement generalizes the result in (Ceylan & Gutmann, 2018) for $f_{\log}$-CondNCE to $f$-CondNCE for any $f$. Below, we explain why this statement may be misleading as the $f$-CondNCE estimator with $\epsilon \to 0$ does *not* behave like the SM estimator. To correctly understand the behavior, we need to consider the *empirical $f$-CondNCE objective* function that defines the empirical estimator, instead of population objective.

**Theorem 3.4** (Asymptotic behavior of empirical $f$-CondNCE for small $\epsilon$). *The empirical $f$-CondNCE objective can be written as*

$$\mathcal{L}_f^{\mathsf{cond}}(\phi_\theta; \hat{q}_{\mathsf{d}}, \hat{q}_{\mathsf{s}}) = -f(1)$$
$$+ 2f''(1)\mathbb{E}_{\hat{q}_{\mathsf{d}}(x)\hat{q}_{\mathsf{s}}(v)}[\nabla_x \log \phi_\theta(x)^\intercal v]\epsilon$$
$$+ f''(1)\mathcal{L}^{\mathsf{ssm}}(\phi_\theta; \hat{q}_{\mathsf{d}}, \hat{q}_{\mathsf{s}})\epsilon^2 + o(\epsilon^2).$$

*Here, we define the empirical sliced SM (SSM) objective (Song et al., 2020)*

$$\mathcal{L}^{\mathsf{ssm}}(\phi_\theta; \hat{q}_{\mathsf{d}}, \hat{q}_{\mathsf{s}})$$
$$\triangleq \mathbb{E}_{\hat{q}_{\mathsf{d}}(x)\hat{q}_{\mathsf{s}}(v)}\left[v^\intercal \nabla_x^2 \log \phi_\theta(x)v + \frac{1}{2}(v^\intercal \nabla_x \log \phi_\theta(x))^2\right].$$

**Remark 3.3.** *Since we assume that $q_{\mathsf{s}}(v)$ has zero mean, Theorem 3.3 readily follows as a corollary of Theorem 3.4, as the $O(\epsilon)$ term will converge to 0 in the population limit of $q_{\mathsf{s}}$. In a finite-sample regime, however, the dominating term of the $f$-CondNCE objective becomes the $O(\epsilon)$ term, i.e., as $\epsilon \to 0$, we have*

$$\frac{1}{\epsilon}\frac{\hat{\mathcal{L}}_f^{\mathsf{cond}}(\phi_\theta; \hat{q}_{\mathsf{d}}, \hat{q}_{\mathsf{s}}) + f(1)}{2f''(1)} \to \mathbb{E}_{\hat{q}_{\mathsf{d}}(x)}[\nabla_x \log \phi_\theta(x)]^\intercal \mathbb{E}_{\hat{q}_{\mathsf{s}}(v)}[v].$$

*Thus, the $f$-CondNCE objective is dominated by this statistical noise term when $\epsilon \ll 1$ with fixed sample size of $v \sim q_s$, and thus too small $\epsilon$ should be avoided in stark contrast to the proposed justification in (Ceylan & Gutmann, 2018). We revisit this degrading behavior after the finite-sample guarantee of $f$-CondNCE in Remark 4.3.*

*It is worth noting, however, that $\mathbb{E}_{\hat{q}_s}[v]$ gets more concentrated around 0 as the number of slicing vectors increases. Therefore, one could consider a carefully chosen $\epsilon$ as a function of the number of slicing vectors and distribution-dependent quantities, so that $\frac{1}{\epsilon}\mathbb{E}_{\hat{q}_d(x)\hat{q}_s(v)}[\nabla_x \log \phi_\theta(x)^\intercal v]$ still vanishes as the number of slicing vectors increases. In this way, the $f$-CondNCE estimator might be still consistent with small $\epsilon$, emulating the behavior of SSM.*

**Simulation.** *To demonstrate this behavior, we considered a simple synthetic setup, where the data generating distribution is $\mathcal{N}(\mu, 1)$ with $\mu = 1$. With a conditional noise distribution $\pi(y|x) = \mathcal{N}(y|x, \epsilon^2 I)$ with varying $\epsilon$, we plot the derivatives of the empirical objective of the original CNCE with varying $K \in \{1, 4, 16, 64\}$, where the sample size is $N = 10^4$. As shown in Figure 1, the empirical derivatives characterize the mean fairly closely when $\epsilon \geq 10^{-2}$ or when $\epsilon$ is small and $K$ is large. This simple 1D Gaussian example clearly shows the undesirable behavior of the CNCE objective when $\epsilon$ is small. More in-depth study on the effect of $\epsilon$ and $K$ for high-dimensional problems is left as a future work.*

## 4. Finite-Sample Analysis

In this section, we provide finite-sample guarantees of regularized versions of the aforementioned NCE estimators, specifically assuming an exponential family distribution model $\phi_\theta(x) = \exp(\langle \theta, \psi(x) \rangle)$. Here, $\theta \in \mathbb{R}^p$ denotes the natural parameter, $\psi : \mathcal{X} \to \mathbb{R}^p$ denotes the natural statistics, and $p$ denotes the number of parameters. In what follows, we assume both well-specifiedness and identifiability, i.e., there exists a *unique* $\theta^\star \in \Theta$ such that $\phi_{\theta^\star}(\cdot) \propto q_d(\cdot)$.

Below, we establish the parametric error rate $O(n^{-1/2})$ of convergence for the regularized NCE estimators. The proofs adapt the analysis in (Shah et al., 2023) for GlobalGISO, which in turn built upon (Negahban et al., 2012; Vuffray et al., 2016; 2021; Shah et al., 2021b). We note in passing that the non-regularized NCE estimators can also be analyzed, but we can only prove a suboptimal rate of $O(n^{-1/4})$ by following the existing analysis in (Shah et al., 2021b).

Following (Shah et al., 2023), we are specifically interested in the case where the statistics are bounded and so is the parameter space. We note that the bounded statistics may not be too restrictive, as in many practical scenarios the domain

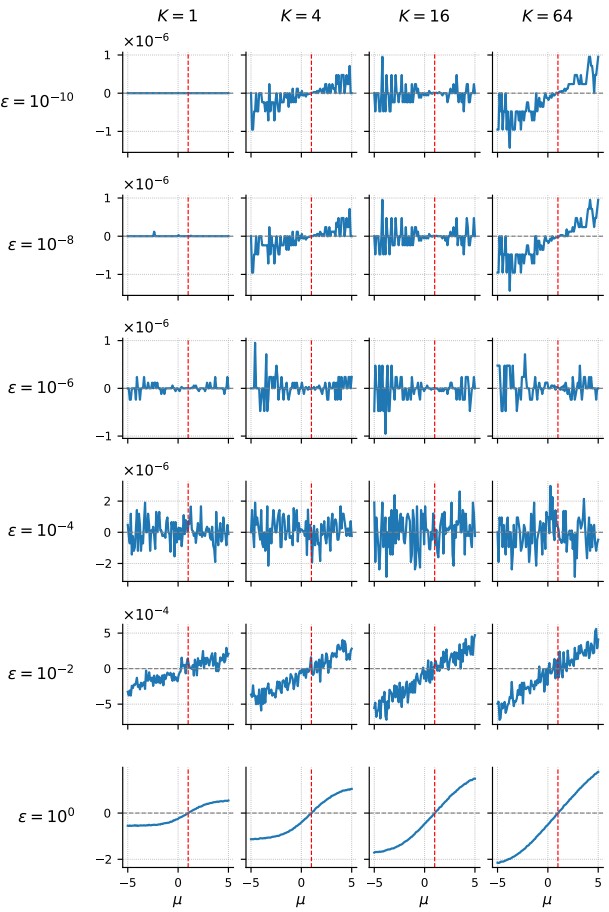

*Figure 1.* Derivatives of the empirical CondNCE objective with varying $\epsilon \in \{10^{-10}, \ldots, 10^0\}$ and $K \in \{1, 4, 16, 64\}$ for 1D Gaussian data with true mean $\mu = 1.0$ (vertical dashed red lines) and a conditional noise distribution $\pi(y|x) = \mathcal{N}(y|x, \epsilon^2 I)$.

$\mathcal{X}$ may naturally be truncated during data acquisition (Liu et al., 2022).

**Assumption 4.1** (Bounded maximum norm of $\psi$). $\sup_{x \in \mathcal{X}} \|\psi(x)\|_\infty \leq \psi_{\max}$ *for some* $\psi_{\max} > 0$.

**Assumption 4.2** (Bounded parameter space). *For some constant $r > 0$, $\sup_{\theta \in \Theta} \mathcal{R}(\theta) \leq r$.*

We note that the gradient and Hessian of the $f$-NCE objective can be written as

$$\nabla \hat{\mathcal{L}}_f^{\text{nce}}(\theta) = \frac{1}{\nu}\mathbb{E}_{\hat{q}_d}[\psi \xi_{\text{nce},f,\text{d}}^{(1)}(\rho_\theta)] + \mathbb{E}_{\hat{q}_n}[\psi \xi_{\text{nce},f,\text{n}}^{(1)}(\rho_\theta)],$$

$$\nabla^2 \hat{\mathcal{L}}_f^{\text{nce}}(\theta) = \frac{1}{\nu}\mathbb{E}_{\hat{q}_d}[\psi \psi^\intercal \xi_{\text{nce},f,\text{d}}^{(2)}(\rho_\theta)] + \mathbb{E}_{\hat{q}_n}[\psi \psi^\intercal \xi_{\text{nce},f,\text{n}}^{(2)}(\rho_\theta)],$$

where the functions $\xi_{\text{nce},f,\text{r}}^{(i)}(\rho)$ for $i \in \{1, 2\}$ and $\text{r} \in \{\text{d}, \text{n}\}$ are defined in the leftmost column of Table 3; see Lemma B.3.

Our analysis relies on the boundedness of the model density ratio $\rho_\theta \in (\rho_{\min}, \rho_{\max})$. In each result, we clarify the defi-

nition of the worst-case density ratios $(\rho_{\min}, \rho_{\max})$. These ratios affect the convergence rate through the following quantities:

$$
\begin{aligned}
b^{(2)}_{\mathsf{nce},f,\mathsf{r}} &\triangleq \inf_{\rho \in (\rho_{\min}, \rho_{\max})} |\xi^{(2)}_{\mathsf{nce},f,\mathsf{r}}(\rho)| \quad \text{and} \\
B^{(i)}_{\mathsf{nce},f,\mathsf{r}} &\triangleq \sup_{\rho \in (\rho_{\min}, \rho_{\max})} |\xi^{(i)}_{\mathsf{nce},f,\mathsf{r}}(\rho)| \quad \text{for} \;\; i \in \{1, 2\},
\end{aligned}
\tag{5}
$$

where $\mathsf{r} \in \{\mathsf{d}, \mathsf{n}\}$. We remark that these quantities differ for each estimator. For the canonical choices of $f(\rho)$, i.e., log and asymmetric power, these quantities are explicitly given in Table 3.

Let $\mathcal{R} \colon \Theta \to \mathbb{R}_{\geq 0}$ be a norm over $\Theta$, and $\mathcal{R}^* \colon \Theta \to \mathbb{R}_{\geq 0}$ be its dual norm. Define

$$
\gamma_{1;2} \triangleq \sup_{\theta \in 4\Theta \setminus \{0\}} \frac{\|\theta\|_1}{\|\theta\|_2},
\tag{6}
$$

$$
\gamma_{\mathcal{R}^*;\infty} \triangleq \sup_{\theta \in \mathbb{R}^k \setminus \{0\}} \frac{\mathcal{R}^*(\theta)}{\|\theta\|_{\max}},
\tag{7}
$$

$$
\gamma_{\mathcal{R};2} \triangleq \sup_{\theta \in \Theta \setminus \{0\}} \frac{\mathcal{R}(\theta)}{\|\theta\|_2}.
\tag{8}
$$

Here $4\Theta \triangleq \{4\theta \colon \theta \in \Theta\}$. These quantities capture the geometry of the norm $\mathcal{R}(\cdot)$ imposed on the parameter space $\Theta$, and appear in the convergence rates.

**Theorem 4.1** ($f$-NCE: finite-sample guarantee)**.** *Pick a strictly convex function $f \colon \mathbb{R}_+ \to \mathbb{R}$. Define*

$$
(\rho_{\min}, \rho_{\max}) \triangleq \left( \inf_{x \in \mathcal{X}, \underline{\theta} \in \Theta \times \mathbb{R}} \rho_{\underline{\theta}}(x), \sup_{x \in \mathcal{X}, \underline{\theta} \in \Theta \times \mathbb{R}} \rho_{\underline{\theta}}(x) \right)
$$

*and define the quantities in Eq. (5) accordingly. For $\mathsf{r} \in \{\mathsf{d}, \mathsf{n}\}$, define*

$$
\lambda^{\mathsf{nce}}_{\min,\mathsf{r}} \triangleq \lambda_{\min}(\mathbb{E}_{q_{\mathsf{r}}}[\psi\psi^{\mathsf{T}}]).
$$

*Let $\hat{\theta}^{\mathsf{nce},\mathcal{R}}_{f,n_{\mathsf{d}},n_{\mathsf{n}}}$ be such that*

$$
\hat{\theta}^{\mathsf{nce},\mathcal{R}}_{f,n_{\mathsf{d}},n_{\mathsf{n}}} \in \arg\min_{\theta \in \Theta} \left\{ \mathcal{L}^{\mathsf{nce}}_f(\theta; \hat{q}_{\mathsf{d}}, \hat{q}_{\mathsf{n}}) + \lambda_{n_{\mathsf{d}},n_{\mathsf{n}}} \mathcal{R}(\theta) \right\}
$$

*for some $\lambda_{n_{\mathsf{d}},n_{\mathsf{n}}} > 0$. Then, for any $\Delta > 0$ and $\delta \in (0,1)$, there exists a choice of $\lambda_{n_{\mathsf{d}},n_{\mathsf{n}}}$ such that $\|\hat{\theta}^{\mathsf{nce},\mathcal{R}}_{f,n_{\mathsf{d}},n_{\mathsf{n}}} - \theta^\star\|_2 \leq \Delta$ with probability $\geq 1 - \delta$, provided that for each $\mathsf{r} \in \{\mathsf{d}, \mathsf{n}\}$,*

$$
n_{\mathsf{r}} = \Omega\left( \max\left\{ \frac{(B^{(1)}_{\mathsf{nce},f,\mathsf{r}})^2 \gamma^2_{\mathcal{R};2} \gamma^2_{\mathcal{R}^*;\infty} \psi^2_{\max}}{\Delta^2(\nu^{-1} b^{(2)}_{\mathsf{nce},f,\mathsf{d}} \lambda^{\mathsf{nce}}_{\min,\mathsf{d}} + b^{(2)}_{\mathsf{nce},f,\mathsf{n}} \lambda^{\mathsf{nce}}_{\min,\mathsf{n}})^2}, \right.\right.
$$
$$
\left.\left. \frac{\gamma^4_{1;2} \psi^4_{\max}}{(\lambda^{\mathsf{nce}}_{\min,\mathsf{r}})^2} \right\} \log\frac{p^2}{\delta} \right).
$$

**Remark 4.1.** *To the best of our knowledge, this result is the first finite-sample convergence rate for $f$-NCE estimators.*

*We state the finite-sample statement with a minimal set of assumptions, along with the bounded statistics and parameter space assumptions. While achieving the parametric rate of convergence $O(n^{-1/2})$ is appealing, to have non-vacuous rates, however, we need all the quantities in the sample complexity expression to be within a range bounded away from $0$ or $\infty$. More concretely, if we further assume that the dual norm of the statistic $\sup_{x \in \mathcal{X}} \mathcal{R}^*(\psi(x)) \leq \tau$ is bounded for some constant $\tau > 0$, it is easy to check that the worst-case density ratios are bounded as $(\rho_{\min}, \rho_{\max}) \subset (e^{-r\tau}, e^{r\tau})$ for $f$-NCE, where $r$ is defined to be the diameter of $\Theta$ measured in the norm $\mathcal{R}(\cdot)$; see Assumption 4.2. We note that the worst-case density ratios affect the quantities in Eq. (5) polynomially for the canonical examples in Table 3, which in turn affect the sample complexity polynomially. Hence, the leading constant grows exponentially in $r$ and $d$ similar to (Shah et al., 2021b; 2023). This remark remains valid for the following two statements for $\alpha$-CentNCE and $f$-CondNCE, as the worst-case density ratio bounds depend similarly on $r$ and $\tau$. We also remark that the minimum eigenvalue conditions are typically assumed in the existing finite-sample analysis (Vuffray et al., 2016; Shah et al., 2021b; 2023), while (Shah et al., 2021a) establishes an explicit lower bound on the minimum eigenvalue for node-wise-sparse Gaussian MRFs.*

**Theorem 4.2** ($\alpha$-CentNCE: finite-sample guarantee)**.** *Pick $\alpha \in \mathbb{R}$. Define*

$$
(\rho_{\min}, \rho_{\max}) \triangleq \left( \inf_{x \in \mathcal{X}, \theta \in \Theta} \tilde{\rho}_{\theta;\alpha}(x), \sup_{x \in \mathcal{X}, \theta \in \Theta} \tilde{\rho}_{\theta;\alpha}(x) \right)
$$

*and define the quantities in Eq. (5) for $f = f_\alpha$ accordingly. Let $\tilde{\rho}^\alpha_{\theta^\star;\alpha}(x) \triangleq \frac{(\frac{q_{\mathsf{d}}(x)}{q_{\mathsf{n}}(x)})^\alpha}{\mathbb{E}_{q_{\mathsf{n}}}[(\frac{q_{\mathsf{d}}}{q_{\mathsf{n}}})^\alpha]}$, and let*

$$
\lambda^{\mathsf{cent}}_{\min,\mathsf{d}} \triangleq \lambda_{\min}(\mathbb{E}_{q_{\mathsf{d}}}[(\psi - \mathbb{E}_{q_{\mathsf{n}}}[\psi\tilde{\rho}^\alpha_{\theta^\star;\alpha}])(\psi - \mathbb{E}_{q_{\mathsf{n}}}[\psi\tilde{\rho}^\alpha_{\theta^\star;\alpha}])^{\mathsf{T}}]),
$$
$$
\lambda^{\mathsf{cent}}_{\min,\mathsf{n}} \triangleq \lambda_{\min}(\mathbb{E}_{q_{\mathsf{n}}}[\psi\psi^{\mathsf{T}}\tilde{\rho}^\alpha_{\theta^\star;\alpha}] - \mathbb{E}_{q_{\mathsf{n}}}[\psi\tilde{\rho}^\alpha_{\theta^\star;\alpha}]\mathbb{E}_{q_{\mathsf{n}}}[\psi\tilde{\rho}^\alpha_{\theta^\star;\alpha}]^{\mathsf{T}}).
$$

*Let $\hat{\theta}^{\mathsf{cent},\mathcal{R}}_{\alpha,n_{\mathsf{d}}}$ be such that*

$$
\hat{\theta}^{\mathsf{cent},\mathcal{R}}_{\alpha,n_{\mathsf{d}}} \in \arg\min_{\theta \in \Theta} \left\{ \mathcal{L}^{\mathsf{cent}}_\alpha(\theta; \hat{q}_{\mathsf{d}}, q_{\mathsf{n}}) + \lambda_{n_{\mathsf{d}}} \mathcal{R}(\theta) \right\}
$$

*for some $\lambda_{n_{\mathsf{d}}} > 0$. Define $\psi_{\max,\alpha} \triangleq \psi_{\max} + \|\mathbb{E}_{q_{\mathsf{n}}}[\psi\tilde{\rho}^\alpha_{\theta^\star;\alpha}]\|_{\max}$. Then, for any $\Delta > 0$ and $\delta \in (0,1)$, there exists a choice of $\lambda_{n_{\mathsf{d}}}$ such that $\|\hat{\theta}^{\mathsf{cent},\mathcal{R}}_{f,n_{\mathsf{d}}} - \theta^\star\|_2 \leq \Delta$ with probability $\geq 1 - \delta$, provided that*

$$
n_{\mathsf{d}} = \Omega\left( \max\left\{ \frac{(B^{(1)}_{\mathsf{nce},f_\alpha,\mathsf{d}})^2 \gamma^2_{\mathcal{R};2} \gamma^2_{\mathcal{R}^*;\infty} \psi^2_{\max,\alpha}}{\Delta^2(b^{(2)}_{\mathsf{nce},f_\alpha,\mathsf{d}})^2\{(1-\alpha)\lambda^{\mathsf{cent}}_{\min,\mathsf{d}} + \alpha\lambda^{\mathsf{cent}}_{\min,\mathsf{n}}\}^2}, \right.\right.
$$
$$
\left.\left. \frac{\gamma^4_{1;2} \psi^4_{\max,\alpha}}{(\lambda^{\mathsf{cent}}_{\min,\mathsf{d}})^2} \right\} \log\frac{p^2}{\delta} \right).
$$

*Table 3.* Definitions of $\xi_{\text{nce},f,\text{r}}^{(i)}(\rho)$ for $i \in \{1,2\}$ and $\text{r} \in \{\text{d},\text{n}\}$ for example generator functions $f$.

| Definitions | Log | Asymmetric power |
|---|---|---|
| $f(\rho)$ | $f_{\log}(\rho)$ | $f_\alpha(\rho)$ |
| $\xi_{\text{nce},f,\text{d}}^{(1)}(\rho) \triangleq -\rho f''(\rho)$ | $-\frac{1}{\rho+1}$ | $-\rho^{\alpha-1}$ |
| $\xi_{\text{nce},f,\text{n}}^{(1)}(\rho) \triangleq \rho^2 f''(\rho)$ | $\frac{\rho}{\rho+1}$ | $\rho^\alpha$ |
| $\xi_{\text{nce},f,\text{d}}^{(2)}(\rho) \triangleq \rho g_f(\rho)$ | $\frac{\rho}{(\rho+1)^2}$ | $(1-\alpha)\rho^{\alpha-1}$ |
| $\xi_{\text{nce},f,\text{n}}^{(2)}(\rho) \triangleq \rho^2(f''(\rho) - g_f(\rho))$ | $\frac{\rho}{(\rho+1)^2}$ | $\alpha\rho^\alpha$ |
| $(B_{\text{nce},f,\text{d}}^{(1)}, B_{\text{nce},f,\text{n}}^{(1)})$ | $(1,1)$ | $(\rho_{\min}^{\alpha-1}, \rho_{\max}^\alpha)$ |
| $(B_{\text{nce},f,\text{d}}^{(2)}, B_{\text{nce},f,\text{n}}^{(2)})$ | $(1,1)$ | $(|1-\alpha|\rho_{\min}^{\alpha-1}, |\alpha|\rho_{\max}^\alpha)$ |
| $(b_{\text{nce},f,\text{d}}^{(2)}, b_{\text{nce},f,\text{n}}^{(2)})$ | $(\kappa,\kappa)$, where $\kappa \triangleq \frac{\rho_{\min}}{(\rho_{\min}+1)^2} \wedge \frac{\rho_{\max}}{(\rho_{\max}+1)^2}$ | $(|1-\alpha|\rho_{\max}^{\alpha-1}, |\alpha|\rho_{\min}^\alpha)$ |

**Remark 4.2** (Special cases). *For $\alpha = 0$, this result generalizes the finite-sample analysis of GlobalGISO of (Shah et al., 2023) beyond when $q_\text{n}$ is the uniform distribution. For $\alpha = 1$, we establish the convergence rate of the MLE, which we believe to be the first result of this kind.*

For the CondNCE estimator, we consider $K = 1$, i.e., we have $\{(x_i, y_i)\}_{i=1}^{n_\text{d}} \sim q_\text{d}(x)\pi(y|x)$ for simplicity.

**Theorem 4.3** (*$f$-CondNCE: finite-sample guarantee*). *Pick a strictly convex function $f : \mathbb{R}_+ \to \mathbb{R}$. Define*

$$\rho_{\min} \triangleq \inf_{(x,y)\in\text{supp}(q_\text{d}(x)\pi(y|x)),\theta\in\Theta} \rho_\theta(x,y),$$
$$\rho_{\max} \triangleq \sup_{(x,y)\in\text{supp}(q_\text{d}(x)\pi(y|x)),\theta\in\Theta} \rho_\theta(x,y).$$

*and define the quantities in Eq. (5) accordingly. Let*

$$\lambda_{\min,\text{d}}^{\text{cond}} \triangleq \lambda_{\min}(\mathbb{E}_{q_\text{d}(x)\pi(y|x)}[(\psi(x)-\psi(y))(\psi(x)-\psi(y))^\intercal]).$$

*Let $\hat{\theta}_{f,n_\text{d}}^{\text{cond},\mathcal{R}}$ be such that*

$$\hat{\theta}_{f,n_\text{d}}^{\text{cond},\mathcal{R}} \in \arg\min_{\theta\in\Theta}\Big\{\mathcal{L}_f^{\text{cond}}(\theta; \hat{q}_\text{d}, \hat{\pi}) + \lambda_{n_\text{d}}\mathcal{R}(\theta)\Big\}$$

*for some $\lambda_{n_\text{d}} > 0$. Then, for any $\Delta > 0$ and $\delta \in (0,1)$, there exists a choice of $\lambda_n$ such that $\|\hat{\theta}_{f,n_\text{d}}^{\text{cond},\mathcal{R}} - \theta^\star\|_2 \leq \Delta$ with probability $\geq 1 - \delta$, provided that*

$$n_\text{d} = \Omega\Bigg(\max\Bigg\{\frac{(B_{\text{cond},f,\text{d}}^{(1)} + B_{\text{cond},f,\text{n}}^{(1)})^2\gamma_{\mathcal{R};2}^2\gamma_{\mathcal{R}^*;\infty}^2\psi_{\max}^2}{\Delta^2(b_{\text{cond},f,\text{d}}^{(2)} + b_{\text{cond},f,\text{n}}^{(2)})^2(\lambda_{\min}^{\text{cond}})^2},$$
$$\frac{\gamma_{1;2}^4\psi_{\max}^4}{(\lambda_{\min}^{\text{cond}})^2}\Bigg\}\log\frac{p^2}{\delta}\Bigg).$$

*Here, $b_{\text{cond},f,\text{r}}^{(2)}$ and $B_{\text{cond},f,\text{r}}^{(i)}$ are defined similar to Eq. (5), where the infimum and supremum are taken over $(\frac{\rho_{\min}}{\rho_{\max}}, \frac{\rho_{\max}}{\rho_{\min}})$ in place of $(\rho_{\min}, \rho_{\max})$.*

**Remark 4.3** (Behavior of *$f$-CondNCE in a small-$\epsilon$ regime*). *As alluded to in Sec. 3.2, the undesirable behavior of $f$-CondNCE with small $\epsilon$ can be also seen from the sample complexity, since the minimum eigenvalue $\lambda_{\min,\text{d}}^{\text{cond}} \approx$*

$\epsilon^2\lambda_{\min}(\mathbb{E}_{q_\text{d}(x)}[\nabla_x\psi(x)\nabla_x\psi(x)^\intercal]) \to 0$ *as $\epsilon \to 0$. In Theorem C.3 in Appendix, we establish that the asymptotic covariance of the estimator is $\check{\mathcal{V}}_f^{\text{cond}} \triangleq \check{\mathcal{I}}_f^{-1}\check{\mathcal{C}}_f\check{\mathcal{I}}_f^{-1}$, where*

$$\check{\mathcal{I}}_f \triangleq \mathbb{E}_{q_{\text{d},\pi}(x,y)}[\rho_{\theta^\star}^2 f''(\rho_{\theta^\star})(\psi(x)-\psi(y))(\psi(x)-\psi(y))^\intercal],$$
$$\check{\mathcal{C}}_f \triangleq \mathbb{E}_{q_{\text{d},\pi}(x,y)}[\xi_{\text{cond},f}^{(1)}(\rho_{\theta^\star})^2(\psi(x)-\psi(y))(\psi(x)-\psi(y))^\intercal],$$

*where we let $q_{\text{d},\pi}(x,y) \triangleq q_\text{d}(x)\pi(y|x)$. For a channel $y \sim \pi(y|x)$ defined as $y = x + \epsilon v$ as in Sec. 3.2, it is easy to check that $\lim_{\epsilon\to 0}\frac{1}{\epsilon^2}\check{\mathcal{I}}_f = \mathbb{E}_{q_\text{d}(x)}[\nabla_x\psi(x)\nabla_x\psi(x)^\intercal] = \lim_{\epsilon\to 0}\frac{1}{\epsilon^2}\check{\mathcal{C}}_f$. Hence, in the small-$\epsilon$ regime, the asymptotic covariance of the $f$-CondNCE also behaves as $\check{\mathcal{V}}_f^{\text{cond}} \approx \frac{1}{\epsilon^2}\mathbb{E}_{q_\text{d}(x)}[\nabla_x\psi(x)\nabla_x\psi(x)^\intercal]$, and hence blows up as $\epsilon \to 0$. These observations are consistent to Theorem 3.4.*

**Proof Sketch.** Our finite-sample analysis of the regularized NCE estimators follows closely that of Shah et al. (2023), which relies on the seminal result of (Negahban et al., 2012) for regularized M-estimators:

**Theorem 4.4** (Negahban et al., 2012, Corollary 1). *Let $z_1, \ldots, z_N$ be i.i.d. samples drawn from a distribution $p(z)$. Let $h_\theta(z)$ be a convex and differentiable function parameterized by $\theta \in \Theta$. Let $\hat{\mathcal{L}}_n(\theta) \triangleq \frac{1}{n}\sum_{i=1}^n h_\theta(z_i)$ denote the empirical objective function. Define*

$$\hat{\theta}_n \in \arg\min_\theta\Big\{\hat{\mathcal{L}}_n(\theta) + \lambda_n\mathcal{R}(\theta)\Big\}, \tag{9}$$

*where $\lambda_n$ is a regularization penalty and $\mathcal{R}: \Theta \to \mathbb{R}_{\geq 0}$ is a norm over $\Theta$. Let $\theta^\star \in \arg\min_\theta \mathbb{E}_{p(z)}[h_\theta(z)]$. Assume that*

1. *The regularization penalty $\lambda_n$ satisfies $\lambda_n \geq 2\mathcal{R}^*(\nabla_\theta\hat{\mathcal{L}}_n(\theta^\star))$, where $\mathcal{R}^*: \Theta^* \to \mathbb{R}_{\geq 0}$ is a dual norm of $\mathcal{R}$ over the dual space $\Theta^*$;*

2. *The empirical objective $\theta \mapsto \hat{\mathcal{L}}_n(\theta)$ satisfies a restricted strong convexity condition at $\theta = \theta^\star$ with curvature $\kappa > 0$, i.e., $\Delta_{\hat{\mathcal{L}}_n(\theta)}(\theta, \theta^\star) \geq \kappa\|\theta - \theta^\star\|_2^2$.*

*Then, the estimator $\hat{\theta}_n$ in Eq. (9) satisfies*

$$\|\hat{\theta}_n - \theta^\star\|_2 \le 3\frac{\lambda_n}{\kappa}\gamma_{\mathcal{R};2}.$$

To ensure the first condition with $\lambda_n$ sufficiently small, we show that, with high probability, the gradient of the empirical objective is sufficiently small, using Hoeffding's inequality under Assumption 4.1. For the second condition, we show that the lowest eigenvalue of the Hessian of the empirical objective is lower bounded, again by Hoeffding's inequality invoking Assumption 4.1 and the positivity of the minimum eigenvalues of some second moment matrices. Combining the two high-probability events by a union bound completes the proof.

**Simulation.** We include a preliminary simulation result of some NCE estimators in Appendix G. We leave a more thorough empirical investigation on the estimators in this paper for high-dimensional problems as a future work.

## 5. Discussion and Conclusion

**Beyond Bounded Exponential Families.** An intriguing question is whether we can relax the boundedness assumption on $\psi(x)$, making our estimators applicable beyond bounded (or truncated) exponential families. Here, we highlight what we need to modify in the proofs to extend the validity beyond this assumption, using $f$-NCE estimators as an example. As sketched above, the proof of Theorem 4.1 consists of two parts: (1) the concentration of the gradient of the empirical objective around 0, at the true parameter $\theta^\star$ (Proposition D.1) and (2) the restricted strong convexity (anti-concentration of the Hessian) of the empirical objective, around the true parameter $\theta^\star$ (Proposition D.2). Invoking the uniform bound via the worst-case density ratios, we apply Hoeffding's inequality using the boundedness of the max-norm of $\psi(x)$. For unbounded sufficient statistics, we need a technique to handle the concentration behaviors, without worst-case density ratios bounded away from 0 and $\infty$. For example, if the exponential family distribution is sub-Gaussian and the sufficient statistics are polynomials, one could use the sub-Weibull concentration bounds.

**Local Versions of NCE-based Estimators.** So far, we take a *global* approach to learning the parameter $\theta$ by treating it as a single object. In the context of exponential families, this is beneficial when exploiting a global structure on $\theta$ such as a bounded maximum norm, a bounded Frobenius norm, or a bounded nuclear norm when $\theta$ is matrix-shaped (Shah et al., 2023). However, for exponential families corresponding to a node-wise sparse Markov random fields (MRFs), the structure to be exploited is inherently *local*. Specifically, in node-wise-sparse MRFs, the conditional distribution of each node given all the other nodes can be expressed by number of parameters which scale with the maximum-degree of the MRF, which is assumed to be much smaller than the

dimension. In such scenarios, it is convenient to learn the conditional distribution for each node rather than learning the joint distribution over all nodes. There exists a long line of work on this approach, e.g., see (Besag, 1975; Vuffray et al., 2016; 2021; Shah et al., 2021a; Ren et al., 2021), a representative of which is the pseudo likelihood estimator of Besag (1975). Maybe not very surprisingly at this point, if we apply the NCE framework in a *local* manner, it provides a unifying view on all of the aforementioned works. We defer a detailed discussion to Appendix E.

**Optimization Complexity.** So far, we have focused on the statistical properties of the proposed estimators. Now, we make a few comments regarding the optimization complexity as concluding remarks. The first-order important property regarding optimization is the convexity of the objective functions with respect to the natural parameter $\theta$. In Appendix F.1, we characterize a sufficient condition for the convexity of $f$-NCE, $\alpha$-CentNCE, as well as $f$-CondNCE. Specifically, we show that $f_{\log}$ and $f_\alpha$ for $\alpha \in [0, 1]$ result in convex objectives. Somewhat surprisingly, a counterexample of convex $f$ which cannot guarantee convexity of the objective function is $f_\alpha(\rho)$ for $\alpha \notin [0, 1]$.

In the optimization community, a recent line of work (Liu et al., 2021; Lee et al., 2023) studied the optimization landscape of the original NCE objective and showed that the landscape can be arbitrarily flat even for a scalar Gaussian mean estimation. This is mainly due to the unbounded and light-tailed nature of Gaussian distributions. Under the boundedness assumption, we prove in Appendix F.2 that the empirical $f$-NCE objective function, for example, is smooth with probability 1. Then, from (Agarwal et al., 2010, Theorem 1), and the restricted strong convexity (Proposition D.2), a projected gradient descent algorithm has a globally geometric rate of convergence. A recent work (Jiang et al., 2023) analyzed the optimization landscape of MC-MLE and proposed an optimization algorithm with efficient optimization complexity guarantee together with a strong empirical result, missing the connection to the original work (Geyer, 1994) and its statistical properties analyzed in (Riou-Durand & Chopin, 2018). Building on top of our work and (Jiang et al., 2023) could be an exciting future direction at the intersection of statistical and optimization complexity for learning unnormalized distributions.

**Conclusion.** We hope that this work offers a unifying perspective on both existing estimators and those yet to be discovered, and that it contributes to a more systematic understanding of the trade-off between statistical and optimization complexity in the context of efficient learning with unnormalized distributions. As emphasized throughout the paper, further investigation is warranted to better understand the empirical behavior of different estimators in high-dimensional settings.

## Acknowledgements

We appreciate the insightful discussions with Devavrat Shah. This work was supported in part by the MIT-IBM Watson AI Lab under Agreement No. W1771646, and by AFRL and by the Department of the Air Force Artificial Intelligence Accelerator under Cooperative Agreement Number FA8750-19-2-1000. The views and conclusions contained in this document are those of the authors and should not be interpreted as representing the official policies, either expressed or implied, of the Department of the Air Force or the U.S. Government. The U.S. Government is authorized to reproduce and distribute reprints for Government purposes notwithstanding any copyright notation herein.

## Impact Statement

This paper presents work whose goal is to advance the field of Machine Learning. There are many potential societal consequences of our work, none which we feel must be specifically highlighted here.

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

# Appendix

## A. Glossary

For a reference, we provide a summary of notations in Table 4.

## B. Basic Properties

In what follows, we use Euler's notation and Lagrange's notation for derivatives. First, we remark the derivatives of the Bregman divergence with respect to the second argument:

$$\Delta_f(x,y) = f(x) - f(y) - f'(y)(x - y),$$
$$\partial_y \Delta_f(x,y) = (y - x)f''(y),$$
$$\partial_{yy} \Delta_f(x,y) = f''(y) + yf'''(y) - xf'''(y),$$
$$\partial_{yy} \Delta_f(x,y)|_{x=y} = f''(y).$$

Further, since we consider exponential family distributions, we have

$$\partial_{\theta_i} \rho_\theta = \rho_\theta \psi_i \quad \text{and} \quad \partial_{\theta_i \theta_j} \rho_\theta = \rho_\theta \psi_i \psi_j.$$

**Lemma B.1.** *For a three-times differentiable function $f$, let $g_f(\rho) = -(\rho f'''(\rho) + f''(\rho))$.*

$$\partial_{\theta_i \theta_j} \Delta_f(\rho^*, \rho_\theta) = \psi_i \psi_j \rho_\theta \{ (\rho_\theta f'''(\rho_\theta) + f''(\rho_\theta))(\rho_\theta - \rho^*) + \rho_\theta f''(\rho_\theta) \}$$
$$= \psi_i \psi_j \rho_\theta (\rho_\theta (f''(\rho_\theta) - g_f \rho_\theta)) + \rho^* g_f \rho_\theta)).$$

*Table 4.* Summary of notations.

| Notation | Definition | Description |
|---:|:---|:---|
| $\mathcal{X}$ | $\subset \mathbb{R}^d$ | domain of $x$ |
| $\Theta$ | $\subset \mathbb{R}^p$ | domain of $\theta$ |
| $\rho_\theta(x)$ | $\frac{\phi_\theta(x)}{\nu q_{\mathsf{n}}(x)}$ | (scaled) density ratio |
| $\Delta_h(z, z')$ | $h(z) - h(z') - \nabla_z h(z')^\intercal (z - z')$ | Bregman divergence of $h\colon \mathbb{R}^D \to \mathbb{R}$ |
| $\theta_f^{\mathsf{nce}}(q_{\mathsf{d}}, q_{\mathsf{n}})$ | $\in \arg\min_{\theta \in \Theta} \mathcal{L}_f^{\mathsf{nce}}(\phi_\theta; q_{\mathsf{d}}, q_{\mathsf{n}})$ | $f$-NCE estimator (population) |
| $\theta_\alpha^{\mathsf{cent}}(q_{\mathsf{d}}, q_{\mathsf{n}})$ | $\in \arg\min_{\theta \in \Theta} \mathcal{L}_\alpha^{\mathsf{cent}}(\phi_\theta; q_{\mathsf{d}}, q_{\mathsf{n}})$ | $\alpha$-CentNCE estimator (population) |
| $\theta_f^{\mathsf{cond}}(q_{\mathsf{d}}, \pi)$ | $\in \arg\min_{\theta \in \Theta} \mathcal{L}_f^{\mathsf{cond}}(\phi_\theta; q_{\mathsf{d}}, \pi)$ | $f$-CondNCE estimator (population) |
| $\theta_f^{\mathsf{nce}}(\hat{q}_{\mathsf{d}}, \hat{q}_{\mathsf{n}})$ | $\in \arg\min_{\theta \in \Theta} \mathcal{L}_f^{\mathsf{nce}}(\phi_\theta; \hat{q}_{\mathsf{d}}, \hat{q}_{\mathsf{n}})$ | $f$-NCE estimator (empirical) |
| $\theta_\alpha^{\mathsf{cent}}(\hat{q}_{\mathsf{d}}, q_{\mathsf{n}})$ | $\in \arg\min_{\theta \in \Theta} \mathcal{L}_\alpha^{\mathsf{cent}}(\phi_\theta; \hat{q}_{\mathsf{d}}, q_{\mathsf{n}})$ | $\alpha$-CentNCE estimator (empirical) |
| $\theta_f^{\mathsf{cond}}(\hat{q}_{\mathsf{d}}, \hat{\pi})$ | $\in \arg\min_{\theta \in \Theta} \mathcal{L}_f^{\mathsf{cond}}(\phi_\theta; \hat{q}_{\mathsf{d}}, \hat{\pi})$ | $f$-CondNCE estimator (empirical) |
| $\mathcal{R}(\cdot)$ | | a norm over $\Theta$ |
| $\mathcal{R}^*(\cdot)$ | | a dual norm over $\Theta^*$ |
| $\rho_{\min}$ | | minimum density ratio |
| $\rho_{\max}$ | | maximum density ratio |

## B.1. $f$-NCE

Recall

$$\hat{\mathcal{L}}_f^{\mathsf{nce}}(\theta) \triangleq \mathcal{L}_f^{\mathsf{nce}}(\phi_\theta; \hat{q}_{\mathsf{d}}, \hat{q}_{\mathsf{n}}) = -\frac{1}{\nu}\mathbb{E}_{\hat{q}_{\mathsf{d}}}[f'(\rho_\theta)] + \mathbb{E}_{\hat{q}_{\mathsf{n}}}[\rho_\theta f'(\rho_\theta) - f(\rho_\theta)].$$

### B.1.1. INVARIANCE

We define an equivalent class of generator functions $f$ that yield the same NCE objective. For a function $f_o$, let $\mathcal{F}^{\mathsf{nce}}(f_o) \triangleq \{f\colon \mathcal{L}_f^{\mathsf{nce}} \sim \mathcal{L}_{f_o}^{\mathsf{nce}}\}$, where the notation $\sim$ denotes that the two objective functions are equivalent up to constants, i.e., there exist $A, B \in \mathbb{R}$ such that $\mathcal{L}_f^{\mathsf{nce}}(\phi_\theta; q_{\mathsf{d}}, q_{\mathsf{n}}) \equiv A\mathcal{L}_{f_o}^{\mathsf{nce}}(\phi_\theta; q_{\mathsf{d}}, q_{\mathsf{n}}) + B$.

**Lemma B.2.** *If $f \in \mathcal{F}^{\mathsf{nce}}(f_o)$, $(\rho \mapsto af(\rho) + b\rho + c) \in \mathcal{F}^{\mathsf{nce}}(f_o)$ for any $a, b, c \in \mathbb{R}$.*

### B.1.2. DERIVATIVES

**Lemma B.3** (NCE: derivatives).

$$\nabla_\theta \hat{\mathcal{L}}_f^{\mathsf{nce}}(\theta) = \frac{1}{\nu}\mathbb{E}_{\hat{q}_{\mathsf{d}}}[-\rho_\theta f''(\rho_\theta)\nabla_\theta \log \rho_\theta] + \mathbb{E}_{\hat{q}_{\mathsf{n}}}[\rho_\theta^2 f''(\rho_\theta)\nabla_\theta \log \rho_\theta],$$

$$\nabla_\theta^2 \hat{\mathcal{L}}_f^{\mathsf{nce}}(\theta) = \frac{1}{\nu}\mathbb{E}_{\hat{q}_{\mathsf{d}}}[(-\rho_\theta f''(\rho_\theta) - \rho_\theta^2 f'''(\rho_\theta))\nabla_\theta \log \rho_\theta \nabla_\theta^\intercal \log \rho_\theta - \rho_\theta f''(\rho_\theta)\nabla_\theta^2 \log \rho_\theta]$$
$$+ \mathbb{E}_{\hat{q}_{\mathsf{n}}}[(2\rho_\theta^2 f''(\rho_\theta) + \rho_\theta^3 f'''(\rho_\theta))\nabla_\theta \log \rho_\theta \nabla_\theta^\intercal \log \rho_\theta + \rho_\theta^2 f''(\rho_\theta)\nabla_\theta^2 \log \rho_\theta].$$

*In particular, we have*

$$\nabla_\theta \mathcal{L}_f^{\mathsf{nce}}(\theta^\star) = \mathbb{E}[\nabla_\theta \hat{\mathcal{L}}_f^{\mathsf{nce}}(\theta^\star)] = 0,$$

$$\nabla_\theta^2 \mathcal{L}_f^{\mathsf{nce}}(\theta^\star) = \frac{1}{\nu}\mathbb{E}_{q_{\mathsf{d}}}\left[\rho_{\theta^\star} f''(\rho_{\theta^\star})\nabla_\theta \log \rho_{\theta^\star} \nabla_\theta^\intercal \log \rho_{\theta^\star}\right].$$

*For an exponential family model $\phi_\theta(x) = \exp(\langle \theta, \psi(x) \rangle)$, we have*

$$\nabla_\theta \hat{\mathcal{L}}_f^{\mathsf{nce}}(\theta) = \frac{1}{\nu}\mathbb{E}_{\hat{q}_{\mathsf{d}}}[\psi \xi_{\mathsf{nce},f,\mathsf{d}}^{(1)}(\rho_\theta)] + \mathbb{E}_{\hat{q}_{\mathsf{n}}}[\psi \xi_{\mathsf{nce},f,\mathsf{n}}^{(1)}(\rho_\theta)],$$

$$\nabla_\theta^2 \hat{\mathcal{L}}_f^{\text{nce}}(\theta) = \frac{1}{\nu}\mathbb{E}_{\hat{q}_\text{d}}[\psi\psi^\mathsf{T}\xi_{\text{nce},f,\text{d}}^{(2)}(\rho_\theta)] + \mathbb{E}_{\hat{q}_\text{n}}[\psi\psi^\mathsf{T}\xi_{\text{nce},f,\text{n}}^{(2)}(\rho_\theta)],$$

*where*

$$\xi_{\text{nce},f,\text{d}}^{(1)}(\rho) = -\rho f''(\rho),$$
$$\xi_{\text{nce},f,\text{n}}^{(1)}(\rho) = \rho^2 f''(\rho),$$
$$\xi_{\text{nce},f,\text{d}}^{(2)}(\rho) = \rho g_f(\rho),$$
$$\xi_{\text{nce},f,\text{n}}^{(2)}(\rho) = \rho^2(f''(\rho) - g_f(\rho))$$

*In particular, if $q_\text{d}(x) \equiv \phi_{\theta^\star}(x)$ for some $\theta^\star$,*

$$\nabla_\theta \mathcal{L}_f^{\text{nce}}(\theta^\star) = \mathbb{E}[\nabla_\theta \hat{\mathcal{L}}_f^{\text{nce}}(\theta^\star)] = 0,$$
$$\nabla_\theta^2 \mathcal{L}_f^{\text{nce}}(\theta^\star) = \mathbb{E}[\nabla_\theta^2 \hat{\mathcal{L}}_f^{\text{nce}}(\theta^\star)] = \frac{1}{\nu}\mathbb{E}_{q_\text{d}}[\psi\psi^\mathsf{T}f''(\rho_{\theta^\star})] = \mathbb{E}_{q_\text{n}}[\psi\psi^\mathsf{T}\rho_{\theta^\star}f''(\rho_{\theta^\star})].$$

## B.2. $\alpha$-CentNCE

Recall that

$$\tilde{r}_{\theta;\alpha}(x) = \frac{r_\theta(x)}{(\mathbb{E}_{q_\text{n}}[r_\theta^\alpha(x)])^{\frac{1}{\alpha}}}$$

and

$$\tilde{\mathcal{L}}_\alpha(\theta) \triangleq \tilde{\mathcal{L}}_\alpha(\theta; q_\text{d}, q_\text{n}) \triangleq \frac{1}{1-\alpha}\mathbb{E}_{q_\text{d}}[\tilde{r}_{\theta;\alpha}^{\alpha-1}(x)] = \frac{1}{1-\alpha}\mathbb{E}_{q_\text{d}}[r_\theta^{\alpha-1}(x)](\mathbb{E}_{q_\text{n}}[r_\theta^\alpha(x)])^{\frac{1-\alpha}{\alpha}}.$$

### B.2.1. DERIVATIVES

It is easy to check that

**Lemma B.4.**

$$\nabla_\theta \log \tilde{r}_{\theta;\alpha} = \psi - \mathbb{E}_{q_\text{n}}[\psi\tilde{r}_{\theta;\alpha}^\alpha],$$
$$\nabla_\theta^2 \log \tilde{r}_{\theta;\alpha} = -\alpha\{\mathbb{E}_{q_\text{n}}[\psi\psi^\mathsf{T}\tilde{r}_{\theta;\alpha}^\alpha] - \mathbb{E}_{q_\text{n}}[\psi\tilde{r}_{\theta;\alpha}^\alpha]\mathbb{E}_{q_\text{n}}[\psi\tilde{r}_{\theta;\alpha}^\alpha]^\mathsf{T}\}.$$

### B.2.2. AN ALTERNATIVE INTERPRETATION OF GLOBALGISO

Consider an unnormalized model $\{\phi_\theta(x) : \theta \in \Theta\}$. For a data distribution $q_\text{d}(x)$, to which we have sample access, assume that there exists $\theta^* \in \Theta$ such that $\phi_{\theta^*}(x) \propto q_\text{d}(x)$. Let $q_\text{n}(x)$ be a reference distribution which makes $\mathbb{E}_{q_\text{n}}[\log \phi_\theta(x)]$ exist for any $\theta \in \Theta$. We define a "centered" unnormalized model

$$\tilde{\phi}_\theta(x) \triangleq \frac{\phi_\theta(x)}{e^{\mathbb{E}_{q_\text{n}}[\log \phi_\theta(x)]}}$$

and denote its partition function as $\tilde{Z}(\theta) \triangleq \int \tilde{\phi}_\theta(x)\,\mathrm{d}x$. We remark that

$$\mathbb{E}_{q_\text{n}}[\log \tilde{\phi}_\theta(x)] = \mathbb{E}_{q_\text{n}}[\log \phi_\theta(x)] - \mathbb{E}_{q_\text{n}}[\log \phi_\theta(x)] = 0. \tag{10}$$

We then define an objective for distribution learning as

$$\mathcal{L}_{\text{giso}}(\theta) \triangleq \mathbb{E}_{q_\text{d}}\left[\frac{q_\text{n}(x)}{\tilde{\phi}_\theta(x)}\right].$$

If $\phi_\theta(x) = \exp(\langle\theta, \psi(x)\rangle)$ is an exponential family distribution over a compact support $\mathcal{X}$ and $q_\text{n}(x)$ is the uniform distribution over $\mathcal{X}$, then it boils down to the objective function studied by (Shah et al., 2021b).

**Fisher Consistency**   To understand the property of the objective, we introduce another unnormalized model

$$\xi_{\theta_1,\theta_2}(x) \triangleq \frac{\tilde{\phi}_{\theta_1}(x)}{\tilde{\phi}_{\theta_2}(x)} q_{\mathsf{n}}(x),$$

and denote its partition function and the normalized distribution as

$$Z(\theta_1,\theta_2) \triangleq \int \xi_{\theta_1,\theta_2}(x)\,\mathrm{d}x \quad \text{and} \quad q_{\theta_1,\theta_2}(x) \triangleq \frac{\xi_{\theta_1,\theta_2}(x)}{Z(\theta_1,\theta_2)}.$$

We can then show that

**Theorem B.1.**

$$\log \mathcal{L}_{\mathsf{giso}}(\theta) = D(q_{\mathsf{n}}\|q_{\theta^*,\theta}) - \log \tilde{Z}(\theta^*).$$

As an immediate corollary, we can prove the Fisher consistency of the objective function.

**Corollary B.1** (Fisher consistency)**.** *Let $\theta^\star \in \arg\min_\theta \mathcal{L}_{\mathsf{giso}}(\theta)$. Then, $\phi_{\theta^\star}(x) \propto q_{\mathsf{d}}(x)$ for $x \in \mathrm{supp}(q_{\mathsf{n}})$.*

The proof of Theorem B.1 readily follows from the following lemmas.

**Lemma B.5.**

$$\mathcal{L}_{\mathsf{giso}}(\theta) = \frac{Z(\theta^*,\theta)}{\tilde{Z}(\theta^*)}.$$

*Proof.* Consider

$$\begin{aligned}
\mathcal{L}_{\mathsf{giso}}(\theta) &\triangleq \int q_{\mathsf{d}}(x)\frac{q_{\mathsf{n}}(x)}{\tilde{\phi}_\theta(x)}\,\mathrm{d}x \\
&= \int \frac{\tilde{\phi}_{\theta^*}(x)}{\tilde{Z}(\theta^*)}\frac{q_{\mathsf{n}}(x)}{\tilde{\phi}_\theta(x)}\,\mathrm{d}x \\
&= \frac{1}{\tilde{Z}(\theta^*)}\int \frac{\tilde{\phi}_{\theta^*}(x)}{\tilde{\phi}_\theta(x)}q_{\mathsf{n}}(x)\,\mathrm{d}x \\
&= \frac{Z(\theta^*,\theta)}{\tilde{Z}(\theta^*)}.
\end{aligned}$$
$\square$

**Lemma B.6.** *For any $\theta_1, \theta_2 \in \Theta$,*

$$D(q_{\mathsf{n}}\|q_{\theta_1,\theta_2}) = \log Z(\theta_1,\theta_2).$$

*Proof.* Consider

$$\begin{aligned}
D(q_{\mathsf{n}}\|q_{\theta_1,\theta_2}) &= \mathbb{E}_{q_{\mathsf{n}}}\left[\log \frac{q_{\mathsf{n}}(x)}{q_{\theta_1,\theta_2}(x)}\right] \\
&= \mathbb{E}_{q_{\mathsf{n}}}\left[\log Z(\theta_1,\theta_2) + \log \frac{\tilde{\phi}_{\theta_2}(x)}{\tilde{\phi}_{\theta_1}(x)}\right] \\
&= \log Z(\theta_1,\theta_2).
\end{aligned}$$

Here, in the last equality, we use the fact that $\log \tilde{\phi}_\theta(x)$ is centered under $q_{\mathsf{n}}(x)$, as alluded to earlier in Eq. (10). $\square$

### B.2.3. PROOF OF THEOREM 3.1

**Theorem 3.1** ($\alpha$-CentNCE subsumes MLE and GlobalGISO)**.** *The following holds:*

1. *($\alpha = 0$: GlobalGISO) For an exponential family $\phi_\theta(x)$, if $\mathcal{X}$ is bounded and $q_{\mathsf{n}}(x)$ is a uniform distribution over $\mathcal{X}$, the 0-CentNCE objective $\tilde{\mathcal{L}}_0(\theta; q_{\mathsf{d}}, q_{\mathsf{n}})$ is equivalent to GlobalGISO (Shah et al., 2021b).*

2. ($\alpha = 1$: MLE) If $Z_1(\theta)$ is assumed to be computable for each $\theta$, the 1-CentNCE objective $\tilde{\mathcal{L}}_1(\theta; \hat{q}_d, q_n)$ is equivalent to MLE (Fisher, 1922).

3. ($\alpha = 1$: MC-MLE) If $Z_1(\theta) = \mathbb{E}_{q_n}[\frac{\phi_\theta(x)}{q_n(x)}]$ is estimated with empirical noise distribution $\hat{q}_n(x)$, the 1-CentNCE objective $\tilde{\mathcal{L}}_1(\theta; \hat{q}_d, \hat{q}_n)$ is equivalent to MC-MLE (Geyer, 1994).

*Proof.* When $\alpha \to 1$, the centering becomes the standard normalization, i.e.,

$$\tilde{\phi}_{\theta;1}(x) \triangleq \lim_{\alpha \to 1} \frac{\phi_\theta(x)}{(\mathbb{E}_{q_n}[(\frac{\phi_\theta(x)}{q_n(x)})^\alpha])^{1/\alpha}} = \frac{\phi_\theta(x)}{\mathbb{E}_{q_n}[\frac{\phi_\theta(x)}{q_n(x)}]},$$

and thus the objective becomes equivalent to the MC-MLE objectives:

$$\tilde{\mathcal{L}}_1(\theta; q_d, q_n) = \mathbb{E}_{q_d(x)}\left[\log \frac{1}{\tilde{\phi}_{\theta;1}(x)}\right] = \mathbb{E}_{q_d(x)}\left[\log \frac{1}{\phi_\theta(x)}\right] + \log \mathbb{E}_{q_n}\left[\frac{\phi_\theta(x)}{q_n(x)}\right].$$

When $Z_1(\theta) = \mathbb{E}_{q_n}[\frac{\phi_\theta(x)}{q_n(x)}] = Z(\theta)$ is assumed to be computable, this becomes equivalent to MLE.

When $\alpha \to 0$, the centering becomes

$$\tilde{\phi}_{\theta;0}(x) \triangleq \lim_{\alpha \to 0} \frac{\phi_\theta(x)}{(\mathbb{E}_{q_n}[(\frac{\phi_\theta(x)}{q_n(x)})^\alpha])^{1/\alpha}} = \frac{\phi_\theta(x)}{e^{\mathbb{E}_{q_n(x)}[\log \frac{\phi_\theta(x)}{q_n(x)}]}}$$

and the objective becomes

$$\tilde{\mathcal{L}}_0(\theta; q_d, q_n) = \mathbb{E}_{q_d(x)}\left[\frac{q_n(x)}{\tilde{\phi}_{\theta;0}(x)}\right] = \mathbb{E}_{q_d(x)}\left[\frac{q_n(x)}{\phi_\theta(x)}\right] e^{\mathbb{E}_{q_n(x)}[\log \frac{\phi_\theta(x)}{q_n(x)}]}. \tag{11}$$

In particular, for the exponential family, we have

$$\log Z_0(\theta) \triangleq \mathbb{E}_{q_n(x)}\left[\log \frac{\phi_\theta(x)}{q_n(x)}\right] = \langle \theta, \bar{\psi}_q \rangle - \mathbb{E}_{q_n(x)}[\log q_n(x)],$$

where $\bar{\psi}_q \triangleq \mathbb{E}_{q_n(x)}[\psi(x)]$, and thus the objective becomes

$$\tilde{\mathcal{L}}_0(\theta; q_d, q_n) = \mathbb{E}_{q_d(x)}[q_n(x) \exp(\langle \theta, \psi(x) - \bar{\psi}_q \rangle)]$$

modulo additive and multiplicative constants. When the underlying domain $\mathcal{X}$ is assumed to be *bounded*, we can set $q_n(x)$ as the uniform distribution over $\mathcal{X}$. In this case, the NCE objective boils down to the global generalized interactive screening objective (GlobalGISO) studied by Shah et al. (2021b). □

To provide a comprehensive view, we summarize the connections in terms of the objective functions that correspond to the unified estimators in Table 5.

### B.2.4. PROOF OF THEOREM 3.2

**Theorem 3.2** ($f_\alpha$-NCE and $\alpha$-CentNCE estimators are equivalent). *For a set $A \subset \Theta \times \mathbb{R}$ in the augmented parameter space, let $A|_\Theta \triangleq \{\theta : (\theta, \nu) \in A \text{ for some } \nu \in \mathbb{R}\}$ denote the subset corresponding to $\Theta$. Then,*

$$\left.\operatorname*{arg\,min}_{\underline{\theta} = (\theta, \nu) \in \Theta \times \mathbb{R}} \mathcal{L}^{\text{nce}}_{f_\alpha}(\underline{\theta}; \hat{q}_d, \hat{q}_n)\right|_\Theta = \operatorname*{arg\,min}_{\theta \in \Theta} \mathcal{L}^{\text{cent}}_\alpha(\theta; \hat{q}_d, \hat{q}_n).$$

*Proof.* On one hand, we first note that $\nu \mapsto \mathcal{L}^{\text{nce}}_{f_\alpha}(\underline{\theta}; \hat{q}_d, \hat{q}_n)$ is convex, and for each $\theta$, the minimizer $\nu^*_\alpha(\theta)$ of the centered objective satisfies

$$e^{\nu^*(\theta)} = \frac{\mathbb{E}_{\hat{q}_d}[r_\theta^{\alpha-1}]}{\mathbb{E}_{\hat{q}_n}[r_\theta^\alpha]}.$$

Table 5. Existing estimators as special instances of NCE estimators.

| Existing estimators | Corresponding NCE objective |
|---|---|
| MLE (Fisher, 1922) | $\mathcal{L}_1^{\text{cent}}(\theta; \hat{q}_{\text{d}}, q_{\text{n}})$ |
| GlobalGISO (Shah et al., 2023) | $\mathcal{L}_0^{\text{cent}}(\theta; \hat{q}_{\text{d}}, q_{\text{n}})$ |
| MC-MLE (Geyer, 1994; Jiang et al., 2023) | $\mathcal{L}_1^{\text{cent}}(\theta; \hat{q}_{\text{d}}, \hat{q}_{\text{n}})$ |
| IS (Pihlaja et al., 2010; Riou-Durand & Chopin, 2018) | $\mathcal{L}_{f_1}^{\text{nce}}(\theta; \hat{q}_{\text{d}}, \hat{q}_{\text{n}})$ |
| eNCE (Liu et al., 2021) | $\mathcal{L}_{f_{\frac{1}{2}}}^{\text{nce}}(\theta; \hat{q}_{\text{d}}, \hat{q}_{\text{n}})$ |
| Pseudo likelihood (Besag, 1975) | $\mathcal{L}_1^{\text{cent}}(\theta; \hat{q}_{\text{d}}, q_{\text{n}})$ (local) |
| GISO (Vuffray et al., 2016; 2021), ISODUS (Ren et al., 2021) | $\mathcal{L}_0^{\text{cent}}(\theta; \hat{q}_{\text{d}}, q_{\text{n}})$ (local) |

Moreover,

$$\nabla_\theta \mathcal{L}_{f_\alpha}^{\text{nce}}(\theta; \hat{q}_{\text{d}}, \hat{q}_{\text{n}}) = -e^{\nu(\alpha-1)} \mathbb{E}_{\hat{q}_{\text{d}}}[r_\theta^{(\alpha-1)} \nabla_\theta \log r_\theta] + e^{\nu\alpha} \mathbb{E}_{\hat{q}_{\text{n}}}[r_\theta^\alpha \nabla_\theta \log r_\theta],$$

so that the $f_\alpha$-NCE estimator $\hat{\underline{\theta}}_{f_\alpha}^{\text{nce}}(\hat{q}_{\text{d}}, \hat{q}_{\text{n}}) = (\hat{\theta}_{f_\alpha}^{\text{nce}}(\hat{q}_{\text{d}}, \hat{q}_{\text{n}}), \hat{\nu}_{f_\alpha}^{\text{nce}}(\hat{q}_{\text{d}}, \hat{q}_{\text{n}}))$ satisfies

$$\mathbb{E}_{\hat{q}_{\text{n}}}[r_\theta^\alpha \nabla_\theta \log r_\theta] \mathbb{E}_{\hat{q}_{\text{d}}}[r_\theta^{\alpha-1}] = \mathbb{E}_{\hat{q}_{\text{n}}}[r_\theta^\alpha] \mathbb{E}_{\hat{q}_{\text{d}}}[r_\theta^{\alpha-1} \nabla_\theta \log r_\theta]. \tag{12}$$

On the other hand, we have

$$\nabla_\theta \mathcal{L}_\alpha^{\text{cent}}(\theta; \hat{q}_{\text{d}}, \hat{q}_{\text{n}}) = -\mathbb{E}_{\hat{q}_{\text{d}}}[r_\theta^{\alpha-1} \nabla_\theta \log r_\theta](\mathbb{E}_{\hat{q}_{\text{n}}}[r_\theta^\alpha])^{\frac{1-\alpha}{\alpha}} + \mathbb{E}_{\hat{q}_{\text{d}}}[r_\theta^{\alpha-1}](\mathbb{E}_{\hat{q}_{\text{n}}}[r_\theta^\alpha])^{\frac{1-2\alpha}{\alpha}} \mathbb{E}_{\hat{q}_{\text{n}}}[r_\theta^\alpha \nabla_\theta \log r_\theta],$$

which implies that the $\alpha$-CentNCE estimator $\hat{\theta}_\alpha^{\text{cent}}(q_{\text{d}}, q_{\text{n}})$ is also a root of Eq. (12). This establishes the desired equivalence. $\qquad\square$

### B.3. $f$-CondNCE

B.3.1. DERIVATIVES

We first note that

**Lemma B.7.**

$$\nabla r_\theta(x, y) = r_\theta(x, y) \nabla_\theta \log r_\theta(x, y),$$
$$\nabla r_\theta^{-1}(x, y) = -\frac{1}{r_\theta^2(x, y)} \nabla r_\theta(x, y) = -\frac{1}{r_\theta(x, y)} \nabla_\theta \log r_\theta(x, y).$$

*In particular, for an exponential family distribution $\phi_\theta(x) = \exp(\langle \theta, \psi(x) \rangle)$, we have*

$$\nabla_\theta \log \rho_\theta(x, y) = \psi(x) - \psi(y),$$
$$\nabla_\theta^2 \log \rho_\theta(x, y) = 0.$$

**Lemma B.8** (Conditional NCE: derivatives). *Let $\rho_\theta(x, y) \triangleq \rho_\theta$ for a shorthand.*

$$\nabla_\theta \mathcal{L}_f^{\text{cond}}(\theta) = \mathbb{E}_{q_{\text{d}}(x)\pi(y|x)}[(\rho_\theta f''(\rho_\theta) + \rho_\theta^{-2} f''(\rho_\theta^{-1})) \nabla_\theta \log \rho_\theta],$$
$$\nabla_\theta^2 \mathcal{L}_f^{\text{cond}}(\theta) = \mathbb{E}_{q_{\text{d}}(x)\pi(y|x)}[(-\rho_\theta f''(\rho_\theta) - \rho_\theta^2 f'''(\rho_\theta)) \nabla_\theta \log \rho_\theta \nabla_\theta^\mathsf{T} \log \rho_\theta - \rho_\theta f''(\rho_\theta) \nabla_\theta^2 \log \rho_\theta$$
$$+ (2\rho_\theta^{-2} f''(\rho_\theta^{-1}) + \rho_\theta^{-3} f'''(\rho_\theta^{-1})) \nabla_\theta \log \rho_\theta \nabla_\theta^\mathsf{T} \log \rho_\theta + \rho_\theta^{-2} f''(\rho_\theta^{-1}) \nabla_\theta^2 \log \rho_\theta].$$

*For an exponential family distribution $\phi_\theta(x) = \exp(\langle \theta, \psi(x) \rangle)$, we have*

$$\nabla_\theta \mathcal{L}_f^{\text{cond}}(\theta) = \mathbb{E}_{q_{\text{d}}(y)\pi(x|y)}[(\psi(x) - \psi(y)) \xi_{\text{cond}, f}^{(1)}(\rho_\theta(x, y))],$$
$$\nabla_\theta^2 \mathcal{L}_f^{\text{cond}}(\theta) = \mathbb{E}_{q_{\text{d}}(y)\pi(x|y)}[(\psi(x) - \psi(y))(\psi(x) - \psi(y))^\mathsf{T} \xi_{\text{cond}, f}^{(2)}(\rho_\theta(x, y))],$$

*where*

$$\xi_{\text{cond},f}^{(1)}(\rho) \triangleq \rho^{-1}f''(\rho^{-1}) + \rho^2 f''(\rho) = -\xi_{\text{nce},f,\text{d}}^{(1)}(\rho^{-1}) + \xi_{\text{nce},f,\text{n}}^{(1)}(\rho),$$

$$\xi_{\text{cond},f}^{(2)}(\rho) \triangleq \rho^{-1}g_f(\rho^{-1}) + \rho^2(f''(\rho) - g_f(\rho)) = \xi_{\text{nce},f,\text{d}}^{(2)}(\rho^{-1}) + \xi_{\text{nce},f,\text{n}}^{(2)}(\rho).$$

*In particular,*

$$\nabla_\theta \mathcal{L}_f^{\text{cond}}(\theta^\star) = 0,$$

$$\nabla_\theta^2 \mathcal{L}_f^{\text{cond}}(\theta^\star) = \mathbb{E}_{q_d(y)\pi(x|y)}[(\psi(x) - \psi(y))(\psi(x) - \psi(y))^\intercal \rho_\theta^2 f''(\rho_\theta)].$$

### B.3.2. PROOF OF THEOREM 3.4

**Theorem 3.4** (Asymptotic behavior of empirical $f$-CondNCE for small $\epsilon$). *The empirical $f$-CondNCE objective can be written as*

$$\mathcal{L}_f^{\text{cond}}(\phi_\theta; \hat{q}_\text{d}, \hat{q}_\text{s}) = -f(1)$$
$$+ 2f''(1)\mathbb{E}_{\hat{q}_\text{d}(x)\hat{q}_\text{s}(v)}[\nabla_x \log \phi_\theta(x)^\intercal v]\epsilon$$
$$+ f''(1)\mathcal{L}^{\text{ssm}}(\phi_\theta; \hat{q}_\text{d}, \hat{q}_\text{s})\epsilon^2 + o(\epsilon^2).$$

*Here, we define the empirical sliced SM (SSM) objective ([Song et al., 2020](#))*

$$\mathcal{L}^{\text{ssm}}(\phi_\theta; \hat{q}_\text{d}, \hat{q}_\text{s})$$
$$\triangleq \mathbb{E}_{\hat{q}_\text{d}(x)\hat{q}_\text{s}(v)}\left[v^\intercal \nabla_x^2 \log \phi_\theta(x)v + \frac{1}{2}(v^\intercal \nabla_x \log \phi_\theta(x))^2\right].$$

*Proof.* Let $\hat{\mathcal{C}}_f(\theta, \epsilon) \triangleq \hat{\mathcal{L}}_f^{\text{cond}}(\phi_\theta; \hat{q}_\text{d}, \hat{q}_\text{s})$. Note that

$$\hat{\mathcal{C}}_f(\theta, \epsilon) = \mathbb{E}_{\hat{q}_\text{d}(x)\hat{q}_\text{s}(v)}[-f'(r) + r^{-1}f'(r^{-1}) - f(r^{-1})],$$

where we set $r = \frac{\phi_\theta(x)}{\phi_\theta(x+\epsilon v)}$ as a shorthand notation. Since by chain rule we have $\frac{\partial}{\partial \epsilon}\log r = -\nabla_x \log \phi_\theta(x + \epsilon v)^\intercal v$, we have

$$\frac{\partial}{\partial \epsilon}\hat{\mathcal{C}}_f(\theta, \epsilon) = \mathbb{E}_{\hat{q}_\text{d}(x)\hat{q}_\text{s}(v)}\left[\nabla_x \log \phi_\theta(x + \epsilon v)^\intercal v\left(rf''(r) + \frac{1}{r^2}f''\left(\frac{1}{r}\right)\right)\right],$$

and

$$\frac{\partial^2}{\partial \epsilon^2}\hat{\mathcal{C}}_f(\theta, \epsilon) = \mathbb{E}_{\hat{q}_\text{d}(x)\hat{q}_\text{s}(v)}\left[v^\intercal \nabla_x^2 \log \phi_\theta(x + \epsilon v)v\left(rf''(r) + \frac{1}{r^2}f''\left(\frac{1}{r}\right)\right)\right.$$
$$\left. + (\nabla_x \log \phi_\theta(x + \epsilon v)^\intercal v)^2\left(rf''(r) + r^2 f'''(r) - \frac{2}{r^2}f''\left(\frac{1}{r}\right) - \frac{1}{r^4}f'''\left(\frac{1}{r}\right)\right)\right].$$

Hence,

$$\hat{\mathcal{C}}_f(\theta, \epsilon)\Big|_{\epsilon=0} = -f(1),$$

$$\frac{\partial}{\partial \epsilon}\hat{\mathcal{C}}_f(\theta, \epsilon)\Big|_{\epsilon=0} = 2f''(1)\mathbb{E}_{\hat{q}_\text{d}(x)\hat{q}_\text{s}(v)}[\nabla_x \log \phi_\theta(x)^\intercal v],$$

$$\frac{\partial^2}{\partial \epsilon^2}\hat{\mathcal{C}}_f(\theta, \epsilon)\Big|_{\epsilon=0} = f''(1)\left(2\mathbb{E}_{\hat{q}_\text{d}(x)\hat{q}_\text{s}(v)}[v^\intercal \nabla_x^2 \log \phi_\theta(x)v] + \mathbb{E}_{\hat{q}_\text{d}(x)\hat{q}_\text{s}(v)}[(\nabla_x^2 \log \phi_\theta(x)^T v)^2]\right)$$
$$= 2f''(1)\hat{\mathcal{L}}^{\text{ssm}}(\phi_\theta; \hat{q}_\text{d}, \hat{q}_\text{s}).$$

Plugging these to the second-order Taylor approximation of $\epsilon \mapsto \hat{\mathcal{C}}_f(\theta, \epsilon)$ around $\epsilon = 0$, i.e.,

$$\hat{\mathcal{C}}_f(\theta, \epsilon) = \hat{\mathcal{C}}_f(\theta, \epsilon)\Big|_{\epsilon=0}\epsilon + \frac{\partial}{\partial \epsilon}\hat{\mathcal{C}}_f(\theta, \epsilon)\Big|_{\epsilon=0}\epsilon + \frac{1}{2}\frac{\partial^2}{\partial \epsilon^2}\hat{\mathcal{C}}_f(\theta, \epsilon)\Big|_{\epsilon=0}\epsilon^2 + o(\epsilon^2),$$

concludes the proof. $\qquad\square$

# C. Asymptotic Guarantees

We can establish the asymptotic consistency and normality of the estimators. Though we present the results for exponential family models for simplicity, one can derive the asymptotic covariances for general unnormalized models and generalize the results. All the proofs are straightforward from the application of standard M-estimation theory, see, e.g., (Van der Vaart, 2000), so we omit the proofs.

## C.1. $f$-NCE

**Theorem C.1** ($f$-NCE: asymptotic guarantee). *Let $\hat{\underline{\theta}}^{\mathsf{nce}}_{f;n_{\mathsf{d}},n_{\mathsf{n}}} \triangleq (\hat{\theta}^{\mathsf{nce}}_{f;n_{\mathsf{d}},n_{\mathsf{n}}}, \hat{c}^{\mathsf{nce}}_{f;n_{\mathsf{d}},n_{\mathsf{n}}})$ be a solution of*

$$\hat{\underline{\theta}}^{\mathsf{nce}}_{f;n_{\mathsf{d}},n_{\mathsf{n}}} \in \arg\min_{\underline{\theta}\in\Theta\times\mathbb{R}} \hat{\mathcal{L}}^{\mathsf{nce}}_f(\underline{\theta}).$$

*Let $n_{\mathsf{n}} \triangleq \beta n_{\mathsf{d}}$ for some $\beta > 0$. If $\hat{\mathcal{L}}^{\mathsf{nce}}_f(\underline{\theta}) \xrightarrow{p} \mathcal{L}^{\mathsf{nce}}_f(\underline{\theta})$ as $n_{\mathsf{d}} \to \infty$ uniformly over $\underline{\theta} \in \Theta \times \mathbb{R}$, $\hat{\underline{\theta}}^{\mathsf{nce}}_{f;n_{\mathsf{d}},n_{\mathsf{n}}} \xrightarrow{p} (\theta^\star, c^\star)$ as $n_{\mathsf{d}} \to \infty$. Further, if $\theta^\star \in \mathrm{int}(\Theta)$, we have $\sqrt{n_{\mathsf{d}}}(\hat{\theta}^{\mathsf{nce}}_{f;n_{\mathsf{d}},n_{\mathsf{n}}} - \theta^\star) \xrightarrow{d} \mathcal{N}(0, \mathcal{V}^{\mathsf{nce}}_f)$, where we define $\mathcal{V}^{\mathsf{nce}}_f \triangleq \mathcal{I}^{-1}_f \mathcal{C}_f \mathcal{I}^{-1}_f$,*

$$\mathcal{I}_f \triangleq \mathbb{E}_{q_{\mathsf{d}}}[\rho_{\theta^\star} f''(\rho_{\theta^\star})\underline{\psi}\underline{\psi}^{\mathsf{T}}],$$

$$\mathcal{C}_f \triangleq \mathbb{E}_{q_{\mathsf{d}}}\left[\left(1 + \frac{\nu}{\beta}\rho_{\theta^\star}\right)\rho^2_{\theta^\star} f''(\rho_{\theta^\star})^2\underline{\psi}\underline{\psi}^{\mathsf{T}}\right] - \left(1 + \frac{1}{\beta}\right)\mathbb{E}_{q_{\mathsf{d}}}[\rho_{\theta^\star} f''(\rho_{\theta^\star})\underline{\psi}]\mathbb{E}_{q_{\mathsf{d}}}[\rho_{\theta^\star} f''(\rho_{\theta^\star})\underline{\psi}]^{\mathsf{T}},$$

*for $\underline{\psi}(x) \triangleq [\psi(x);1]^{\mathsf{T}} \in \mathbb{R}^{p+1}$, provided that $\mathcal{I}_f$ is invertible. In particular, the asymptotic covariance $\mathcal{V}^{\mathsf{nce}}_f$ satisfies $\mathcal{V}^{\mathsf{nce}}_f \succeq \mathcal{V}^{\mathsf{nce}}_{f_{\log}}$, or equivalently $\mathcal{V}^{\mathsf{nce}}_f - \mathcal{V}^{\mathsf{nce}}_{f_{\log}}$ is a PSD matrix, for any $f$.*

This result has been known, but we present a rephrased version here to contextualize our contribution. The asymptotic convergence beyond exponential family was established in (Gutmann & Hyvärinen, 2012) for $f_{\log}$-NCE and in (Pihlaja et al., 2010; Uehara et al., 2018) for $f$-NCE. The optimality of $f_{\log}$ was established in (Uehara et al., 2018). It was independently proved that the original $f_{\log}$-NCE estimator asymptotic covariance not larger than that of the $f_1$-NCE estimator (and thus the MC-MLE estimator), which they call the IS estimator, in Loewner order (Riou-Durand & Chopin, 2018). In the same paper, the asymptotic guarantee for the $f_{\log}$-NCE and IS estimators was shown for a general unnormalized distribution under a non-i.i.d. setting in (Barthelmé & Chopin, 2015).

## C.2. $\alpha$-CentNCE

**Theorem C.2** (CentNCE: asymptotic guarantee). *Assume that any expectation over $q_{\mathsf{n}}$ in the $\alpha$-CentNCE objective can be computed for any $\theta$ without samples from $q_{\mathsf{n}}$. Let $\hat{\theta}^{\mathsf{cent}}_{\alpha;n_{\mathsf{d}}}$ be a solution of*

$$\hat{\theta}^{\mathsf{cent}}_{\alpha;n_{\mathsf{d}}} \in \arg\min_{\theta\in\Theta} \mathcal{L}^{\mathsf{cent}}_\alpha(\theta; \hat{q}_{\mathsf{d}}, q_{\mathsf{n}}).$$

*If $\mathcal{L}^{\mathsf{cent}}_\alpha(\theta; \hat{q}_{\mathsf{d}}, q_{\mathsf{n}}) \xrightarrow{p} \mathcal{L}^{\mathsf{cent}}_\alpha(\theta; q_{\mathsf{d}}, q_{\mathsf{n}})$ as $n_{\mathsf{d}} \to \infty$ uniformly over $\theta \in \Theta$, $\hat{\theta}^{\mathsf{cent}}_{\alpha;n_{\mathsf{d}}} \xrightarrow{p} \theta^\star$ as $n_{\mathsf{d}} \to \infty$. Further, if $\theta^\star \in \mathrm{int}(\Theta)$, we have $\sqrt{n_{\mathsf{d}}}(\hat{\theta}^{\mathsf{cent}}_{\alpha;n_{\mathsf{d}}} - \theta^\star) \xrightarrow{d} \mathcal{N}(0, \mathcal{V}^{\mathsf{cent}}_\alpha)$, where we define $\mathcal{V}^{\mathsf{cent}}_\alpha \triangleq \tilde{\mathcal{I}}^{-1}_\alpha \tilde{\mathcal{C}}_\alpha \tilde{\mathcal{I}}^{-1}_\alpha$,*

$$\tilde{\mathcal{I}}_\alpha \triangleq (1-\alpha)\mathbb{E}_{q_{\mathsf{d}}}[\tilde{r}^{\alpha-1}_{\theta^\star;\alpha}(\psi - \mathbb{E}_{q_{\mathsf{n}}}[\tilde{r}^\alpha_{\theta^\star;\alpha}\psi])(\psi - \mathbb{E}_{q_{\mathsf{n}}}[\tilde{r}^\alpha_{\theta^\star;\alpha}\psi])^{\mathsf{T}}]$$
$$+ \alpha\mathbb{E}_{q_{\mathsf{d}}}[\tilde{r}^{\alpha-1}_{\theta^\star;\alpha}](\mathbb{E}_{q_{\mathsf{n}}}[\tilde{r}^\alpha_{\theta^\star;\alpha}\psi\psi^{\mathsf{T}}] - \mathbb{E}_{q_{\mathsf{n}}}[\tilde{r}^\alpha_{\theta^\star;\alpha}\psi]\mathbb{E}_{q_{\mathsf{n}}}[\tilde{r}^\alpha_{\theta^\star;\alpha}\psi]^{\mathsf{T}}),$$

$$\tilde{\mathcal{C}}_\alpha \triangleq \mathbb{E}_{q_{\mathsf{d}}}[\tilde{r}^{2(\alpha-1)}_{\theta^\star;\alpha}(\psi - \mathbb{E}_{q_{\mathsf{n}}}[\tilde{r}^\alpha_{\theta^\star;\alpha}\psi])(\psi - \mathbb{E}_{q_{\mathsf{n}}}[\tilde{r}^\alpha_{\theta^\star;\alpha}\psi])^{\mathsf{T}}],$$

*provided that $\tilde{\mathcal{I}}_\alpha$ is invertible. Here, note that $\tilde{r}^\alpha_{\theta^\star;\alpha}(x) = \frac{(\frac{q_{\mathsf{d}}(x)}{q_{\mathsf{n}}(x)})^\alpha}{\mathbb{E}_{q_{\mathsf{n}}}[(\frac{q_{\mathsf{d}}}{q_{\mathsf{n}}})^\alpha]}$.*

In particular, this result recovers the asymptotic convergence of MLE for $\alpha = 1$, and generalizes the analysis of GlobalGISO of (Shah et al., 2023) beyond when $q_{\mathsf{n}}$ is the uniform distribution.

## C.3. $f$-CondNCE

**Theorem C.3** ($f$-CondNCE: asymptotic guarantee). *Let $\hat{\theta}^{\mathsf{cond}}_{f;n_{\mathsf{d}}}$ be a solution of*

$$\hat{\theta}^{\mathsf{cond}}_{f;n_{\mathsf{d}}} \in \arg\min_{\theta\in\Theta} \hat{\mathcal{L}}^{\mathsf{cond}}_f(\theta).$$

*If $\hat{\mathcal{L}}_f^{\mathsf{cond}}(\theta) \overset{p}{\to} \mathcal{L}_f^{\mathsf{cond}}(\theta)$ as $n_{\mathsf{d}} \to \infty$ uniformly over $\theta \in \Theta$, $\hat{\theta}_{f;n_{\mathsf{d}}}^{\mathsf{cond}} \overset{p}{\to} \theta^\star$ as $n_{\mathsf{d}} \to \infty$. Further, if $\theta^\star \in \mathrm{int}(\Theta)$, we have $\sqrt{n_{\mathsf{d}}}(\hat{\theta}_{f;n_{\mathsf{d}}}^{\mathsf{cent}} - \theta^\star) \overset{d}{\to} \mathcal{N}(0, \check{\mathcal{V}}_f^{\mathsf{cond}})$, where we define $\check{\mathcal{V}}_f^{\mathsf{cond}} \triangleq \check{\mathcal{I}}_f^{-1} \check{\mathcal{C}}_f \check{\mathcal{I}}_f^{-1}$,*

$$\check{\mathcal{I}}_f \triangleq \mathbb{E}_{q_{\mathsf{d}}(x)\pi(y|x)}[\rho_{\theta^\star}^2 f''(\rho_{\theta^\star})(\psi(x) - \psi(y))(\psi(x) - \psi(y))^{\mathsf{T}}],$$

$$\check{\mathcal{C}}_f \triangleq \mathbb{E}_{q_{\mathsf{d}}(x)\pi(y|x)}[\xi_{\mathsf{cond},f}^{(1)}(\rho_{\theta^\star})^2 (\psi(x) - \psi(y))(\psi(x) - \psi(y))^{\mathsf{T}}],$$

*provided that $\check{\mathcal{I}}_f$ is invertible. Here, $\rho_\theta = \rho_\theta(x,y)$ and $\xi_{\mathsf{cond},f}^{(1)}(\rho) \triangleq \rho^{-1} f''(\rho^{-1}) + \rho^2 f''(\rho)$.*

# D. Finite-Sample Guarantees

For the finite-sample analysis of the regularized NCE estimators, we invoke the result of Negahban et al. (2012):

**Theorem 4.4** (Negahban et al., 2012, Corollary 1). *Let $z_1, \ldots, z_N$ be i.i.d. samples drawn from a distribution $p(z)$. Let $h_\theta(z)$ be a convex and differentiable function parameterized by $\theta \in \Theta$. Let $\hat{\mathcal{L}}_n(\theta) \triangleq \frac{1}{n} \sum_{i=1}^{n} h_\theta(z_i)$ denote the empirical objective function. Define*

$$\hat{\theta}_n \in \arg\min_\theta \Big\{ \hat{\mathcal{L}}_n(\theta) + \lambda_n \mathcal{R}(\theta) \Big\}, \tag{9}$$

*where $\lambda_n$ is a regularization penalty and $\mathcal{R} \colon \Theta \to \mathbb{R}_{\geq 0}$ is a norm over $\Theta$. Let $\theta^\star \in \arg\min_\theta \mathbb{E}_{p(z)}[h_\theta(z)]$. Assume that*

1. *The regularization penalty $\lambda_n$ satisfies $\lambda_n \geq 2\mathcal{R}^*(\nabla_\theta \hat{\mathcal{L}}_n(\theta^\star))$, where $\mathcal{R}^* \colon \Theta^* \to \mathbb{R}_{\geq 0}$ is a dual norm of $\mathcal{R}$ over the dual space $\Theta^*$;*

2. *The empirical objective $\theta \mapsto \hat{\mathcal{L}}_n(\theta)$ satisfies a restricted strong convexity condition at $\theta = \theta^\star$ with curvature $\kappa > 0$, i.e., $\Delta_{\hat{\mathcal{L}}_n(\theta)}(\theta, \theta^\star) \geq \kappa \|\theta - \theta^\star\|_2^2$.*

*Then, the estimator $\hat{\theta}_n$ in Eq. (9) satisfies*

$$\|\hat{\theta}_n - \theta^\star\|_2 \leq 3 \frac{\lambda_n}{\kappa} \gamma_{\mathcal{R};2}.$$

## D.1. $f$-NCE

**Theorem 4.1** ($f$-NCE: finite-sample guarantee). *Pick a strictly convex function $f \colon \mathbb{R}_+ \to \mathbb{R}$. Define*

$$(\rho_{\min}, \rho_{\max}) \triangleq \Big( \inf_{x \in \mathcal{X}, \underline{\theta} \in \Theta \times \mathbb{R}} \rho_{\underline{\theta}}(x), \sup_{x \in \mathcal{X}, \underline{\theta} \in \Theta \times \mathbb{R}} \rho_{\underline{\theta}}(x) \Big)$$

*and define the quantities in Eq. (5) accordingly. For $\mathsf{r} \in \{\mathsf{d}, \mathsf{n}\}$, define*

$$\lambda_{\min,\mathsf{r}}^{\mathsf{nce}} \triangleq \lambda_{\min}(\mathbb{E}_{q_{\mathsf{r}}}[\psi\psi^{\mathsf{T}}]).$$

*Let $\hat{\theta}_{f,n_{\mathsf{d}},n_{\mathsf{n}}}^{\mathsf{nce},\mathcal{R}}$ be such that*

$$\hat{\theta}_{f,n_{\mathsf{d}},n_{\mathsf{n}}}^{\mathsf{nce},\mathcal{R}} \in \arg\min_{\theta \in \Theta} \Big\{ \mathcal{L}_f^{\mathsf{nce}}(\theta; \hat{q}_{\mathsf{d}}, \hat{q}_{\mathsf{n}}) + \lambda_{n_{\mathsf{d}},n_{\mathsf{n}}} \mathcal{R}(\theta) \Big\}$$

*for some $\lambda_{n_{\mathsf{d}},n_{\mathsf{n}}} > 0$. Then, for any $\Delta > 0$ and $\delta \in (0, 1)$, there exists a choice of $\lambda_{n_{\mathsf{d}},n_{\mathsf{n}}}$ such that $\|\hat{\theta}_{f,n_{\mathsf{d}},n_{\mathsf{n}}}^{\mathsf{nce},\mathcal{R}} - \theta^\star\|_2 \leq \Delta$ with probability $\geq 1 - \delta$, provided that for each $\mathsf{r} \in \{\mathsf{d}, \mathsf{n}\}$,*

$$n_{\mathsf{r}} = \Omega\bigg( \max\bigg\{ \frac{(B_{\mathsf{nce},f,\mathsf{r}}^{(1)})^2 \gamma_{\mathcal{R};2}^2 \gamma_{\mathcal{R}^*;\infty}^2 \psi_{\max}^2}{\Delta^2 (\nu^{-1} b_{\mathsf{nce},f,\mathsf{d}}^{(2)} \lambda_{\min,\mathsf{d}}^{\mathsf{nce}} + b_{\mathsf{nce},f,\mathsf{n}}^{(2)} \lambda_{\min,\mathsf{n}}^{\mathsf{nce}})^2},$$
$$\frac{\gamma_{1;2}^4 \psi_{\max}^4}{(\lambda_{\min,\mathsf{r}}^{\mathsf{nce}})^2} \bigg\} \log \frac{p^2}{\delta} \bigg).$$

We need to show two properties. First, the empirical gradient $\nabla_\theta \hat{\mathcal{L}}_f^{\text{nce}}(\theta)$ is nearly zero at $\theta = \theta^\star$ with high probability (Proposition D.1). Second, the empirical Hessian $\nabla_\theta^2 \hat{\mathcal{L}}_f^{\text{nce}}(\theta)$ has a strictly positive curvature (i.e., exhibiting restricted strong convexity) at $\theta = \theta^\star$ with high probability (Proposition D.2).

**Proposition D.1** (Vanishing gradient). *(cf. (Shah et al., 2021b, Proposition F.1).) Assume Assumption 4.1. For any $\delta \in (0, 1)$, $\epsilon > 0$,*

$$\|\nabla_\theta \hat{\mathcal{L}}_f^{\text{nce}}(\theta^\star)\|_{\max} \leq \epsilon$$

*with probability $\geq 1 - \delta$, if $n_{\mathsf{r}} \geq \frac{2\psi_{\max}^2 (B_{\text{nce},f,\mathsf{r}}^{(1)})^2}{\epsilon^2} \log \frac{2p}{\delta}$ for each $\mathsf{r} \in \{\mathsf{d}, \mathsf{n}\}$.*

*Proof.* Recall from Lemma B.3 that

$$\nabla_\theta \hat{\mathcal{L}}_f^{\text{nce}}(\theta) = -\frac{1}{\nu} \mathbb{E}_{\hat{q}_{\mathsf{d}}}[\psi \rho_\theta f''(\rho_\theta)] + \mathbb{E}_{\hat{q}_{\mathsf{n}}}[\psi \rho_\theta^2 f''(\rho_\theta)].$$

Therefore, we have

$$\mathbb{E}[\partial_{\theta_i} \hat{\mathcal{L}}_f^{\text{nce}}(\theta^\star)] = \partial_{\theta_i} \mathcal{L}_f^{\text{nce}}(\theta^\star) = 0.$$

Since $|\psi_i(x) \xi_{\text{nce},f,\mathsf{d}}^{(1)}(\rho_\theta(x))| \leq \psi_{\max} B_{\text{nce},f,\mathsf{d}}^{(1)}$ and $|\psi_i(x) \xi_{\text{nce},f,\mathsf{d}}^{(1)}(\rho_\theta(x))| \leq \psi_{\max} B_{\text{nce},f,\mathsf{n}}^{(1)}$, by Hoeffding's inequality and union bound, we have

$$\mathbb{P}(|\partial_{\theta_i} \hat{\mathcal{L}}_f^{\text{nce}}(\theta^\star)| \geq \epsilon) \leq 2 \exp\left(-\frac{n_{\mathsf{d}} \epsilon^2}{2\psi_{\max}^2 (B_{\text{nce},f,\mathsf{d}}^{(1)})^2}\right) + 2 \exp\left(-\frac{n_{\mathsf{n}} \epsilon^2}{2\psi_{\max}^2 (B_{\text{nce},f,\mathsf{n}}^{(1)})^2}\right) = \delta,$$

if $n_{\mathsf{d}} \geq \frac{2\psi_{\max}^2 (B_{\text{nce},f,\mathsf{d}}^{(1)})^2}{\epsilon^2} \log \frac{2}{\delta}$ and $n_{\mathsf{n}} \geq \frac{2\psi_{\max}^2 (B_{\text{nce},f,\mathsf{n}}^{(1)})^2}{\epsilon^2} \log \frac{2}{\delta}$. By taking a union bound over $p$ different coordinates of $\theta$, we conclude the proof. $\square$

**Lemma D.1.** *(cf. (Shah et al., 2021a, Lemma E.1)) Assume Assumption 4.1. Let $r$ be either $q_{\mathsf{d}}$ or $q_{\mathsf{n}}$. For any $\epsilon_2 > 0$,*

$$\max_{ij} |\mathbb{E}_{\hat{r}}[\psi_i \psi_j] - \mathbb{E}_r[\psi_i \psi_j]| \leq \epsilon_2,$$

*with probability $\geq 1 - \delta_2$, if*

$$n_r \geq \frac{2\psi_{\max}^4}{\epsilon_2^2} \log \frac{2p^2}{\delta_2}.$$

*Proof.* Since $|\psi_i(x) \psi_j(x)| \leq \psi_{\max}^2$ is a bounded random variable, by Hoeffding's inequality, we have

$$\mathbb{P}_r\{|\mathbb{E}_{\hat{r}}[\psi_i \psi_j] - \mathbb{E}_r[\psi_i \psi_j]| > \epsilon_2\} \leq 2 \exp\left(-\frac{n_r \epsilon_2^2}{2\psi_{\max}^4}\right)$$

Taking a union bound over $i, j \in [p]$ leads to the desired bound. $\square$

Recall that for a function $h: \Theta \to \mathbb{R}$, the Bregman divergence is defined as

$$\Delta_h(\theta, \theta_o) \triangleq h(\theta) - h(\theta_o) - \langle \nabla_\theta h(\theta_o), \theta - \theta_o \rangle.$$

**Proposition D.2** (Restricted strong convexity). *(cf. (Shah et al., 2021a, Proposition E.1)) Under Assumption 4.1,*

$$\Delta_{\hat{\mathcal{L}}_f^{\text{nce}}}(\theta, \theta^\star) \geq \frac{1}{4}\left(\frac{b_{\text{nce},f,\mathsf{d}}^{(2)}}{\nu} \lambda_{\min,\mathsf{d}} + b_{\text{nce},f,\mathsf{n}}^{(2)} \lambda_{\min,\mathsf{n}}\right) \|\theta - \theta^\star\|_2^2$$

*with probability $\geq 1 - \delta$, if $n_{\mathsf{r}} \geq \frac{8\gamma_{1;2}^4 \psi_{\max}^4}{\lambda_{\min,r}^2} \log \frac{4p^2}{\delta}$ for each $\mathsf{r} \in \{\mathsf{d}, \mathsf{n}\}$.*

*Proof.* By the intermediate value theorem, there exists $\xi \in \{t\theta + (1-t)\theta^\star : t \in [0,1]\}$ such that

$$\Delta_{\hat{\mathcal{L}}_f^{\text{nce}}}(\theta, \theta^\star) = \hat{\mathcal{L}}_f^{\text{nce}}(\theta) - \hat{\mathcal{L}}_f^{\text{nce}}(\theta^\star) - \langle \nabla_\theta \hat{\mathcal{L}}_f^{\text{nce}}(\theta^\star), \theta - \theta^\star \rangle$$

$$= \frac{1}{2}(\theta - \theta^\star)^\intercal \nabla_\theta^2 \hat{\mathcal{L}}_f^{\text{nce}}(\xi)(\theta - \theta^\star).$$

Here, note that $\xi$ depends on $\hat{q}_{\text{d}}$ and $\hat{q}_{\text{n}}$. Let $z \triangleq \langle \psi(x), \theta - \theta^\star \rangle$.

$$(\theta - \theta^\star)^\intercal \nabla_\theta^2 \hat{\mathcal{L}}_f^{\text{nce}}(\xi)(\theta - \theta^\star) = \frac{1}{\nu} \mathbb{E}_{\hat{q}_{\text{d}}}[z^2 \rho_\xi g_f(\rho_\xi)] + \mathbb{E}_{\hat{q}_{\text{n}}}[z^2 \rho_\xi^2 (f''(\rho_\xi) - g_f(\rho_\xi)]$$

$$\geq \frac{b_{\text{nce},f,\text{d}}^{(2)}}{\nu} \mathbb{E}_{\hat{q}_{\text{d}}}[z^2] + b_{\text{nce},f,\text{n}}^{(2)} \mathbb{E}_{\hat{q}_{\text{n}}}[z^2]$$

$$= (\theta - \theta^\star)^\intercal \Big( \frac{b_{\text{nce},f,\text{d}}^{(2)}}{\nu} \mathbb{E}_{\hat{q}_{\text{d}}}[\psi\psi^\intercal] + b_{\text{nce},f,\text{n}}^{(2)} \mathbb{E}_{\hat{q}_{\text{n}}}[\psi\psi^\intercal] \Big)(\theta - \theta^\star).$$

We can lower bound the quadratic form as follows. The first term can be lower bounded as

$$(\theta - \theta^\star)^\intercal \mathbb{E}_{\hat{q}_{\text{d}}}[\psi\psi^\intercal](\theta - \theta^\star)$$

$$= (\theta - \theta^\star)^\intercal (\mathbb{E}_{\hat{q}_{\text{d}}}[\psi\psi^\intercal] - \mathbb{E}_{q_{\text{d}}}[\psi\psi^\intercal] + \mathbb{E}_{q_{\text{d}}}[\psi\psi^\intercal])(\theta - \theta^\star)$$

$$= \sum_{ij} (\theta - \theta^\star)_i (\mathbb{E}_{\hat{q}_{\text{d}}}[\psi_i\psi_j] - \mathbb{E}_{q_{\text{d}}}[\psi_i\psi_j])(\theta - \theta^\star)_j + (\theta - \theta^\star)^\intercal \mathbb{E}_{q_{\text{d}}}[\psi\psi^\intercal](\theta - \theta^\star)$$

$$\geq - \sum_{ij} |\theta_i - \theta^\star_i| \cdot |\mathbb{E}_{\hat{q}_{\text{d}}}[\psi_i\psi_j] - \mathbb{E}_{q_{\text{d}}}[\psi_i\psi_j]| \cdot |\theta_j - \theta^\star_j| + \lambda_{\text{min,d}} \|\theta - \theta^\star\|_2^2$$

$$\overset{(a)}{\geq} -\epsilon_2 \|\theta - \theta^\star\|_1^2 + \lambda_{\text{min,d}} \|\theta - \theta^\star\|_2^2$$

$$\overset{(b)}{\geq} -\epsilon_2 \gamma_{1;2}^2 \|\theta - \theta^*\|_2^2 + \lambda_{\text{min,d}} \|\theta - \theta^\star\|_2^2$$

$$= \frac{1}{2} \lambda_{\text{min,d}} \|\theta - \theta^\star\|_2^2$$

with probability $\geq 1 - \delta'$ if $n_{\text{d}} \geq \frac{2\psi_{\max}^4}{\epsilon_2^2} \log \frac{2p^2}{\delta'}$ with $\epsilon_2 = \frac{\lambda_{\text{min,d}}}{2\gamma_{1;2}^2}$. Here, we apply Lemma D.1 in $(a)$, and use the definition of $\gamma_{1;2}$ to bound $\|\theta - \theta^\star\|_1 \leq \gamma_{1;2}\|\theta - \theta^\star\|_2$ in $(b)$.

Hence, by a union bound with $\delta' = \delta/2$, with probability $\geq 1 - \delta$, we have

$$\Delta_{\hat{\mathcal{L}}_f^{\text{nce}}}(\theta, \theta^\star) \geq \frac{1}{4}\Big( \frac{b_{\text{nce},f,\text{d}}^{(2)}}{\nu} \lambda_{\text{min,d}} + b_{\text{nce},f,\text{n}}^{(2)} \lambda_{\text{min,n}} \Big) \|\theta - \theta^\star\|_2^2,$$

if $n_{\text{d}} \geq \frac{8\gamma_{1;2}^4 \psi_{\max}^4}{\lambda_{\text{min,d}}^2} \log \frac{4p^2}{\delta}$ and $n_{\text{n}} \geq \frac{8\gamma_{1;2}^4 \psi_{\max}^4}{\lambda_{\text{min,n}}^2} \log \frac{4p^2}{\delta}$. $\qquad\square$

*Proof of Theorem 4.1.* First, note that

$$\mathcal{R}^*(\nabla_\theta \hat{\mathcal{L}}_f^{\text{nce}}(\theta^\star)) \leq \gamma_{\mathcal{R}^*;\infty} \|\nabla_\theta \hat{\mathcal{L}}_f^{\text{nce}}(\theta^\star)\|_{\max}$$

by definition of $\gamma_{\mathcal{R}^*;\infty}$. Then, by Proposition D.1, we have $\|\nabla_\theta \hat{\mathcal{L}}_f^{\text{nce}}(\theta^\star)\|_{\max} \leq \epsilon$ with probability $\geq 1 - \delta_1$, if

$$n_{\text{r}} \geq \frac{2(B_{\text{nce},f,\text{r}}^{(1)})^2 \psi_{\max}^2}{\epsilon^2} \log \frac{2p}{\delta_1}$$

for each $\text{r} \in \{\text{d}, \text{n}\}$. Given that this event occurs, $\mathcal{R}^*(\nabla_\theta \hat{\mathcal{L}}_f^{\text{nce}}(\theta^\star)) \leq \gamma_{\mathcal{R}^*;\infty}\epsilon$, and thus we set $\lambda_n \leftarrow 2\gamma_{\mathcal{R}^*;\infty}\epsilon$ to satisfy the first condition in Theorem 4.4.

Now, given $\lambda_n \geq 2\mathcal{R}^*(\nabla_\theta \hat{\mathcal{L}}_f^{\text{nce}}(\theta^\star))$, (Negahban et al., 2012, Lemma 1) implies that $\mathcal{R}(\hat{\theta}_{f,n_{\text{d}},n_{\text{n}}}^{\text{nce},\mathcal{R}} - \theta^\star) \leq 4\mathcal{R}(\theta^\star)$, i.e., $\hat{\theta}_{f,n_{\text{d}},n_{\text{n}}}^{\text{nce},\mathcal{R}} - \theta^\star \in 4\Theta$. Then, by Proposition D.2, we have

$$\Delta_{\hat{\mathcal{L}}_f^{\text{nce}}}(\theta, \theta^\star) \geq \kappa \|\theta - \theta^\star\|_2^2$$

with probability $\geq 1 - \delta_2$ if $n_{\sf r} \geq \frac{8\gamma_{1;2}^4\psi_{\max}^4}{\lambda_{\min,{\sf r}}^2}\log\frac{4p^2}{\delta_2}$ for each ${\sf r} \in \{{\sf d},{\sf n}\}$, where

$$\kappa = \frac{1}{4}\Big(\frac{b_{{\sf nce},f,{\sf d}}^{(2)}}{\nu}\lambda_{\min,{\sf d}} + b_{{\sf nce},f,{\sf n}}^{(2)}\lambda_{\min,{\sf n}}\Big).$$

Now, by taking a union bound with $\delta_1 = \delta_2 = \delta/2$, with probability $\geq 1 - \delta$, we have

$$\|\theta - \theta^\star\|_2 \leq \frac{3\lambda_n\gamma_{\mathcal{R};2}}{\kappa} = \frac{6\gamma_{\mathcal{R}^*;\infty}\gamma_{\mathcal{R};2}}{\kappa}\epsilon = \Delta$$

with $\epsilon \leftarrow \frac{\Delta\kappa}{6\gamma_{\mathcal{R}^*;\infty}\gamma_{\mathcal{R};2}}$, provided that

$$n_{\sf r} \geq \max\Big\{\frac{72(B_{{\sf nce},f,{\sf r}}^{(1)})^2\gamma_{\mathcal{R};2}^2\gamma_{\mathcal{R}^*;\infty}^2\psi_{\max}^2}{\Delta^2\kappa^2}\log\frac{4p}{\delta}, \frac{8\gamma_{1;2}^4\psi_{\max}^4}{\lambda_{\min,{\sf r}}^2}\log\frac{8p^2}{\delta}\Big)$$

$$= \max\Big\{\frac{1152(B_{{\sf nce},f,{\sf r}}^{(1)})^2\gamma_{\mathcal{R};2}^2\gamma_{\mathcal{R}^*;\infty}^2\psi_{\max}^2}{\Delta^2(\nu^{-1}b_{{\sf nce},f,{\sf d}}^{(2)}\lambda_{\min,{\sf d}} + b_{{\sf nce},f,{\sf n}}^{(2)}\lambda_{\min,{\sf n}})^2}\log\frac{4p}{\delta}, \frac{8\gamma_{1;2}^4\psi_{\max}^4}{\lambda_{\min,{\sf r}}^2}\log\frac{8p^2}{\delta}\Big\}$$

$$= \Omega\Big(\max\Big\{\frac{(B_{{\sf nce},f,{\sf r}}^{(1)})^2\gamma_{\mathcal{R};2}^2\gamma_{\mathcal{R}^*;\infty}^2\psi_{\max}^2}{\Delta^2(\nu^{-1}b_{{\sf nce},f,{\sf d}}^{(2)}\lambda_{\min,{\sf d}} + b_{{\sf nce},f,{\sf n}}^{(2)}\lambda_{\min,{\sf n}})^2}, \frac{\gamma_{1;2}^4\psi_{\max}^4}{\lambda_{\min,{\sf r}}^2}\Big\}\log\frac{p^2}{\delta}\Big)$$

for each ${\sf r} \in \{{\sf d},{\sf n}\}$. $\qquad\square$

### D.2. $\alpha$-CentNCE

**Lemma D.2** ($\alpha$-CentNCE: derivatives).

$$\nabla_\theta\tilde{\mathcal{L}}_\alpha(\theta) = \mathbb{E}_{q_{\sf d}}[-\tilde{r}_{\theta;\alpha}^{\alpha-1}\nabla_\theta\log\tilde{r}_{\theta;\alpha}],$$

$$\nabla_\theta^2\tilde{\mathcal{L}}_\alpha(\theta) = \mathbb{E}_{q_{\sf d}}[\tilde{r}_{\theta;\alpha}^{\alpha-1}((1-\alpha)\nabla_\theta\log\tilde{r}_{\theta;\alpha}\nabla_\theta\log\tilde{r}_{\theta;\alpha}^{\intercal} - \nabla_\theta^2\log\tilde{r}_{\theta;\alpha})].$$

*Define*

$$\tilde{\mathcal{C}}_{\alpha,n_{\sf d}} \triangleq \mathrm{Cov}(\sqrt{n_{\sf d}}\nabla_\theta\hat{\tilde{\mathcal{L}}}_\alpha(\theta^\star))$$

*for $n_{\sf d} \geq 1$. Then, $\tilde{\mathcal{C}}_{\alpha,n_{\sf d}} = \tilde{\mathcal{C}}_\alpha$ for any $n_{\sf d} \geq 1$, where*

$$\tilde{\mathcal{C}}_\alpha \triangleq \mathbb{E}_{q_{\sf d}}[\tilde{r}_{\theta^\star;\alpha}^{2(\alpha-1)}\nabla_\theta\log\tilde{r}_{\theta^\star;\alpha}\nabla_\theta\log\tilde{r}_{\theta^\star;\alpha}^{\intercal}].$$

*We also define*

$$\tilde{\mathcal{I}}_\alpha \triangleq \nabla_\theta^2\tilde{\mathcal{L}}_\alpha(\theta^\star) = \mathbb{E}_{q_{\sf d}}[\tilde{r}_{\theta^\star;\alpha}^{\alpha-1}((1-\alpha)\nabla_\theta\log\tilde{r}_{\theta^\star;\alpha}\nabla_\theta\log\tilde{r}_{\theta^\star;\alpha}^{\intercal} - \nabla_\theta^2\log\tilde{r}_{\theta^\star;\alpha})].$$

*Proof.* From Lemma B.4, we have

$$\partial_{\theta_i}\hat{\tilde{\mathcal{L}}}_\alpha(\theta) = \frac{1}{(1-\alpha)}\mathbb{E}_{\hat{q}_{\sf d}}[\partial_{\theta_i}\tilde{r}_{\theta;\alpha}^{\alpha-1}]$$

$$= -\mathbb{E}_{\hat{q}_{\sf d}}[\tilde{r}_{\theta;\alpha}^{\alpha-2}\,\partial_{\theta_i}\tilde{r}_{\theta;\alpha}]$$

$$= -\mathbb{E}_{\hat{q}_{\sf d}}[\tilde{r}_{\theta;\alpha}^{\alpha-1}(\psi_i - \mathbb{E}_{q_{\sf n}}[\psi_i\tilde{r}_{\theta;\alpha}^\alpha])].$$

From this derivative expression, the computation is straightforward. $\qquad\square$

**Corollary D.1** (GISO: derivatives).

$$\nabla_\theta\tilde{\mathcal{L}}_0(\theta) = \mathbb{E}_{q_{\sf d}}[-\tilde{r}_{\theta;0}^{-1}(\psi - \mathbb{E}_q[\psi])],$$

$$\nabla_\theta^2\tilde{\mathcal{L}}_0(\theta) = \mathbb{E}_{q_{\sf d}}[\tilde{r}_{\theta;0}^{-1}(\psi - \mathbb{E}_q[\psi])(\psi - \mathbb{E}_q[\psi])^{\intercal}],$$

$$\tilde{\mathcal{C}}_0 = \mathbb{E}_{q_{\sf d}}[\tilde{r}_{\theta^\star;0}^{-2}(\psi - \mathbb{E}_q[\psi])(\psi - \mathbb{E}_q[\psi])^{\intercal}],$$

$$\tilde{\mathcal{I}}_0 = \mathbb{E}_{q_{\sf d}}[\tilde{r}_{\theta^\star;0}^{-1}(\psi - \mathbb{E}_q[\psi])(\psi - \mathbb{E}_q[\psi])^{\intercal}].$$

*Proof.* From Proposition D.2,

$$\nabla_\theta \tilde{\mathcal{L}}_0(\theta) = \mathbb{E}_{q_d}[-\tilde{r}_{\theta;0}^{-1}\nabla_\theta \log \tilde{r}_{\theta;0}],$$
$$\nabla_\theta^2 \tilde{\mathcal{L}}_0(\theta) = \mathbb{E}_{q_d}[\tilde{r}_{\theta;0}^{-1}(\nabla_\theta \log \tilde{r}_{\theta;0}\nabla_\theta \log \tilde{r}_{\theta;0}^\mathsf{T} - \nabla_\theta^2 \log \tilde{r}_{\theta;0})],$$
$$\tilde{\mathcal{C}}_0 = \mathbb{E}_{q_d}[\tilde{r}_{\theta^\star;0}^{-2}\nabla_\theta \log \tilde{r}_{\theta^\star;0}\nabla_\theta \log \tilde{r}_{\theta^\star;0}^\mathsf{T}],$$
$$\tilde{\mathcal{I}}_0 = \mathbb{E}_{q_d}[\tilde{r}_{\theta^\star;0}^{-1}(\nabla_\theta \log \tilde{r}_{\theta^\star;0}\nabla_\theta \log \tilde{r}_{\theta^\star;0}^\mathsf{T} - \nabla_\theta^2 \log \tilde{r}_{\theta^\star;0})].$$

Since

$$\tilde{r}_{\theta;0} = \frac{\exp(\langle\theta, \psi - \mathbb{E}_q[\psi]\rangle)}{q(x)}e^{-\mathbb{E}_q[\log q]},$$
$$\nabla_\theta \log \tilde{r}_{\theta;0} = \psi - \mathbb{E}_q[\psi],$$
$$\nabla_\theta^2 \log \tilde{r}_{\theta;0} = 0,$$

the quantities can be further simplified as stated. □

**Corollary D.2** (MLE: derivatives).

$$\nabla_\theta \tilde{\mathcal{L}}_1(\theta) = \mathbb{E}_{q_d}[-\nabla_\theta \log p_\theta],$$
$$\nabla_\theta^2 \tilde{\mathcal{L}}_1(\theta) = \mathbb{E}_{q_d}[-\nabla_\theta^2 \log p_\theta],$$
$$\tilde{\mathcal{C}}_1 = \mathbb{E}_{q_d}[\nabla_\theta \log p_{\theta^\star}\nabla_\theta \log p_{\theta^\star}^\mathsf{T}],$$
$$\tilde{\mathcal{I}}_1 = \mathbb{E}_{q_d}[-\nabla_\theta^2 \log p_{\theta^\star}].$$

**Theorem 4.2** ($\alpha$-CentNCE: finite-sample guarantee). *Pick $\alpha \in \mathbb{R}$. Define*

$$(\rho_{\min}, \rho_{\max}) \triangleq \left(\inf_{x\in\mathcal{X},\theta\in\Theta} \tilde{\rho}_{\theta;\alpha}(x), \sup_{x\in\mathcal{X},\theta\in\Theta} \tilde{\rho}_{\theta;\alpha}(x)\right)$$

*and define the quantities in Eq. (5) for $f = f_\alpha$ accordingly. Let $\tilde{\rho}_{\theta^\star;\alpha}^\alpha(x) \triangleq \frac{(\frac{q_d(x)}{q_n(x)})^\alpha}{\mathbb{E}_{q_n}[(\frac{q_d}{q_n})^\alpha]}$, and let*

$$\lambda_{\min,d}^{\mathsf{cent}} \triangleq \lambda_{\min}(\mathbb{E}_{q_d}[(\psi - \mathbb{E}_{q_n}[\psi\tilde{\rho}_{\theta^\star;\alpha}^\alpha])(\psi - \mathbb{E}_{q_n}[\psi\tilde{\rho}_{\theta^\star;\alpha}^\alpha])^\mathsf{T}]),$$
$$\lambda_{\min,n}^{\mathsf{cent}} \triangleq \lambda_{\min}(\mathbb{E}_{q_n}[\psi\psi^\mathsf{T}\tilde{\rho}_{\theta^\star;\alpha}^\alpha] - \mathbb{E}_{q_n}[\psi\tilde{\rho}_{\theta^\star;\alpha}^\alpha]\mathbb{E}_{q_n}[\psi\tilde{\rho}_{\theta^\star;\alpha}^\alpha]^\mathsf{T}).$$

*Let $\hat{\theta}_{\alpha,n_d}^{\mathsf{cent},\mathcal{R}}$ be such that*

$$\hat{\theta}_{\alpha,n_d}^{\mathsf{cent},\mathcal{R}} \in \arg\min_{\theta\in\Theta}\Big\{\mathcal{L}_\alpha^{\mathsf{cent}}(\theta; \hat{q}_d, q_n) + \lambda_{n_d}\mathcal{R}(\theta)\Big\}$$

*for some $\lambda_{n_d} > 0$. Define $\psi_{\max,\alpha} \triangleq \psi_{\max} + \|\mathbb{E}_{q_n}[\psi\tilde{\rho}_{\theta^\star;\alpha}^\alpha]\|_{\max}$. Then, for any $\Delta > 0$ and $\delta \in (0,1)$, there exists a choice of $\lambda_{n_d}$ such that $\|\hat{\theta}_{f,n_d}^{\mathsf{cent},\mathcal{R}} - \theta^\star\|_2 \leq \Delta$ with probability $\geq 1 - \delta$, provided that*

$$n_d = \Omega\Bigg(\max\Bigg\{\frac{(B_{\mathsf{nce},f_\alpha,d}^{(1)})^2\gamma_{\mathcal{R};2}^2\gamma_{\mathcal{R}^*;\infty}^2\psi_{\max,\alpha}^2}{\Delta^2(b_{\mathsf{nce},f_\alpha,d}^{(2)})^2\{(1-\alpha)\lambda_{\min,d}^{\mathsf{cent}} + \alpha\lambda_{\min,n}^{\mathsf{cent}}\}^2},$$
$$\frac{\gamma_{1;2}^4\psi_{\max,\alpha}^4}{(\lambda_{\min,d}^{\mathsf{cent}})^2}\Bigg\}\log\frac{p^2}{\delta}\Bigg).$$

**Proposition D.3** (Vanishing gradient). *(cf. (Shah et al., 2021b, Proposition F.1).) Assume Assumption 4.1. For any $\delta \in (0,1)$, $\epsilon > 0$,*

$$\|\nabla_\theta \tilde{\mathcal{L}}_\alpha(\theta^\star)\|_{\max} \leq \epsilon$$

*with probability $\geq 1 - \delta$, if $n_d \geq \frac{2r_{\min,\alpha}^{2(\alpha-1)}(\psi_{\max} + \|\mathbb{E}_{q_n}[\psi\tilde{r}_{\theta^\star;\alpha}^\alpha]\|_{\max})^2}{\epsilon^2}\log\frac{2p}{\delta}$.*

*Proof.* Recall from Lemma D.2 that

$$\nabla_\theta \hat{\tilde{\mathcal{L}}}_\alpha(\theta) = \mathbb{E}_{\hat{q}_{\mathsf{d}}}[-\tilde{r}_{\theta;\alpha}^{\alpha-1}\nabla_\theta \log \tilde{r}_{\theta;\alpha}] = -\mathbb{E}_{\hat{q}_{\mathsf{d}}}[\psi \tilde{r}_{\theta;\alpha}^{\alpha-1}] + \mathbb{E}_{\hat{q}_{\mathsf{d}}}[\tilde{r}_{\theta;\alpha}^{\alpha-1}]\mathbb{E}_{q_{\mathsf{n}}}[\psi \tilde{r}_{\theta;\alpha}^{\alpha}],$$

and it is easy to check that

$$\mathbb{E}[\nabla_\theta \hat{\tilde{\mathcal{L}}}_\alpha(\theta^\star)] = \nabla_\theta \tilde{\mathcal{L}}_\alpha(\theta^\star) = -\mathbb{E}_{q_{\mathsf{d}}}[\psi \tilde{r}_{\theta^\star;\alpha}^{\alpha-1}] + \mathbb{E}_{q_{\mathsf{d}}}[\tilde{r}_{\theta^\star;\alpha}^{\alpha-1}]\mathbb{E}_{q_{\mathsf{n}}}[\psi \tilde{r}_{\theta^\star;\alpha}^{\alpha}] = 0.$$

Since $|\tilde{r}_{\theta^\star;\alpha}^{\alpha-1}(\psi_i(x) - \mathbb{E}_{q_{\mathsf{n}}}[\psi_i \tilde{r}_{\theta^\star;\alpha}^\alpha])| \leq r_{\min,\alpha}^{\alpha-1}(\psi_{\max} + \|\mathbb{E}_{q_{\mathsf{n}}}[\psi \tilde{r}_{\theta^\star;\alpha}^\alpha]\|_{\max})$, by Hoeffding's inequality, we have

$$\mathbb{P}(|\partial_{\theta_i}\hat{\tilde{\mathcal{L}}}_\alpha(\theta^\star)| \geq \epsilon) \leq 2\exp\left(-\frac{n_{\mathsf{d}}\epsilon^2}{2r_{\min,\alpha}^{2(\alpha-1)}(\psi_{\max} + \|\mathbb{E}_{q_{\mathsf{n}}}[\psi \tilde{r}_{\theta^\star;\alpha}^\alpha]\|_{\max})^2}\right) = \delta,$$

if $n_{\mathsf{d}} \geq \frac{2r_{\min,\alpha}^{2(\alpha-1)}(\psi_{\max} + \|\mathbb{E}_{q_{\mathsf{n}}}[\psi \tilde{r}_{\theta^\star;\alpha}^\alpha]\|_{\max})^2}{\epsilon^2}\log\frac{2}{\delta}$. By taking a union bound over $p$ different coordinates of $\theta$, we conclude the proof. $\qquad\square$

**Proposition D.4** (Restricted strong convexity). *(cf. (Shah et al., 2021a, Proposition E.1)) Under Assumption 4.1, we have*

$$\Delta_{\hat{\tilde{\mathcal{L}}}_\alpha}(\theta,\theta^\star) \geq \tilde{r}_{\max,\alpha}^{\alpha-1}\left\{\frac{1}{2}(1-\alpha)\lambda_{\min,\mathsf{d}}^{\mathsf{cent}} + \alpha\lambda_{\min,\mathsf{n}}^{\mathsf{cent}}\right\}\|\theta - \theta^\star\|_2^2,$$

*with probability $\geq 1 - \delta$, if $n_{\mathsf{d}} \geq \frac{8\gamma_{1;2}^4(\psi_{\max} + \|\mathbb{E}_{q_{\mathsf{n}}}[\psi \tilde{r}_{\theta^\star;\alpha}^\alpha]\|_{\max})^4}{(\lambda_{\min,\mathsf{d}}^{\mathsf{cent}})^2}\log\frac{2p^2}{\delta}$.*

*Proof.* By the intermediate value theorem, there exists $\xi \in \{t\theta + (1-t)\theta^\star : t \in [0,1]\}$ such that

$$\Delta_{\hat{\tilde{\mathcal{L}}}_\alpha}(\theta,\theta^\star) = \hat{\tilde{\mathcal{L}}}_\alpha(\theta) - \hat{\tilde{\mathcal{L}}}_\alpha(\theta^\star) - \langle\nabla_\theta\hat{\tilde{\mathcal{L}}}_\alpha(\theta^\star),\theta - \theta^\star\rangle$$
$$= \frac{1}{2}(\theta - \theta^\star)^\mathsf{T}\nabla_\theta^2\hat{\tilde{\mathcal{L}}}_\alpha(\xi)(\theta - \theta^\star).$$

Define $\overline{\overline{\psi}} \triangleq \mathbb{E}_{q_{\mathsf{n}}}[\psi \tilde{r}_{\theta;\alpha}^\alpha]$ and $\overline{\overline{\psi\psi^\mathsf{T}}} \triangleq \mathbb{E}_{q_{\mathsf{n}}}[\psi\psi^\mathsf{T}\tilde{r}_{\theta;\alpha}^\alpha]$ for shorthand notation. Here, note that $\xi$ depends on $\hat{q}_{\mathsf{d}}$. Recall from Lemma D.2 that

$$\nabla_\theta^2\hat{\tilde{\mathcal{L}}}_\alpha(\theta) = \mathbb{E}_{\hat{q}_{\mathsf{d}}}[\tilde{r}_{\theta;\alpha}^{\alpha-1}((1-\alpha)\nabla_\theta \log\tilde{r}_{\theta;\alpha}\nabla_\theta \log\tilde{r}_{\theta;\alpha}^\mathsf{T} - \nabla_\theta^2 \log\tilde{r}_{\theta;\alpha})]$$
$$= (1-\alpha)\mathbb{E}_{\hat{q}_{\mathsf{d}}}[\tilde{r}_{\theta;\alpha}^{\alpha-1}(\psi - \overline{\overline{\psi}})(\psi - \overline{\overline{\psi}})^\mathsf{T}] + \alpha\mathbb{E}_{\hat{q}_{\mathsf{d}}}[\tilde{r}_{\theta;\alpha}^{\alpha-1}](\overline{\overline{\psi\psi^\mathsf{T}}} - \overline{\overline{\psi}}\,\overline{\overline{\psi}}^\mathsf{T}).$$

Let $z \triangleq \langle\psi(x),\theta - \theta^\star\rangle$.

$$(\theta - \theta^\star)^\mathsf{T}\nabla_\theta^2\hat{\tilde{\mathcal{L}}}_\alpha(\xi)(\theta - \theta^\star)$$
$$= (1-\alpha)\mathbb{E}_{\hat{q}_{\mathsf{d}}}[\tilde{r}_{\xi;\alpha}^{\alpha-1}((\theta - \theta^\star)^\mathsf{T}(\psi - \overline{\overline{\psi}}))^2] + \alpha\mathbb{E}_{\hat{q}_{\mathsf{d}}}[\tilde{r}_{\xi;\alpha}^{\alpha-1}](\theta - \theta^\star)^\mathsf{T}(\overline{\overline{\psi\psi^\mathsf{T}}} - \overline{\overline{\psi}}\,\overline{\overline{\psi}}^\mathsf{T})(\theta - \theta^\star)$$
$$\geq (1-\alpha)\tilde{r}_{\max,\alpha}^{\alpha-1}\mathbb{E}_{\hat{q}_{\mathsf{d}}}[((\theta - \theta^\star)^\mathsf{T}(\psi - \overline{\overline{\psi}}))^2] + \alpha\tilde{r}_{\max,\alpha}^{\alpha-1}\lambda_{\min,\mathsf{n}}^{\mathsf{cent}}\|\theta - \theta^\star\|^2$$
$$= \tilde{r}_{\max,\alpha}^{\alpha-1}\{(1-\alpha)(\theta - \theta^\star)^\mathsf{T}\mathbb{E}_{\hat{q}_{\mathsf{d}}}[(\psi - \overline{\overline{\psi}})(\psi - \overline{\overline{\psi}})^\mathsf{T}](\theta - \theta^\star) + \alpha\lambda_{\min,\mathsf{n}}^{\mathsf{cent}}\|\theta - \theta^\star\|^2\}.$$

We can lower bound the first term as follows.

$$(\theta - \theta^\star)^\mathsf{T}\mathbb{E}_{\hat{q}_{\mathsf{d}}}[(\psi - \overline{\overline{\psi}})(\psi - \overline{\overline{\psi}})^\mathsf{T}](\theta - \theta^\star) \overset{(a)}{\geq} -\epsilon_2\|\theta - \theta^\star\|_1^2 + \lambda_{\min,\mathsf{d}}^{\mathsf{cent}}\|\theta - \theta^\star\|_2^2$$
$$\overset{(b)}{\geq} -\epsilon_2\gamma_{1;2}^2\|\theta - \theta^*\|_2^2 + \lambda_{\min,\mathsf{d}}^{\mathsf{cent}}\|\theta - \theta^\star\|_2^2$$
$$= \frac{1}{2}\lambda_{\min,\mathsf{d}}^{\mathsf{cent}}\|\theta - \theta^\star\|_2^2$$

with probability $\geq 1 - \delta$ if $n_{\mathsf{d}} \geq \frac{2(\psi_{\max} + \|\mathbb{E}_{q_{\mathsf{n}}}[\psi \tilde{r}^{\alpha}_{\theta^\star;\alpha}]\|_{\max})^4}{\epsilon_2^2} \log \frac{2p^2}{\delta}$ with $\epsilon_2 = \frac{\lambda^{\mathsf{cent}}_{\min,\mathsf{d}}}{2\gamma^2_{1;2}}$. Here, we apply Hoeffding's inequality similar to Lemma D.1 in $(a)$, and use the definition of $\gamma_{1;2}$ to bound $\|\theta - \theta^\star\|_1 \leq \gamma_{1;2}\|\theta - \theta^\star\|_2$ in $(b)$. Hence, with probability $\geq 1 - \delta$, we have

$$\Delta_{\hat{\tilde{\mathcal{L}}}_\alpha}(\theta, \theta^\star) \geq \tilde{r}^{\alpha-1}_{\max,\alpha}\left\{\frac{1}{2}(1-\alpha)\lambda^{\mathsf{cent}}_{\min,\mathsf{d}} + \alpha\lambda^{\mathsf{cent}}_{\min,\mathsf{n}}\right\}\|\theta - \theta^\star\|_2^2,$$

provided that $n_{\mathsf{d}} \geq \frac{8\gamma^4_{1;2}(\psi_{\max} + \|\mathbb{E}_{q_{\mathsf{n}}}[\psi\tilde{r}^{\alpha}_{\theta^\star;\alpha}]\|_{\max})^4}{(\lambda^{\mathsf{cent}}_{\min,\mathsf{d}})^2} \log \frac{2p^2}{\delta}$. $\qquad\square$

*Proof of Theorem 4.2.* First, note that

$$\mathcal{R}^*(\nabla_\theta \hat{\tilde{\mathcal{L}}}_\alpha(\theta^\star)) \leq \gamma_{\mathcal{R}^*;\infty}\|\nabla_\theta \hat{\tilde{\mathcal{L}}}_\alpha(\theta^\star)\|_{\max}$$

by definition of $\gamma_{\mathcal{R}^*;\infty}$. Then, by Proposition D.3, we have $\|\nabla_\theta \hat{\tilde{\mathcal{L}}}_\alpha(\theta^\star)\|_{\max} \leq \epsilon$ with probability $\geq 1 - \delta_1$, if

$$n_{\mathsf{d}} \geq \frac{2r^{2(\alpha-1)}_{\min,\alpha}(\psi_{\max} + \|\mathbb{E}_{q_{\mathsf{n}}}[\psi\tilde{r}^{\alpha}_{\theta^\star;\alpha}]\|_{\max})^2}{\epsilon^2} \log \frac{2p}{\delta_1}.$$

Given that this event occurs, $\mathcal{R}^*(\nabla_\theta \hat{\tilde{\mathcal{L}}}_\alpha(\theta^\star)) \leq \gamma_{\mathcal{R}^*;\infty}\epsilon$, and thus we set $\lambda_n \leftarrow 2\gamma_{\mathcal{R}^*;\infty}\epsilon$ to satisfy the first condition in Theorem 4.4.

Now, given $\lambda_n \geq 2\mathcal{R}^*(\nabla_\theta \hat{\tilde{\mathcal{L}}}_\alpha(\theta^\star))$, (Negahban et al., 2012, Lemma 1) implies that $\mathcal{R}(\hat{\theta}^{\mathsf{cent},\mathcal{R}}_{\alpha,n_{\mathsf{d}}} - \theta^\star) \leq 4\mathcal{R}(\theta^\star)$, i.e., $\hat{\theta}^{\mathsf{cent},\mathcal{R}}_{\alpha,n_{\mathsf{d}}} - \theta^\star \in 4\Theta$. Then, by Proposition D.2, we have

$$\Delta_{\hat{\mathcal{L}}^{\mathsf{nce}}_f}(\theta, \theta^\star) \geq \kappa\|\theta - \theta^\star\|_2^2$$

with probability $\geq 1 - \delta_2$ if $n_{\mathsf{d}} \geq \frac{8\gamma^4_{1;2}(\psi_{\max} + \|\mathbb{E}_{q_{\mathsf{n}}}[\psi\tilde{r}^{\alpha}_{\theta^\star;\alpha}]\|_{\max})^4}{(\lambda^{\mathsf{cent}}_{\min,\mathsf{d}})^2} \log \frac{2p^2}{\delta}$, where

$$\kappa = \tilde{r}^{\alpha-1}_{\max,\alpha}\left\{\frac{1}{2}(1-\alpha)\lambda^{\mathsf{cent}}_{\min,\mathsf{d}} + \alpha\lambda^{\mathsf{cent}}_{\min,\mathsf{n}}\right\} \geq \frac{1}{2}\tilde{r}^{\alpha-1}_{\max,\alpha}\{(1-\alpha)\lambda^{\mathsf{cent}}_{\min,\mathsf{d}} + \alpha\lambda^{\mathsf{cent}}_{\min,\mathsf{n}}\}.$$

Now, by taking a union bound with $\delta_1 = \delta_2 = \delta/2$, with probability $\geq 1 - \delta$, we have

$$\|\theta - \theta^\star\|_2 \leq \frac{3\lambda_n \gamma_{\mathcal{R};2}}{\kappa} = \frac{6\gamma_{\mathcal{R}^*;\infty}\gamma_{\mathcal{R};2}}{\kappa}\epsilon = \Delta$$

with $\epsilon \leftarrow \frac{\Delta\kappa}{6\gamma_{\mathcal{R}^*;\infty}\gamma_{\mathcal{R};2}}$, provided that

$$n_{\mathsf{r}} \geq \max\left\{\frac{72r^{2(\alpha-1)}_{\min,\alpha}(\psi_{\max} + \|\mathbb{E}_{q_{\mathsf{n}}}[\psi\tilde{r}^{\alpha}_{\theta^\star;\alpha}]\|_{\max})^2\gamma^2_{\mathcal{R};2}\gamma^2_{\mathcal{R}^*;\infty}}{\Delta^2\kappa^2} \log \frac{2p}{\delta}, \frac{8\gamma^4_{1;2}(\psi_{\max} + \|\mathbb{E}_{q_{\mathsf{n}}}[\psi\tilde{r}^{\alpha}_{\theta^\star;\alpha}]\|_{\max})^4}{(\lambda^{\mathsf{cent}}_{\min,\mathsf{d}})^2} \log \frac{4p^2}{\delta}\right)$$

$$\geq \max\left\{\frac{1152r^{2(\alpha-1)}_{\min,\alpha}(\psi_{\max} + \|\mathbb{E}_{q_{\mathsf{n}}}[\psi\tilde{r}^{\alpha}_{\theta^\star;\alpha}]\|_{\max})^2\gamma^2_{\mathcal{R};2}\gamma^2_{\mathcal{R}^*;\infty}}{\Delta^2\tilde{r}^{2(\alpha-1)}_{\max,\alpha}\{(1-\alpha)\lambda^{\mathsf{cent}}_{\min,\mathsf{d}} + \alpha\lambda^{\mathsf{cent}}_{\min,\mathsf{n}}\}^2} \log \frac{2p}{\delta}, \frac{8\gamma^4_{1;2}(\psi_{\max} + \|\mathbb{E}_{q_{\mathsf{n}}}[\psi\tilde{r}^{\alpha}_{\theta^\star;\alpha}]\|_{\max})^4}{(\lambda^{\mathsf{cent}}_{\min,\mathsf{d}})^2} \log \frac{4p^2}{\delta}\right)$$

$$= \Omega\left(\max\left\{\frac{r^{2(\alpha-1)}_{\min,\alpha}(\psi_{\max} + \|\mathbb{E}_{q_{\mathsf{n}}}[\psi\tilde{r}^{\alpha}_{\theta^\star;\alpha}]\|_{\max})^2\gamma^2_{\mathcal{R};2}\gamma^2_{\mathcal{R}^*;\infty}}{\Delta^2\tilde{r}^{2(\alpha-1)}_{\max,\alpha}\{(1-\alpha)\lambda^{\mathsf{cent}}_{\min,\mathsf{d}} + \alpha\lambda^{\mathsf{cent}}_{\min,\mathsf{n}}\}^2}, \frac{\gamma^4_{1;2}(\psi_{\max} + \|\mathbb{E}_{q_{\mathsf{n}}}[\psi\tilde{r}^{\alpha}_{\theta^\star;\alpha}]\|_{\max})^4}{(\lambda^{\mathsf{cent}}_{\min,\mathsf{d}})^2}\right\} \log \frac{p^2}{\delta}\right).$$

$\qquad\square$

### D.3. $f$-CondNCE

**Theorem 4.3** ($f$-CondNCE: finite-sample guarantee). *Pick a strictly convex function $f : \mathbb{R}_+ \to \mathbb{R}$. Define*

$$\rho_{\min} \triangleq \inf_{(x,y)\in\mathsf{supp}(q_{\mathsf{d}}(x)\pi(y|x)),\theta\in\Theta} \rho_\theta(x, y),$$

$$\rho_{\max} \triangleq \sup_{(x,y)\in\text{supp}(q_\mathsf{d}(x)\pi(y|x)),\theta\in\Theta} \rho_\theta(x,y).$$

*and define the quantities in Eq. (5) accordingly. Let*

$$\lambda_{\min,\mathsf{d}}^{\mathsf{cond}} \triangleq \lambda_{\min}(\mathbb{E}_{q_\mathsf{d}(x)\pi(y|x)}[(\psi(x)-\psi(y))(\psi(x)-\psi(y))^\mathsf{T}]).$$

*Let $\hat{\theta}_{f,n_\mathsf{d}}^{\mathsf{cond},\mathcal{R}}$ be such that*

$$\hat{\theta}_{f,n_\mathsf{d}}^{\mathsf{cond},\mathcal{R}} \in \arg\min_{\theta\in\Theta}\Big\{\mathcal{L}_f^{\mathsf{cond}}(\theta;\hat{q}_\mathsf{d},\hat{\pi}) + \lambda_{n_\mathsf{d}}\mathcal{R}(\theta)\Big\}$$

*for some $\lambda_{n_\mathsf{d}} > 0$. Then, for any $\Delta > 0$ and $\delta \in (0,1)$, there exists a choice of $\lambda_n$ such that $\|\hat{\theta}_{f,n_\mathsf{d}}^{\mathsf{cond},\mathcal{R}} - \theta^\star\|_2 \leq \Delta$ with probability $\geq 1 - \delta$, provided that*

$$n_\mathsf{d} = \Omega\bigg(\max\bigg\{\frac{(B_{\mathsf{cond},f,\mathsf{d}}^{(1)} + B_{\mathsf{cond},f,\mathsf{n}}^{(1)})^2\gamma_{\mathcal{R};2}^2\gamma_{\mathcal{R}^*;\infty}^2\psi_{\max}^2}{\Delta^2(b_{\mathsf{cond},f,\mathsf{d}}^{(2)} + b_{\mathsf{cond},f,\mathsf{n}}^{(2)})^2(\lambda_{\min}^{\mathsf{cond}})^2},$$
$$\frac{\gamma_{1;2}^4\psi_{\max}^4}{(\lambda_{\min}^{\mathsf{cond}})^2}\bigg\}\log\frac{p^2}{\delta}\bigg).$$

*Here, $b_{\mathsf{cond},f,\mathsf{r}}^{(2)}$ and $B_{\mathsf{cond},f,\mathsf{r}}^{(i)}$ are defined similar to Eq. (5), where the infimum and supremum are taken over $\big(\frac{\rho_{\min}}{\rho_{\max}}, \frac{\rho_{\max}}{\rho_{\min}}\big)$ in place of $(\rho_{\min}, \rho_{\max})$.*

**Proposition D.5** (Vanishing gradient)**.** *(cf. (Shah et al., 2021b, Proposition F.1).) Assume Assumption 4.1. For any $\delta \in (0,1)$, $\epsilon > 0$,*

$$\|\nabla_\theta\hat{\mathcal{L}}_f^{\mathsf{cond}}(\theta^\star)\|_{\max} \leq \epsilon$$

*with probability $\geq 1 - \delta$, if $n_\mathsf{d} \geq \frac{8\psi_{\max}^2(B_{\mathsf{nce},f,\mathsf{d}}^{(1)}+B_{\mathsf{nce},f,\mathsf{n}}^{(1)})^2}{\epsilon^2}\log\frac{2p}{\delta}$.*

*Proof.* Recall from Lemma B.8 that

$$\nabla_\theta\mathcal{L}_f^{\mathsf{cond}}(\theta) = \mathbb{E}_{q_\mathsf{d}(y)\pi(x|y)}[(\psi(x)-\psi(y))(-\xi_{\mathsf{nce},f,\mathsf{d}}^{(1)}(\rho_\theta^{-1}) + \xi_{\mathsf{nce},f,\mathsf{n}}^{(1)}(\rho_\theta))],$$

and it is easy to check that

$$\mathbb{E}[\nabla_\theta\hat{\mathcal{L}}_f^{\mathsf{cond}}(\theta^\star)] = \nabla_\theta\mathcal{L}_f^{\mathsf{cond}}(\theta^\star) = 0.$$

Since

$$|(\psi_i(y) - \psi_i(x))(-\xi_{\mathsf{nce},f,\mathsf{d}}^{(1)}(\rho_\theta^{-1}(x,y)) + \xi_{\mathsf{nce},f,\mathsf{n}}^{(1)}(\rho_\theta^{-1}(x,y)))|$$
$$\leq |(\psi_i(y) - \psi_i(x))\xi_{\mathsf{nce},f,\mathsf{d}}^{(1)}(\rho_\theta^{-1}(x,y))| + |(\psi_i(y) - \psi_i(x))\xi_{\mathsf{nce},f,\mathsf{n}}^{(1)}(\rho_\theta(x,y))|$$
$$\leq 2\psi_{\max}(B_{\mathsf{cond},f,\mathsf{d}}^{(1)} + B_{\mathsf{cond},f,\mathsf{n}}^{(2)}),$$

by Hoeffding's inequality and union bound, we have

$$\mathbb{P}(|\,\partial_{\theta_i}\hat{\mathcal{L}}_f^{\mathsf{nce}}(\theta^\star)| \geq \epsilon) \leq 2\exp\Big(-\frac{n_\mathsf{d}\epsilon^2}{8\psi_{\max}^2(B_{\mathsf{cond},f,\mathsf{d}}^{(1)} + B_{\mathsf{cond},f,\mathsf{n}}^{(1)})^2}\Big) = \delta,$$

if $n_\mathsf{d} \geq \frac{8\psi_{\max}^2(B_{\mathsf{cond},f,\mathsf{d}}^{(1)}+B_{\mathsf{cond},f,\mathsf{n}}^{(1)})^2}{\epsilon^2}\log\frac{2}{\delta}$. By taking a union bound over $p$ different coordinates of $\theta$, we conclude the proof. $\qquad\square$

**Proposition D.6** (Restricted strong convexity)**.** *(cf. (Shah et al., 2021a, Proposition E.1)) Under Assumption 4.1,*

$$\Delta_{\hat{\mathcal{L}}_f^{\mathsf{cond}}}(\theta, \theta^\star) \geq \frac{1}{2}(b_{\mathsf{cond},f,\mathsf{d}}^{(2)} + b_{\mathsf{cond},f,\mathsf{n}}^{(2)})\lambda_{\min}^{\mathsf{cond}}\|\theta - \theta^\star\|_2^2,$$

*with probability $\geq 1 - \delta$, if $n_\mathsf{d} \geq \frac{128\gamma_{1;2}^4\psi_{\max}^4}{(\lambda_{\min}^{\mathsf{cond}})^2}\log\frac{2p^2}{\delta}$.*

*Proof.* By the intermediate value theorem, there exists $\xi \in \{t\theta + (1-t)\theta^\star : t \in [0,1]\}$ such that

$$\Delta_{\hat{\mathcal{L}}_f^{\mathsf{cond}}}(\theta, \theta^\star) = \hat{\mathcal{L}}_f^{\mathsf{cond}}(\theta) - \hat{\mathcal{L}}_f^{\mathsf{cond}}(\theta^\star) - \langle \nabla_\theta \hat{\mathcal{L}}_f^{\mathsf{cond}}(\theta^\star), \theta - \theta^\star \rangle$$

$$= \frac{1}{2}(\theta - \theta^\star)^\mathsf{T} \nabla_\theta^2 \hat{\mathcal{L}}_f^{\mathsf{cond}}(\xi)(\theta - \theta^\star).$$

Here, note that $\xi$ depends on $\hat{q}_{\mathsf{d}}(x)\hat{\pi}(y|x)$. Let $z \triangleq \langle \psi(x) - \psi(y), \theta - \theta^\star \rangle$.

$$(\theta - \theta^\star)^\mathsf{T} \nabla_\theta^2 \hat{\mathcal{L}}_f^{\mathsf{cond}}(\xi)(\theta - \theta^\star) = \mathbb{E}_{\hat{q}_{\mathsf{d}}(x)\hat{\pi}(y|x)}[(\xi_{\mathsf{nce},f,\mathsf{d}}^{(2)}(\rho_\xi^{-1}) + \xi_{\mathsf{nce},f,\mathsf{n}}^{(2)}(\rho_\xi))z^2]$$

$$\geq (b_{\mathsf{cond},f,\mathsf{d}}^{(2)} + b_{\mathsf{cond},f,\mathsf{n}}^{(2)})\mathbb{E}_{\hat{q}_{\mathsf{d}}(x)\hat{\pi}(y|x)}[z^2]$$

$$= (b_{\mathsf{cond},f,\mathsf{d}}^{(2)} + b_{\mathsf{cond},f,\mathsf{n}}^{(2)})(\theta - \theta^\star)^\mathsf{T} \mathbb{E}_{\hat{q}_{\mathsf{d}}(x)\hat{\pi}(y|x)}[(\psi(x) - \psi(y))(\psi(x) - \psi(y))^\mathsf{T}](\theta - \theta^\star).$$

We can lower bound the quadratic form as follows. The first term can be lower bounded as

$$(\theta - \theta^\star)^\mathsf{T} \mathbb{E}_{\hat{q}_{\mathsf{d}}(x)\hat{\pi}(y|x)}[(\psi(x) - \psi(y))(\psi(x) - \psi(y))^\mathsf{T}](\theta - \theta^\star) \overset{(a)}{\geq} -\epsilon_2 \|\theta - \theta^\star\|_1^2 + \lambda_{\min}^{\mathsf{cond}}\|\theta - \theta^\star\|_2^2$$

$$\overset{(b)}{\geq} -\epsilon_2 \gamma_{1;2}^2 \|\theta - \theta^*\|_2^2 + \lambda_{\min}^{\mathsf{cond}}\|\theta - \theta^\star\|_2^2$$

$$= \frac{1}{2}\lambda_{\min}^{\mathsf{cond}}\|\theta - \theta^\star\|_2^2$$

with probability $\geq 1 - \delta$ if $n_{\mathsf{d}} \geq \frac{32\psi_{\max}^4}{\epsilon_2^2}\log\frac{2p^2}{\delta}$ with $\epsilon_2 = \frac{\lambda_{\min}^{\mathsf{cond}}}{2\gamma_{1;2}^2}$. Here, we apply Hoeffding's inequality as in Lemma D.1 in $(a)$, and use the definition of $\gamma_{1;2}$ to bound $\|\theta - \theta^\star\|_1 \leq \gamma_{1;2}\|\theta - \theta^\star\|_2$ in $(b)$.

Hence, with probability $\geq 1 - \delta$, we have

$$\Delta_{\hat{\mathcal{L}}_f^{\mathsf{cond}}}(\theta, \theta^\star) \geq \frac{1}{2}(b_{\mathsf{cond},f,\mathsf{d}}^{(2)} + b_{\mathsf{cond},f,\mathsf{n}}^{(2)})\lambda_{\min}^{\mathsf{cond}}\|\theta - \theta^\star\|_2^2,$$

if $n_{\mathsf{d}} \geq \frac{128\gamma_{1;2}^4\psi_{\max}^4}{(\lambda_{\min}^{\mathsf{cond}})^2}\log\frac{2p^2}{\delta}$. $\qquad\square$

*Proof of Theorem 4.3.* First, note that

$$\mathcal{R}^*(\nabla_\theta \hat{\mathcal{L}}_f^{\mathsf{cond}}(\theta^\star)) \leq \gamma_{\mathcal{R}^*;\infty}\|\nabla_\theta \hat{\mathcal{L}}_f^{\mathsf{cond}}(\theta^\star)\|_{\max}$$

by definition of $\gamma_{\mathcal{R}^*;\infty}$. Then, by Proposition D.5, we have $\|\nabla_\theta \hat{\mathcal{L}}_f^{\mathsf{cond}}(\theta^\star)\|_{\max} \leq \epsilon$ with probability $\geq 1 - \delta_1$, if

$$n_{\mathsf{d}} \geq \frac{8\psi_{\max}^2(B_{\mathsf{cond},f,\mathsf{d}}^{(1)} + B_{\mathsf{cond},f,\mathsf{n}}^{(1)})^2}{\epsilon^2}\log\frac{2p}{\delta_1}.$$

Given that this event occurs, $\mathcal{R}^*(\nabla_\theta \hat{\mathcal{L}}_f^{\mathsf{cond}}(\theta^\star)) \leq \gamma_{\mathcal{R}^*;\infty}\epsilon$, and thus we set $\lambda_n \leftarrow 2\gamma_{\mathcal{R}^*;\infty}\epsilon$ to satisfy the first condition in Theorem 4.4.

Now, given $\lambda_n \geq 2\mathcal{R}^*(\nabla_\theta \hat{\mathcal{L}}_f^{\mathsf{cond}}(\theta^\star))$, (Negahban et al., 2012, Lemma 1) implies that $\mathcal{R}(\hat{\theta}_{f,n_{\mathsf{d}}}^{\mathsf{cond},\mathcal{R}} - \theta^\star) \leq 4\mathcal{R}(\theta^\star)$, i.e., $\hat{\theta}_{f,n_{\mathsf{d}}}^{\mathsf{cond},\mathcal{R}} - \theta^\star \in 4\Theta$. Then, by Proposition D.6, we have

$$\Delta_{\hat{\mathcal{L}}_f^{\mathsf{cond}}}(\theta, \theta^\star) \geq \kappa\|\theta - \theta^\star\|_2^2$$

with probability $\geq 1 - \delta_2$ if $n_{\mathsf{d}} \geq \frac{128\gamma_{1;2}^4\psi_{\max}^4}{(\lambda_{\min}^{\mathsf{cond}})^2}\log\frac{2p^2}{\delta_2}$, where

$$\kappa = \frac{1}{2}(b_{\mathsf{cond},f,\mathsf{d}}^{(2)} + b_{\mathsf{cond},f,\mathsf{n}}^{(2)})\lambda_{\min}^{\mathsf{cond}}.$$

Now, by taking a union bound with $\delta_1 = \delta_2 = \delta/2$, with probability $\geq 1 - \delta$, we have

$$\|\theta - \theta^\star\|_2 \leq \frac{3\lambda_n\gamma_{\mathcal{R};2}}{\kappa} = \frac{6\gamma_{\mathcal{R}^*;\infty}\gamma_{\mathcal{R};2}}{\kappa}\epsilon = \Delta$$

with $\epsilon \leftarrow \frac{\Delta\kappa}{6\gamma_{\mathcal{R}^*;\infty}\gamma_{\mathcal{R};2}}$, provided that

$$
\begin{aligned}
n_{\mathsf{d}} &\geq \max\Big\{ \frac{288(B^{(1)}_{\mathsf{cond},f,\mathsf{d}} + B^{(1)}_{\mathsf{cond},f,\mathsf{n}})^2\gamma_{\mathcal{R};2}^2\gamma_{\mathcal{R}^*;\infty}^2\psi_{\max}^2}{\Delta^2\kappa^2} \log\frac{4p}{\delta}, \frac{128\gamma_{1;2}^4\psi_{\max}^4}{(\lambda_{\min}^{\mathsf{cond}})^2}\log\frac{4p^2}{\delta}\Big) \\
&= \max\Big\{ \frac{1152(B^{(1)}_{\mathsf{cond},f,\mathsf{d}} + B^{(1)}_{\mathsf{cond},f,\mathsf{n}})^2\gamma_{\mathcal{R};2}^2\gamma_{\mathcal{R}^*;\infty}^2\psi_{\max}^2}{\Delta^2(b^{(2)}_{\mathsf{cond},f,\mathsf{d}} + b^{(2)}_{\mathsf{cond},f,\mathsf{n}})^2(\lambda_{\min}^{\mathsf{cond}})^2} \log\frac{4p}{\delta}, \frac{128\gamma_{1;2}^4\psi_{\max}^4}{(\lambda_{\min}^{\mathsf{cond}})^2}\log\frac{4p^2}{\delta}\Big) \\
&= \Omega\Big(\max\Big\{ \frac{(B^{(1)}_{\mathsf{cond},f,\mathsf{d}} + B^{(1)}_{\mathsf{cond},f,\mathsf{n}})^2\gamma_{\mathcal{R};2}^2\gamma_{\mathcal{R}^*;\infty}^2\psi_{\max}^2}{\Delta^2(b^{(2)}_{\mathsf{cond},f,\mathsf{d}} + b^{(2)}_{\mathsf{cond},f,\mathsf{n}})^2(\lambda_{\min}^{\mathsf{cond}})^2}, \frac{\gamma_{1;2}^4\psi_{\max}^4}{(\lambda_{\min}^{\mathsf{cond}})^2}\Big\} \log\frac{p^2}{\delta}\Big). \qquad\square
\end{aligned}
$$

# E. Local NCE for Node-Wise-Sparse MRFs

In this section, we illustrate how one can construct a *local* version of the NCE principles introduced in the main text for, e.g., node-wise-sparse Markov random fields (MRFs). The notation herein follows the convention of (Ren et al., 2021) with modification. We use a boldface notation $\mathbf{x} = (x_1, \ldots, x_p) \in \mathcal{X} \subset \mathbb{R}^p$ for the purpose, and a regular-font variable $x$ is assumed to be scalar-valued. We assume that the exponential family distribution we consider is described as $\phi_{\boldsymbol{\theta}}(x) = \exp(\mathcal{E}(\mathbf{x}))$, where the (negative) energy function is

$$
\mathcal{E}(\mathbf{x}) \triangleq \sum_{I \in \mathcal{I}} \theta_I f_I(\mathbf{x}_I),
$$

where $\mathcal{F} \triangleq \{f_I : I \in \mathcal{I}\}$ for some $\mathcal{I} \subset 2^{[p]}$ is a collection of basis functions $f_I : \prod_{k \in I} \mathcal{X}_k \to \mathbb{R}$, each acting upon subsets of variables $\mathbf{x}_I$. Note that $\mathcal{F}$ is often called the sufficient statistics of the model.

To describe a conditional model, for each $i \in [p]$, define $\mathcal{I}_i \triangleq \{I \in \mathcal{I} : i \in I\}$. Then, we have

$$
p_{\boldsymbol{\theta}}(x_i|\mathbf{x}_{\backslash i}) \propto \phi_{\boldsymbol{\theta}}(x_i|\mathbf{x}_{\backslash i}) \triangleq \exp(\mathcal{E}_i(\mathbf{x})),
$$

where

$$
\mathcal{E}_i(\mathbf{x}) \triangleq \sum_{I \in \mathcal{I}_i} \theta_I f_I(\mathbf{x}_I).
$$

We remark that the pseudo-likelihood estimator of Besag (1975) is defined as

$$
\hat{\boldsymbol{\theta}}_i \triangleq \arg\min_{\boldsymbol{\theta}_i} \sum_{n=1}^{n_{\mathsf{d}}} \log\frac{1}{p_\theta(x_i^{(n)}|\mathbf{x}_{\backslash i}^{(n)})},
$$

where $\boldsymbol{\theta}_i$ is a collection of all parameters that affect the node conditional model $\phi_{\boldsymbol{\theta}}(x_i|\mathbf{x}_{\backslash i})$ among all parameters $\boldsymbol{\theta}$.

For $n$-th sample $\mathbf{x}^{(n)}$, let $x_j^{(n)}$ denote the $j$-th coordinate of $\mathbf{x}^{(n)}$. To apply the $f$-NCE principle, define the density ratio model

$$
\rho_{\boldsymbol{\theta}}(x_i|\mathbf{x}_{\backslash i}) \triangleq \frac{\phi_{\boldsymbol{\theta}}(x_i|\mathbf{x}_{\backslash i})}{\nu q_{\mathsf{n}}(x_i)}
$$

for a choice of reference distribution $q_{\mathsf{n}}(x)$. For each node $i \in [p]$, we can derive the *local* $f$-NCE objective as

$$
\begin{aligned}
&\mathbb{E}_{q_{\mathsf{d}}(\mathbf{x}_{\backslash i})}[\mathcal{L}_f^{\mathsf{nce}}(\phi_{\boldsymbol{\theta}}(x_i|\mathbf{x}_{\backslash i}); q_{\mathsf{d}}(x_i|\mathbf{x}_{\backslash i}), q_{\mathsf{n}}(x_i))] \\
&= \mathbb{E}_{q_{\mathsf{d}}(\mathbf{x}_{\backslash i})q_{\mathsf{n}}(x_i)}\Big[\Delta_f\Big(\frac{q_{\mathsf{d}}(x_i|\mathbf{x}_{\backslash i})}{\nu q_{\mathsf{n}}(x_i)}, \frac{\phi_{\boldsymbol{\theta}}(x_i|\mathbf{x}_{\backslash i})}{\nu q_{\mathsf{n}}(x_i)}\Big) - f\Big(\frac{q_{\mathsf{d}}(x_i|\mathbf{x}_{\backslash i})}{\nu q_{\mathsf{n}}(x_i)}\Big)\Big] \\
&= -\frac{1}{\nu}\mathbb{E}_{q_{\mathsf{d}}(\mathbf{x})}[f'(\rho_{\boldsymbol{\theta}}(x_i|\mathbf{x}_{\backslash i}))] + \mathbb{E}_{q_{\mathsf{d}}(\mathbf{x}_{\backslash i})q_{\mathsf{n}}(x_i)}[\rho_{\boldsymbol{\theta}}(x_i|\mathbf{x}_{\backslash i})f'(\rho_{\boldsymbol{\theta}}(x_i|\mathbf{x}_{\backslash i})) - f(\rho_{\boldsymbol{\theta}}(x_i|\mathbf{x}_{\backslash i}))].
\end{aligned}
\tag{13}
$$

In a similar manner, one can derive the local $\alpha$-CentNCE, which recovers pseudo-likelihood (Besag, 1975) for $\alpha = 1$ and GISO (Vuffray et al., 2016; 2021; Shah et al., 2021a) and ISODUS (Ren et al., 2021) for $\alpha = 0$, respectively. We note that Ren et al. (2021) justified ISODUS only from the stationarity of the objective function at the optimal parameter, while the connection established here between these interactive screening objectives (ISO) (i.e., GISO and ISODUS) to NCE provides a natural theoretical justification.

# F. Optimization Complexity

## F.1. Convexity

**Proposition F.1** ($f$-NCE: convexity). *Let $g_f(\rho) \triangleq -(\rho f'''(\rho) + f''(\rho))$. If $f''(\rho) \geq g_f(\rho) \geq 0$, then $\theta \to \hat{\mathcal{L}}_f^{\mathsf{nce}}(\theta)$ is convex. In particular, $\theta \to \hat{\mathcal{L}}_f^{\mathsf{nce}}(\theta)$ is convex for $f_{\log}$ and $f_\alpha$ for $\alpha \in [0,1]$.*

*Proof.* By Lemma B.3, we have

$$\nabla_\theta^2 \hat{\mathcal{L}}_f^{\mathsf{nce}}(\theta) = \frac{1}{\nu} \mathbb{E}_{\hat{q}_\mathsf{d}}[\psi\psi^\mathsf{T} \xi_{\mathsf{nce},f,\mathsf{d}}^{(2)}(\rho_\theta)] + \mathbb{E}_{\hat{q}_\mathsf{n}}[\psi\psi^\mathsf{T} \xi_{\mathsf{nce},f,\mathsf{n}}^{(2)}(\rho)].$$

Hence, if $g_f(\rho) \geq 0$ and $f''(\rho) - g_f(\rho) \geq 0$, then $\nabla_\theta^2 \hat{\mathcal{L}}_{\mathsf{nce}}(\theta)$ is a nonnegative combination of two positive definite matrices, and so must be positive semidefinite.

It remains to show that the condition holds for all $f$'s in Table 1. For the asymmetric power score $f_\alpha(\rho) = \rho^\alpha$ with $0 < \alpha < 1$, first, it is easy to check that $\rho \mapsto f_\alpha(\rho)$ is convex.

$$f_\alpha''(\rho) = \rho^{\alpha-2},$$
$$g_\alpha(\rho) = -(\rho f_\alpha'''(\rho) + f_\alpha''(\rho)) = (1-\alpha)\rho^{\alpha-2}.$$

Since $g_\alpha(\rho) \geq 0$ and $f_\alpha''(\rho) - g_\alpha(\rho) = \alpha\rho^{\alpha-2} \geq 0$, $\theta \to \hat{\mathcal{L}}_{\mathsf{nce}}(\theta)$ is convex by Lemma F.1. Note that the same calculation holds for $\alpha \in \{0,1\}$. $\square$

A counter example of convex functions $f$ which do not result in convex objectives is $f_\alpha(\rho)$ for $\alpha \notin [0,1]$. For $f$-NCE, while $f_\alpha''(\rho) = \rho^{\alpha-2} \geq 0$ for any $\alpha$, $g_{f_\alpha}(\rho) = (1-\alpha)\rho^{\alpha-2} < 0$ for $\alpha > 1$ and $\alpha \neq 2$ and $f_\alpha''(\rho) - g_{f_\alpha}(\rho) = \alpha\rho^{\alpha-2} < 0$ for $\alpha < 0$. For $\alpha = 2$, $g_{f_\alpha}(\rho) = -1 < 0$.

**Proposition F.2** (CentNCE: convexity). *For $\alpha \in [0,1]$, $\theta \mapsto \mathcal{L}_\alpha^{\mathsf{cent}}(\theta; q_\mathsf{d}, q_\mathsf{n})$ is convex.*

*Proof.* For $\alpha \in (0,1)$, note that we can write

$$\theta \mapsto \log \tilde{\mathcal{L}}_\alpha(\theta; q_\mathsf{d}, q_\mathsf{n}) = -\log(1-\alpha) + \log \mathbb{E}_{q_\mathsf{d}}[\rho_\theta^{\alpha-1}(x)] + \frac{1-\alpha}{\alpha} \log \mathbb{E}_{q_\mathsf{n}}[\rho_\theta^\alpha(x)].$$

Here, the second and third terms can be understood as LogSumExp operations applied on the linear function $\theta \mapsto \log \rho_\theta(x)$, and the resulting function becomes also convex. For $\alpha \in \{0,1\}$, the MLE and GISO objectives are well-known to be convex. $\square$

The proof of the following proposition is similar as above, and we thus omit the proof.

**Proposition F.3** ($f$-CondNCE: convexity). *If $g_f(\rho^{-1}) + \rho^3(f''(\rho) - g_f(\rho)) \geq 0$, then $\theta \to \hat{\mathcal{L}}_f^{\mathsf{cond}}(\theta)$ is convex. In particular, $\theta \to \hat{\mathcal{L}}_f^{\mathsf{cond}}(\theta)$ is convex for $f_{\log}$ and $f_\alpha$ for $\alpha \in [0,1]$.*

## F.2. Smoothness

Under the boundedness assumption, we can show that $f$-NCE objective function is smooth with probability 1.

**Proposition F.4** (Smoothness). *(cf. (Shah et al., 2021b, Proposition B.1).) Assume Assumption 4.1. $\theta \mapsto \hat{\mathcal{L}}_f^{\mathsf{nce}}(\theta)$ is a smooth function with smoothness constant*

$$p\psi_{\max}^2 \left( \frac{B_{\mathsf{nce},f,\mathsf{d}}^{(2)}}{\nu} + B_{\mathsf{nce},f,\mathsf{n}}^{(2)} \right).$$

*Proof.* Recall from Lemma B.3 that

$$\nabla_\theta^2 \hat{\mathcal{L}}_f^{\mathsf{nce}}(\theta) = \frac{1}{\nu} \mathbb{E}_{\hat{q}_\mathsf{d}}[\psi\psi^\mathsf{T} \xi_{\mathsf{nce},f,\mathsf{d}}^{(2)}(\rho_\theta)] + \mathbb{E}_{\hat{q}_\mathsf{n}}[\psi\psi^\mathsf{T} \xi_{\mathsf{nce},f,\mathsf{n}}^{(2)}(\rho)].$$

By Geršgorin's theorem ([Horn & Johnson](), [2012](), Theorem 6.1.1), the largest eigenvalue of a matrix is upper bounded by the largest absolute row sum or column sum. Therefore, we have

$$\lambda_{\max}(\nabla_\theta^2 \hat{\mathcal{L}}_f^{\mathsf{nce}}(\theta)) \leq \max_j \sum_i |\partial_{\theta_i \theta_j} \hat{\mathcal{L}}_f^{\mathsf{nce}}(\theta)|$$

$$\leq \max_j \sum_i \frac{1}{\nu} |\mathbb{E}_{\hat{q}_d}[\psi_i \psi_j \xi_{\mathsf{nce},f,\mathsf{d}}^{(2)}(\rho_\theta)]| + |\mathbb{E}_{\hat{q}_n}[\psi_i \psi_j \xi_{\mathsf{nce},f,\mathsf{n}}^{(2)}(\rho)]|$$

$$\leq \max_j \psi_{\max}^2 \sum_i \Big( \frac{1}{\nu} \sup_{x,\theta} \xi_{\mathsf{nce},f,\mathsf{d}}^{(2)}(\rho_\theta) + \sup_{x,\theta} \xi_{\mathsf{nce},f,\mathsf{n}}^{(2)}(\rho) \Big)$$

$$\leq p\psi_{\max}^2 \Big( \frac{B_{\mathsf{nce},f,\mathsf{d}}^{(2)}}{\nu} + B_{\mathsf{nce},f,\mathsf{n}}^{(2)} \Big). \qquad \square$$

We note that [Shah et al.]() ([2021b](), Lemma 3.1) shows that the projected gradient descent algorithm returns an $\epsilon$-optimal solution for GlobalGISO in polynomial optimization complexity, based on the similarly established smoothness of GlobalGISO. We can establish a similar optimization complexity guarantee, but we omit the statement.

## G. Experiments

In this section, we present a preliminary empirical evaluation of a selected set of estimators on a synthetic data, following a setting in ([Shah et al.](), [2023](), Section 5.1). We consider a unnormalized exponential family model

$$\phi_\theta(x) \triangleq \exp(x^\mathsf{T} \theta x),$$

where $\theta \in \mathbb{R}^{p \times p}$ for $x \in [-1, 1]^p$. The data generating distribution is chosen as the model with $\theta = \theta^\star$ defined as

$$\Theta_{ij}^\star \triangleq \begin{cases} \frac{1}{\sqrt{p}} & \text{if } i = 1, \text{ or } j = 1, \text{ or } i = j, \\ 0 & \text{otherwise.} \end{cases}$$

The samples were generated by brute-force sampling by discretizing each axis by 100 bins. We generated $N = 10^5$ samples for $p \in \{11, 13, 15, 17, 19\}$ and computed the estimates for each estimator with varying sample size $\{0.04N, 0.08N, \ldots, 0.64N\}$. We repeated the experiments with random subsamples for 5 times for each configuration.

Assuming the parameter space $\Theta$ is bounded under the Frobenius norm, we consider NCE estimators regularized by the Frobenius norm and optimized via gradient descent. We used a regularization weight $\lambda_n = 10^{-2}$ and a learning rate $\eta = 0.1$ across all settings, except for the $f_{\log}$-NCE estimator, where we used $\eta = 1.0$. Each optimization was run for 1000 gradient steps. As shown in Figure 2, the selected estimators exhibit an empirical convergence rate of $n^{-1/2}$. However, we observed that the $f_1$-NCE estimator (asymmetric log NCE; see Table 1) and the CNCE estimator did not display convergent behavior, despite the theoretical guarantees available for this example. This discrepancy highlights the need for further investigation into the empirical behavior of various estimators, particularly in high-dimensional settings.

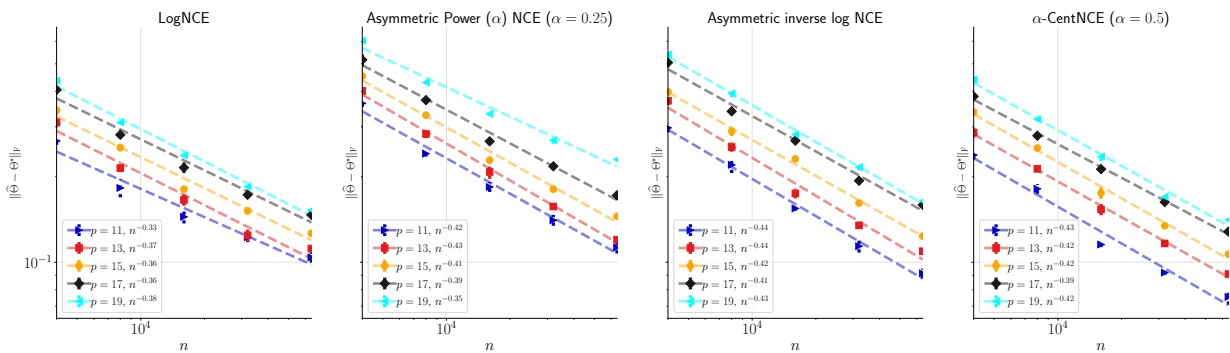

*Figure 2.* Convergence rate of different NCE estimators.

