# OpenReview forum: "A Unified View on Learning Unnormalized Distributions via Noise-Contrastive Estimation"
_ICML.cc/2025/Conference — ICML 2025 poster_

### Official Review · Reviewer_SNf8 · 2025-03-07

**Overall Recommendation:** 4

**Summary:**

In this work the authors present a unification of noise contrastive estimation (NCE) losses. In particular they consider a class of risks based on optimizing the density ratio of the model density compared to a known noise density (canonically a uniform distribution on a set known to contain the support of the true distribution) with a convex function in a Bregman-Divergence-style loss. From here they consider "$\alpha$-centered NCE", where the risk is normalized in some sense, and "$f$-conditional NCE" where the noise density is based on the training samples.

For these settings they prove that the population version of these losses will indeed work in a realizable modeling scenario (Prop 1.1, Prop 2.1, Prop 2.2). I addition they show that flexible forms of these losses using the correct loss parameters includes several well-known NCE losses, showing that the framework presented covers useful cases (Theorem 3.1, 3.2) and follow up with asymptotic and finite sample analysis (middle of section 3 and onwards).

## Update after rebuttal:

Nothing to update, I keep my score.

**Claims And Evidence:**

This is a purely theory paper so this isn't so applicable.

**Essential References Not Discussed:**

None that I am aware of.

**Experimental Designs Or Analyses:**

N/A

**Methods And Evaluation Criteria:**

N/A

**Other Comments Or Suggestions:**

None

**Other Strengths And Weaknesses:**

Overall the paper seems very strong to me. It is a natural sort of paper which generalizes many previous results and expands upon them.

**Questions For Authors:**

None

**Relation To Broader Scientific Literature:**

This does indeed seem to be a nice framework covering many NCE estimators with have previously been explored and includes some nice general results on the behavior of these. As stated in the intro, this covers MLE, MC0MLE, Global GISO, along with others.

**Theoretical Claims:**

I have not checked the proofs carefully, but scanned them, and nothing immediately stands out as being problematic.

---

> ### Author Rebuttal · Authors · 2025-04-01
>
> We thank the reviewer for their effort in reviewing our paper. We will incorporate all the comments from the reviews and revise our manuscript accordingly.

---

### Official Review · Reviewer_46T4 · 2025-03-11

**Overall Recommendation:** 4

**Summary:**

This paper provides a unified perspective on various estimators for learning unnormalized distributions (also known as energy-based models) using Noise-Contrastive Estimation (NCE). Specifically, they introduces $\alpha$-Centered NCE ($\alpha$-CentNCE) and f-Conditional NCE (f-CondNCE) as generalized versions of traditional NCE.

Building on this, their contributions include
1. they show that estimators such as Maximum Likelihood Estimation (MLE), Monte Carlo MLE (MC-MLE), and GlobalGISO can be viewed as special cases of $\alpha$-CentNCE.
2. f-Conditional NCE (f-CondNCE) generalizes previous conditional noise-contrastive estimation methods and clarifies their relationship with score matching.
3. they obtain finite-sample convergence rates for different estimators under regularity assumptions, which are novel for most NCE-based estimators.
4. they provide conditions under which these estimators achieve the parametric rate of convergence.

The paper is mainly a theoretical paper.

**Claims And Evidence:**

yes

**Essential References Not Discussed:**

I am not an expert in this area but to my best knowledge, the authors tried their best to relate works that are essential to understanding their key contributions.

**Experimental Designs Or Analyses:**

The paper has no experiments. It is a pure theoretical paper.

**Methods And Evaluation Criteria:**

This paper is not a methodology focusing paper but rather a theoretical paper. Their proposed framework of $\alpha$-Centered NCE ($\alpha$-CentNCE) and f-Conditional NCE (f-CondNCE) are sound with supporting theoretical results.

**Other Comments Or Suggestions:**

NA

**Other Strengths And Weaknesses:**

Strengths:
1. They paper theoretically unifies learning methods for unnormalized models including different estimation techniques (MLE, MC-MLE, Score Matching, etc.) while these estimators have only been developed independently, often with limited understanding of their connections until this paper.
2. It also introduces f-Conditional NCE (f-CondNCE), which extends conditional NCE and clarifies its theoretical behavior. Importantly, f-CondNCE corrects a misleading interpretation in prior work (Ceylan & Gutmann, 2018), showing that it does NOT converge to Score Matching in small-noise regimes.
3. The paper establishes finite-sample convergence rates for a broad class of NCE-based estimators, including $\alpha$-CentNCE and f-CondNCE. Moreover, they showed that most of these estimators achieve a parametric convergence rate of O(n^{-1/2}), which is the best possible rate in standard statistical settings.

Weakness:
1. The paper suggests that $\alpha$-CentNCE offers advantages over traditional Noise-Contrastive Estimation (NCE) in certain cases, but it does not fully formalize when and why $\alpha$-CentNCE is strictly better (while I only believe this is more a future direction rather than being completely solved in this one paper). For example, the paper proposes that centering may reduce variance but does not formally prove how variance changes as a function of $\alpha$. Studying this would provide guidance on how to pick $\alpha$ in practice. I would assume it involves some bias-variance tradeoff analysis.
2. The paper proves that f-CondNCE suffers from variance explosion in the small-noise limit, debunking prior claims, while it leaves an open theoretical problem: Can a modified version of f-CondNCE control variance growth while still approximating Score Matching?
3. The paper mentions that noise selection is crucial for the performance of NCE-based methods. However, the paper does not derive any results on optimal noise selection or analyze how different choices of $q_n(x)$ affect statistical efficiency.
4. I am not sure if these are standard assumptions in this area but The paper assumes that the sufficient statistic $\psi(x)$ is bounded while in commonly used models like Gaussian models $\psi(x)=x$ and log-linear models where $\psi(x)=\log(x)$, they are not. Also, the paper assumes that the noise distribution $q_n(x)$ is fixed and independent of the learned model while I believe in practice, this noise distribution should be tuned and selected dependent on the data.

**Questions For Authors:**

Please see the four questions listed in "Other Strengths And Weaknesses".

**Relation To Broader Scientific Literature:**

The problem of estimating unnormalized probability distributions that this paper focuses on is fundamental in energy-based models (EBMs) and appears in various fields, including generative modeling, density estimation, and graphical models. There exist many standard estimators including Maximum Likelihood Estimation (MLE), Monte Carlo MLE (MC-MLE), and GlobalGISO. For them, the paper reframes these seemingly different estimators under a unified NCE-based framework, showing that various inference methods can be viewed as special cases of α-Centered NCE (α-CentNCE). Moreover, the paper also connects to the score matching (SM), introduced by Hyvärinen (2005), provides an alternative estimation principle that avoids the computation of partition functions. Previous research (Ceylan & Gutmann, 2018) suggested that f-CondNCE approximates Score Matching in the limit as noise approaches zero while this paper challenges this claim, showing that f-CondNCE does NOT converge to Score Matching but instead exhibits diverging variance in the small-noise regime.

**Theoretical Claims:**

I haven't been able to check every detail of the proof but the proofs of their main theorems 4.1-4.3 are correct to me.

---

> ### Author Rebuttal · Authors · 2025-04-01
>
> We appreciate the reviewer’s careful assessment of our manuscript and their constructive feedback. Below, we respond to the points raised under *Weaknesses* to clarify our contributions. In summary, we believe that all the issues raised are important and warrant separate, dedicated future investigations.
>
> * As pointed out by the reviewer, studying the $\alpha$-CentNCE and its advantage over the NCE counterpart $f_\alpha$-NCE is a valuable direction for future work. We leave this for subsequent study.
>
> * As noted in Remark 3.3 and Remark 4.3, $f$-CondNCE suffers from variance explosion. However, our finite-sample analysis assumes that a single conditional sample $y_i$ is drawn for each data point $x_i$ (see line 201). As alluded to in Remark 3.4, this variance may be controlled by using a large number of conditional samples to reduce statistical noise. This reflects a fundamental trade-off between computational efficiency and statistical accuracy. A more thorough analysis of this trade-off is left for future work.
>
> * Choosing an optimal noise distribution for NCE is an important but open problem in the literature; see, for example, the recent study by Chehab et al. (UAI 2022). We will add a paragraph discussing this point to clarify the current understanding in the literature.
>
> * As the reviewer noted, we assume bounded exponential family distributions throughout. This assumption is fairly standard in the theoretical analysis of such estimators to enable tractable analysis; see, for example, (Shah et al., 2023). One possible justification is that many real-world distributions have bounded statistics and can thus be modeled by bounded exponential family distributions. That said, extending the analysis to unbounded distributions is of theoretical interest and also important for handling heavy-tailed data. In Section 5, we remark on the technical challenges involved in such an extension under `Beyond Bounded Exponential Families`. We leave this theoretical direction for future work.
>
> * Lastly, as the reviewer pointed out, for NCE-type arguments to work well in practice, using a data-adaptive noise distribution is crucial, as a mismatch between the noise and data distributions can significantly degrade performance. A theoretical investigation into data-adaptive choices of noise distributions is another compelling direction for future research.

---

### Official Review · Reviewer_H2zi · 2025-03-14

**Overall Recommendation:** 3

**Summary:**

This paper presents a unified framework for learning unnormalized distributions through noise-contrastive estimation (NCE), introducing two variants: alpha-CentNCE and f-CondNCE. It demonstrates that alpha-CentNCE generalizes existing methods like MLE, MC-MLE, and GlobalGISO, while f-CondNCE reveals limitations in prior connections to score matching, showing diverging variance in small-noise regimes. The analysis provides novel finite-sample convergence guarantees for exponential families, establishing theoretical foundations for several NCE-based estimators previously studied in isolated contexts.

**Claims And Evidence:**

The claims are supported by explicit connections to established methods like MLE and GlobalGISO via Theorem 3.1. The analysis leverages rigorous optimization principles (Bregman divergences) and builds convergence guarantees for exponential families using prior frameworks (Shah et al., 2023), offering a structured unification of disparate estimators through NCE variants.

**Essential References Not Discussed:**

No

**Experimental Designs Or Analyses:**

This is a theoretical paper, so no experiments need to be reviewed.

**Methods And Evaluation Criteria:**

The proposed methods are well-aligned with the problem of learning unnormalized distributions. By showing that alpha-CentNCE recovers MLE (Fisher, 1922), MC-MLE (Geyer, 1994), and GlobalGISO (Shah et al., 2023) as special cases (Theorem 3.1), the authors demonstrate consistency with established estimators for exponential families. Similarly, connecting f-CondNCE to pseudo-likelihood (Besag, 1975) and ISO (Vuffray et al., 2016) aligns with prior theoretical frameworks for Markov random fields. The use of Bregman divergences further roots the approach in foundational optimization principles, ensuring compatibility with existing density ratio estimation literature (Sugiyama et al., 2012). These connections validate the methods’ coherence with both classical and modern paradigms.

**Other Comments Or Suggestions:**

typo:
-  On page 1, in the right column, line 4: there is a double "is."

**Other Strengths And Weaknesses:**

This paper is not well-written and is somewhat difficult to follow.

**Questions For Authors:**

Your unified framework elegantly connects existing estimators (e.g., MLE, pseudo-likelihood, ISO) via alpha-CentNCE, but how generalizable is this unification to non-exponential family models? For instance, do the finite-sample convergence guarantees for alpha-CentNCE extend to broader classes of unnormalized distributions, such as energy-based models with deep parameterizations, or are they inherently limited to bounded exponential families?

Additionally, while your analysis reveals divergence in f-CondNCE’s variance under vanishing noise, what practical guidance does this imply for choosing noise distributions in real-world applications?

**Relation To Broader Scientific Literature:**

The paper proposes a unified NCE-based framework that connects disparate estimators (e.g., MLE, pseudo likelihood, GlobalGISO) as special cases of alpha-CentNCE or localized NCE, resolving their fragmented theoretical treatment. This view aligns with and generalizes prior results, such as Shah et al. (2023)’s GlobalGISO analysis and Besag (1975)’s pseudo likelihood, while clarifying their implicit ties to contrastive learning. Building on this framework, the authors derive new results, including finite-sample convergence rates for NCE estimators and a counterexample showing f-CondNCE’s variance diverges in small-noise regimes—contradicting earlier claims of equivalence to score matching. The synthesis enables systematic extensions of classical methods under a single theoretical lens.

**Theoretical Claims:**

I’ve taken a brief look at the proof,and it seems fine for now.

---

> ### Author Rebuttal · Authors · 2025-04-01
>
> We thank the reviewer for their effort in reviewing our manuscript and for the valuable feedback. Below, we provide responses to the questions.
>
> * We highlight that the discussion in Section 3 does not assume exponential family distributions, except in the comparison with GlobalGISO, which is specifically designed for exponential family distributions. (We remark that GlobalGISO requires the underlying model to be an exponential family distribution in order to assume that a certain statistic is analytically computable.) However, as pointed out by the reviewer, our current asymptotic and finite-sample analysis is only applicable to bounded exponential family distributions. In Section 5, we discuss the technical challenges of extending the results to unbounded distributions under `Beyond Bounded Exponential Families`. Analyzing this type of estimator for energy-based models beyond exponential family distributions is an important but open problem.
>
> * While our Theorem 3.4 reveals the asymptotic behavior of the empirical conditional NCE objective, we currently do not provide practical guidance on the choice of the noise distribution. As we mention in Remark 3.4, the statistical noise in the empirical conditional NCE objective can potentially be controlled by using a large number of *slicing vectors* (which essentially corresponds to samples from the conditional distribution). We believe this direction merits a separate investigation and leave it for future work.

---

### Official Review · Reviewer_7ihf · 2025-03-23

**Overall Recommendation:** 3

**Summary:**

The paper provides a unified perspective on noise-contrastive estimation (NCE) methods for learning unnormalized distributions, integrating several previously separate approaches under a common framework. It introduces two new variants: $\alpha$-centered NCE ($\alpha$-CentNCE) and $f$-conditional NCE ($f$-CondNCE), which generalize prior estimators and clarify their relationships. The analysis also revisits the theoretical properties of $f$-CondNCE, challenging prior claims about its connection to score matching.

**Claims And Evidence:**

Theorems and propositions accompany the claims. I read through the remarks and comparisons but did not check the detailed proofs.

**Essential References Not Discussed:**

I'm not an expert on this specific line of research, but the current literature discussion looks good to me.

**Experimental Designs Or Analyses:**

The paper does not contain experimental/numerical evaluations.

**Methods And Evaluation Criteria:**

Formulating the problem as an instance of Bregman divergence minimization makes sense, as it appears to align with the existing approaches in the literature.

**Other Comments Or Suggestions:**

- Formulas and tables often do not appear on the same page where they are referenced. Maybe this is a property of the paper template, but having them on the same page would be nice.
- Adding proof sketches to help readers understand the underlying connections among the theorems/claims would be beneficial.
- On page 1, right panel, line 13 reads, "The motivation ... is (is) to ..."

**Other Strengths And Weaknesses:**

Here are the strengths:
- The paper presents a unified theoretical framework for NCE-based estimators, connecting prior methods such as MLE, MC-MLE, GlobalGISO, and pseudo-likelihood estimation within a single perspective (and objective).
- The work corrects a misconception about CondNCE and score matching, showing that its variance diverges in the small-noise regime.
- The paper establishes finite-sample convergence guarantees, while most prior works provided only asymptotic consistency results.

There are also some weaknesses:
- The paper is highly theoretical and does not have empirical validation. Since we have a concrete (convex) optimization objective here, I wonder if the authors could find an exponential family and conduct experiments for, say, different $\alpha$ values.
- The finite-sample convergence rates rely on bounded statistics assumptions, which may limit applicability to real-world unnormalized models. The authors also pointed this out in the final discussions.
- The claim that $\alpha$-CentNCE unifies multiple estimators lacks a discussion of practical trade-offs. For example, under what conditions would one prefer the estimator in the middle of the second row of Table 2 over others?

**Questions For Authors:**

Please look and respond to the questions and suggestions in the earlier sections. Thanks!

**Relation To Broader Scientific Literature:**

The paper builds on a rich body of noise-contrastive estimation (NCE) work. It generalizes and unifies several estimators previously developed in the research communities. For example, it connects MLE (Fisher, 1922), MC-MLE (Geyer, 1994), and GlobalGISO (Shah et al., 2023) as "special cases" of its proposed $\alpha$-centered NCE framework. Finally, the paper provides a finite-sample convergence guarantee for these estimators.

**Theoretical Claims:**

The main paper does not appear to contain proof sketches or outlines. Detailed proofs in the appendix should essentially follow the work of Shah et al. (2021b, 2023) and are left unchecked.

---

> ### Author Rebuttal · Authors · 2025-04-01
>
> We appreciate the reviewer’s effort in reviewing our manuscript. We will incorporate the constructive feedback in our revision.
>
> ### Comments on Weaknesses
>
> We acknowledge that our work primarily focuses on the theoretical unification of different estimators. As suggested by the reviewer, however, experiments involving certain exponential family distributions would indeed provide valuable insights into the behavior of the proposed estimators under various choices of $f$ and/or $\\alpha$, as well as for unbounded distributions that go beyond the boundedness assumption in our analysis. In our revision, we will include experiments in the Appendix on the convergence rates of the estimators for both bounded and unbounded distributions, as demonstrated in (Shah et al., 2023). Although a thorough study of the trade-offs associated with different choices of $\\alpha$ or $f$ warrants a separate investigation, we will include these experiments to offer an initial indication of such behaviors.
>
> ### On Suggestions
>
> We thank the reviewer for the thoughtful suggestions following a careful reading of our manuscript. We will incorporate these points in our revision, including proof sketches in the main text using an additional page.

---

### Decision · Program_Chairs · 2025-05-01

**Decision:**

Accept (poster)

**Comment:**

This paper presents a theoretical framework that unifies existing methods such as MLE, MC-MLE, and GlobalGISO for learning unnormalized distributions via noise-contrastive estimation (NCE). The reviewers appreciated the conceptual clarity, the finite-sample guarantees, and the correction of misconceptions in prior work. While the paper is theoretical in nature and does not include empirical results, we find the contributions to be meaningful and significant. We therefore recommend acceptance. In the revision, please supplement the paper with experimental results and outlines of the main proofs.